# Conformal Classification with Equalized Coverage for Adaptively Selected Groups

**Yanfei Zhou**
Department of Data Sciences and Operations
University of Southern California
Los Angeles, California, USA
yanfei.zhou@marshall.usc.edu

**Matteo Sesia**
Department of Data Sciences and Operations
University of Southern California
Los Angeles, California, USA
sesia@marshall.usc.edu

## Abstract

This paper introduces a conformal inference method to evaluate uncertainty in classification by generating prediction sets with valid coverage conditional on adaptively chosen features. These features are carefully selected to reflect potential model limitations or biases. This can be useful to find a practical compromise between efficiency—by providing informative predictions—and algorithmic fairness—by ensuring equalized coverage for the most sensitive groups. We demonstrate the validity and effectiveness of this method on simulated and real data sets.

## 1 Introduction

### 1.1 Uncertainty, Fairness, and Efficiency in Machine Learning

Increasingly sophisticated machine learning (ML) models, like deep neural networks, are revolutionizing decision-making in many high-stakes domains, including medical diagnostics [1], job screening [2], and recidivism prediction [3, 4]. However, serious concerns related to *uncertainty quantification* [5, 6] and *algorithmic fairness* [7–10] underscore the need for novel methods that can provide reliable and unbiased measures of confidence, applicable to any model.

Uncertainty quantification is crucial because ML models, although effective on average, can make errors while displaying overconfidence [11]. Consequently, in some situations users may lack sufficient warning about the potential unreliability of a prediction, raising trust and safety concerns. A promising solution is *conformal inference* [12–14], which enables converting the output of any model into prediction sets with precise coverage guarantees. These sets reflect the model's confidence on a case-by-case basis, with smaller sets indicating higher confidence in a specific prediction.

Algorithmic fairness focuses on the challenges of prediction inaccuracies that disproportionately impact specific groups, often identified by sensitive attributes like race, sex, and age. Among the many sources of algorithmic bias are training data that do not adequately represent the population's heterogeneity and a focus on maximizing average performance. However, fairness is partly subjective and lacks a universally accepted definition [15], leading to sometimes conflicting interpretations [16].

This complexity makes conformal inference with *equalized coverage* [17] an appealing approach. Equalized coverage aims to ensure that the prediction sets attain their coverage not only on average for the whole population (e.g., above 90%) but also at the same level within each group of interest. While this does not necessarily imply that the prediction sets will have equal size on average across different groups—since it is possible the predictive model may be more or less accurate for different groups—it objectively communicates the possible limitations of a model. This transparency helps decision-makers recognize when predictions may be less reliable for specific subgroups, allowing them to either avoid unnecessary actions or adopt more cautious strategies in cases of higher uncertainty, thereby minimizing the potential harm from inaccurate predictions.

A limitation of the method developed in [17] for conformal inference with equalized coverage is that it does not scale well to situations involving diverse populations with multiple sensitive attributes. In such cases, it necessitates splitting the data into exponentially many subsets, significantly reducing the effective sample size and leading to less informative predictions. Balancing this trade-off [18] between *efficiency*—aiming for highly informative predictions with small set sizes—and *fairness*—ensuring unbiased treatment—is challenging and requires novel approaches. This paper introduces a method to address this by providing equalized coverage conditional on carefully chosen features, informed by the model and data. While it cannot guarantee equalized coverage for *all* sensitive groups, it seeks a *reasonable compromise* with finite data sets, mitigating significant biases while retaining predictive power.

## 1.2 Background on Conformal Inference for Classification

Consider a data set comprising $n$ exchangeable (e.g., i.i.d.) observations $Z_i$ for $i \in \mathcal{D} := [n] := \{1, \ldots, n\}$, sampled from an arbitrary and unknown distribution $P_Z$. In classification, one can write $Z_i = (X_i, Y_i)$, where $Y_i \in [L] := \{1, \ldots, L\}$ is a categorical label and $X_i \in \mathcal{X}$ represents the individual's features, taking values in some space $\mathcal{X}$. As explained below, we will assume these features include some sensitive attributes. Further, we consider a test point $Z_{n+1} = (X_{n+1}, Y_{n+1})$, also sampled exchangeably from $P_Z$, and whose label $Y_{n+1} \in [L]$ has not yet been observed.

A standard goal for split conformal prediction methods is to quantify the predictive uncertainty of a given "black-box" ML model (e.g., pre-trained on an independent data set) by constructing a prediction set $\hat{C}(X_{n+1})$ for $Y_{n+1}$, guaranteeing *marginal coverage* at some desired level $\alpha \in (0, 1)$:

$$\mathbb{P}[Y_{n+1} \in \hat{C}(X_{n+1})] \geq 1 - \alpha. \tag{1}$$

This probability is taken over the randomness in $Y_{n+1}$ and $X_{n+1}$, as well as in the data indexed by $\mathcal{D}$. Intuitively, marginal coverage means the prediction sets are expected to cover the correct outcomes for a fraction $1 - \alpha$ of the population. However, this is not always satisfactory, especially if the miscoverage errors may disproportionately affect individuals characterized by well-defined features.

To address these concerns, one might consider *feature-conditional coverage*, $\mathbb{P}[Y_{n+1} \in \hat{C}(X_{n+1}) \mid X_{n+1} = x] \geq 1 - \alpha$ for all $x \in \mathcal{X}$. This would ensure consistent coverage for all possible test features $X_{n+1}$. However, it is impossible to achieve without additional assumptions, such as modeling the distribution $P_Z$ [19] or significantly restricting the feature space $\mathcal{X}$ [20]. Given that such assumptions may be unrealistic in real-world settings, exact feature-conditional coverage is typically unachievable.

Equalized coverage [17] seeks a practical middle ground between the two extremes of marginal and feature-conditional coverage, focusing on accounting for specific *discrete* attributes encapsulated by $X_{n+1}$. To facilitate the subsequent exposition of our method, it is useful to recall the definition of equalized coverage with the following notation.

Let $K$ denote the number of sensitive attributes, and for each $k \in [K]$ let $M_k \in \mathbb{N}$ count the possible values of the $k$-th attribute. Consider a function $\phi : \mathcal{X} \times \{0,1\}^K \to \mathbb{N}^d$ for any subset $A \subseteq [K]$ with $|A| = d$ elements, so that $\phi(x, A)$ is a vector of length $|A|$ representing the values of all attributes indexed by $A$ for an individual with features $x$. In the special case where $A$ is an empty set, $\phi$ returns a constant. If $A$ is a singleton, e.g., $A = \{k\}$ for some $k \in [K]$, then $\phi(x, \{k\}) \in [M_k]$ denotes the value of the $k$-th attribute; e.g., someone's academic degree. More generally, $\phi(x, \{k, l\}) \in [M_k] \times [M_l]$, for any distinct $k, l \in [K]$, denotes the joint values of two attributes, characterizing a smaller group, such as "males with a bachelor's degree.".

When multiple sensitive attributes are involved, i.e., $K > 1$, the concept of equalized coverage introduced by [17] can be naturally extended to *exhaustive equalized coverage*, defined as:

$$\mathbb{P}[Y_{n+1} \in \hat{C}(X_{n+1}) \mid \phi(X_{n+1}, [K])] \geq 1 - \alpha. \tag{2}$$

In words, this says $\hat{C}(X_{n+1})$ has valid coverage conditional on all $K$ sensitive attributes. Prediction sets satisfying (2) can be obtained by applying the standard conformal calibration method separately within each of the $M = \prod_{k=1}^{K} M_k$ groups characterized by a specific combination of the protected attributes represented by $\phi(X_i, [K])$; see Appendix A1 for details. However, a downside of this approach is that the calibration subsets may be too small if $M$ is large, leading to uninformative predictions for even moderate values of $K$. This limitation forms the starting point of our work.

## 1.3 Preview of Our Contributions: Adaptive Equalized Coverage

In practice, for a given model and data set, different groups may not exhibit the same need for rigorous equalized coverage guarantees (2), as conformal predictions may be able to approximately achieve the desired coverage even without explicit constraints. Algorithmic bias typically affects only a minority of the population, so standard prediction sets with marginal coverage (1) may approximately satisfy (2) for most groups. Therefore, we propose Adaptively Fair Conformal Prediction (AFCP), a new method that efficiently identifies and addresses groups suffering from algorithmic bias in a data-driven way, adjusting their prediction sets to equalize coverage without sacrificing informativeness.

AFCP involves two main steps. First, as sketched in Figure 1, it carefully selects a sensitive attribute $\hat{A}(X_{n+1}) \in \{\emptyset, \{1\}, \ldots, \{K\}\}$, based on $X_{n+1}$ and the data in $\mathcal{D}$. Although AFCP can be extended to select multiple attributes, we begin by focusing on this simpler version for clarity. Intuitively, AFCP searches for the attribute corresponding to the group most negatively affected by algorithmic bias. It may also opt to select no attribute ($\hat{A}(X_{n+1}) = \emptyset$) in the absence of significant biases.

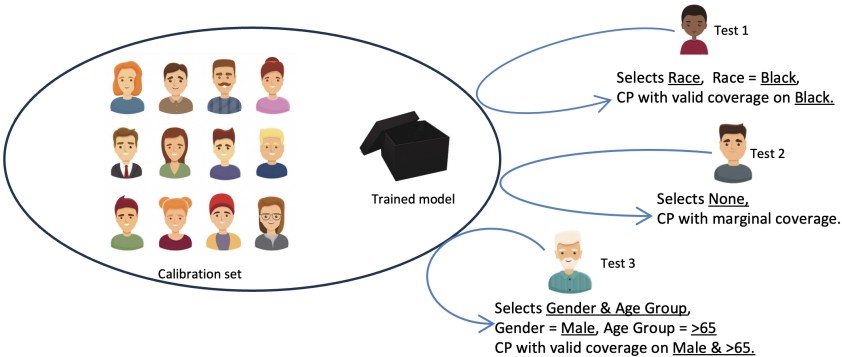

Figure 1: Schematic visualization of the automatic sensitive attribute selection carried out by our Adaptively Fair Conformal Prediction (AFCP) method. This method is designed to find the attribute corresponding to the group most negatively affected by algorithmic bias, on a case-by-case basis.

Next, AFCP constructs a prediction set $\hat{C}(X_{n+1})$ for $Y_{n+1}$ that guarantees the following notion of *adaptive equalized coverage* at the desired level $\alpha \in (0, 1)$:

$$\mathbb{P}[Y_{n+1} \in \hat{C}(X_{n+1}) \mid \phi(X_{n+1}, \hat{A}(X_{n+1}))] \geq 1 - \alpha. \tag{3}$$

In words, this tells us $\hat{C}(X_{n+1})$ is well-calibrated for the groups defined by the selected attribute $\hat{A}(X_{n+1})$. It is worth highlighting the key distinctions between (3) and the existing notions of coverage reviewed above. On the one hand, if AFCP identifies no significant bias, selecting $\hat{A}(X_{n+1}) = \emptyset$, then (3) reduces to marginal coverage (1), following the convention that $\phi(X_{n+1}, \emptyset)$ is a constant. On the other hand, exhaustive equalized coverage (2) would correspond to simultaneously selecting all possible sensitive attributes instead of only that identified by $\hat{A}(X_{n+1})$. To clarify the terminology, in this paper we will say that an attribute is *sensitive* if it may identify a group affected by algorithmic bias. By contrast, a *protected* attribute is one for which equalized coverage is explicitly sought.

Figure 2 illustrates this intuition through a simulated example. In this scenario, we generate synthetic medical diagnosis data, considering six possible diagnosis labels, and designate race, sex, and age group as potentially sensitive attributes alongside other demographic factors. Notably, the female group, identified by sex, is characterized by fewer samples and higher algorithmic bias, resulting in marginal prediction sets with low group-conditional coverage. By contrast, the model leads to no significant disparities across races and age groups in this dataset.

For two example patients from the critical group, the standard marginal prediction sets fail to cover the true label. Conversely, sets calibrated for exhaustive equalized coverage are too conservative to be informative. By contrast, AFCP generates prediction sets that are both efficient and fair.

Without additional sample splitting, which would be inefficient, constructing informative prediction sets that satisfy (3) is challenging due to potential selection bias from using the same data for attribute selection and conformal calibration. This paper presents a novel solution to address this challenge.

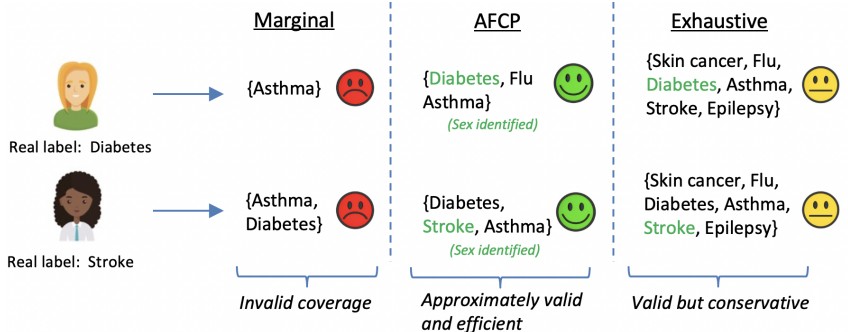

Figure 2: Prediction sets constructed with different methods for patients in groups negatively affected by algorithm bias. Our method (AFCP) is designed to provide informative prediction sets that are well-calibrated conditional on the automatically identified critical sensitive attribute.

## 1.4 Table of Contents

Section 2 presents our AFCP method, focusing on the special case in which at most one sensitive attribute may be selected. Section 3 demonstrates the empirical performance of AFCP on synthetic and real data. Section 4 discusses some limitations and suggests ideas for future work.

Additional content is presented in the Appendices. Appendix A1 reviews relevant details of existing approaches. Appendices A2 and A3 present two extensions of our method, respectively enabling the selection of more than one sensitive attribute and providing valid coverage also conditional on the true test label; both extensions involve distinct technical challenges. Additionally, a variation of AFCP designed for outlier detection tasks is detailed in Appendix A4. Appendix A5 contains all mathematical proofs. Appendix A6 explains how to implement our method efficiently and studies its computational cost. Appendix A7 describes the results of numerous additional experiments.

## 1.5 Related Works

Conformal inference is a very active research area, with numerous methods addressing diverse tasks, including outlier detection [21–23], classification [24–28], and regression [29–31]. Overcoming the limitations of the standard marginal coverage guarantees (1) is a main interest in this field.

Some works have proposed *conformity scores* designed to seek high feature-conditional coverage while calibrating prediction sets for marginal coverage [27, 30]. Others attempt to mitigate over-confidence while training the ML model [32, 33], and several have developed calibration methods for non-exchangeable data, accounting for possible distribution shifts [34–38]. These works are complementary to ours, as we focus on guaranteeing a new adaptive notion of equalized coverage.

In addition to [17], several other works have considered constructing prediction sets adhering to various notions of equalized coverage and have empirically investigated the performance of conformal predictors in this regard [39]. In the context of regression, [40] and [41] proposed strategies to enhance conditional coverage given several protected attributes, but they targeted a different notion of equalized coverage designed for continuous outcomes. In classification, a classical approach to move beyond marginal coverage is label-conditional coverage, where the "protected" groups are defined not based on the features $X_{n+1}$ but by the label itself, $Y_{n+1}$ [42–44]. As explained in Appendix A3, the method proposed in this paper can also be extended to provide label-conditional coverage.

More closely related to the notion of equalized coverage [17] are the works of [45, 46], which differ from ours as they do not consider the automatic selection of the sensitive groups. To tackle a related challenge due to unknown biased attributes, [47] studied how to identify unfairly treated groups by establishing a simultaneously valid confidence bound on group-wise disparities. In principle, their approach can be integrated within the selection component of our method. Very recently, [48] proposed an elegant method to obtain valid conformal prediction sets for adaptively selected subsets of test cases. While their perspective aligns more closely with ours, their approach and focus differ as they study different selection rules not specifically aimed at mitigating algorithmic bias.

## 2  Method

### 2.1  Automatic Attribute Selection

Given a pre-trained classifier, an independent calibration data set $\mathcal{D}$, and a test point $Z_{n+1} = (X_{n+1}, Y_{n+1})$ with an unknown label $Y_{n+1}$, we will select (at most) one sensitive attribute, $\hat{A}(X_{n+1}) \in \{\emptyset, \{1\}, \ldots, \{K\}\}$, according to the following *leave-one-out* procedure.

For each $y \in [L]$, imagine $Y_{n+1}$ is equal to $y$, and define an *augmented* calibration set $\mathcal{D}'_y := \mathcal{D} \cup \{(X_{n+1}, y)\}$. For each $i \in [n + 1]$, define also the leave-one-out set $\mathcal{D}'_{y,i} := \mathcal{D}'_y \setminus \{(X_i, Y_i)\}$, with $y$ acting as a placeholder for $Y_{n+1}$. Then, for each $i \in [n + 1]$, we construct a conformal prediction set $\hat{C}_y^{\text{loo}}(X_i)$ for $Y_i$ given $X_i$ by calibrating the classifier using the data in $\mathcal{D}'_{y,i}$. Any method can be applied for this purpose, although it may be helpful for concreteness to focus on employing the standard approach seeking marginal coverage (1) using the adaptive conformity scores proposed by [27]. Let $E_{y,i}$ denote the binary indicator of whether $\hat{C}_y^{\text{loo}}(X_i)$ fails to cover $Y_i$:

$$E_{y,i} := \mathbf{1}\{Y_i \notin \hat{C}_y^{\text{loo}}(X_i)\}. \tag{4}$$

After evaluating $E_{y,i}$ for all $i \in [n + 1]$, we will assess the leave-one-out miscoverage rate for the worst-off group identified by each sensitive attribute $k \in [K]$. That is, we evaluate

$$\delta_{y,k} := \max_{m \in [M_k]} \frac{\sum_{i=1}^{n+1} E_{y,i} \cdot \mathbf{1}\{\phi(X_i, \{k\}) = m\}}{\sum_{i=1}^{n+1} \mathbf{1}\{\phi(X_i, \{k\}) = m\}}.$$

Intuitively, $\delta_{y,k}$ denotes the maximum miscoverage rate across all groups identified by the $k$-th attribute. Large values of $\delta_{y,k}$ suggest that the $k$-th attribute may be a sensitive attribute corresponding to at least one group suffering from algorithmic bias.

To assess whether there is evidence of significant algorithmic bias, we can perform a statistical test for the null hypothesis that no algorithmic bias exists. Note that this test can be heuristic since it does not need to be exact for our method to rigorously guarantee (3). Therefore, we do not need to carefully consider the assumptions underlying this test. As a useful heuristic, we define:

$$\hat{q}_y := \max_{k \in [K]} \delta_{y,k}, \tag{5}$$

and carry out a one-sided t-test for the null hypothesis $H_0 : \hat{q}_y \leq \alpha$ against $H_1 : \hat{q}_y > \alpha$.

If $H_0$ is rejected (at any desired level, like 5%), we conclude there exists a group suffering from significant algorithmic bias, and we identify the corresponding attribute through

$$\hat{A}(X_{n+1}, y) = \{\arg\max_{k \in [K]} \delta_{y,k}\}. \tag{6}$$

Otherwise, we set $\hat{A}(X_{n+1}, y) = \emptyset$, which corresponds to selecting no attribute. See Algorithm 1 for an outline of this procedure, as a function of the placeholder label $y$.

After repeating this procedure for each $y \in [L]$, the final selected attribute $\hat{A}(X_{n+1})$ is:

$$\hat{A}(X_{n+1}) = \cap_{y \in [L]} \hat{A}(X_{n+1}, y). \tag{7}$$

Therefore, an attribute is selected if and only if it is consistently flagged by our leave-one-out procedure for all values of the placeholder label $y \in [L]$. This approach minimizes the potential arbitrariness due to the use of a placeholder label and is necessary to guarantee that our method constructs prediction sets achieving (3), as discussed in the next section.

Before explaining how our method utilizes the selected sensitive attribute obtained in (7) to construct prediction sets satisfying (3), we pause to make two remarks. First, as long as $n$ is large enough, $\hat{A}(X_{n+1}, y)$ is quite stable with respect to both $X_{n+1}$ and $y$, as each of these variables plays a relatively small role in determining the leave-one-out miscoverage rates. Therefore, the selected attribute $\hat{A}(X_{n+1})$ given by (7) is also quite stable for different values of $X_{n+1}$. This stability will be demonstrated empirically in Section 3. Second, despite its iterative nature, our method can be implemented efficiently; see Appendix A6. Further, if $n$ is very large, our method could be streamlined using cross-validation instead of a leave-one-out approach.

---

**Algorithm 1** Automatic attribute selection using a placeholder test label.

---

1: **Input**: calibration data $\mathcal{D}$; test point with features $X_{n+1}$; list of $K$ sensitive attributes;
2:        pre-trained classifier $\hat{f}$; fixed rule for computing nonconformity scores; level $\alpha \in (0,1)$;
3:        placeholder label $y \in [L]$.
4: Assume $Y_{n+1} = y$ and define the augmented data set $\mathcal{D}'_y := \mathcal{D} \cup \{(X_{n+1}, y)\}$.
5: **for** $i \in [n+1]$ **do**
6:    Pretend that $(X_i, Y_i)$ is the test point and $\mathcal{D}'_y \setminus \{(X_i, Y_i)\}$ is the calibration set.
7:    Construct a conformal prediction set $\hat{C}^{\text{loo}}_y(X_i)$ for $Y_i$.
8:    Evaluate the miscoverage indicator $E_{y,i}$ using (4).
9: **end for**
10: Perform a one-sided test for $H_0 : \hat{q}_y \le \alpha$ vs. $H_1 : \hat{q}_y > \alpha$, with $\hat{q}_y$ defined as in (5).
11: Select the attribute $\hat{A}(X_{n+1}, y)$ using (6) if $H_0$ is rejected, else set $\hat{A}(X_{n+1}, y) = \emptyset$.
12: **Output**: $\hat{A}(X_{n+1}, y)$, either a selected sensitive attribute or an empty set.

---

## 2.2 Constructing the Adaptive Prediction Sets

After evaluating $\hat{A}(X_{n+1}, y)$ by applying Algorithm 1 with placeholder label $y$ for $Y_{n+1}$ for all $y \in [L]$, and selecting either an empty set or a single attribute $\hat{A}(X_{n+1})$ using (7), AFCP constructs an adaptive prediction set for $Y_{n+1}$ that satisfies (3) as follows.

First, it constructs a *marginal* conformal prediction set $\hat{C}^{\text{m}}(X_{n+1})$ targeting (1), by applying the standard approach reviewed in Appendix A1. Then, for each $y \in [L]$, it constructs a conformal prediction set $\hat{C}(X_{n+1}, \hat{A}(X_{n+1}, y))$ with equalized coverage for the group identified by attribute $\hat{A}(X_{n+1}, y)$, as if it had been fixed. This is achieved by applying the standard marginal method based on a restricted calibration sample indexed by $\{i \in [n] : \phi(X_i, \hat{A}(X_{n+1}, y)) = \phi(X_{n+1}, \hat{A}(X_{n+1}, y))\}$; see Algorithm A1 in Appendix A1 for further details. Therefore, note that $\phi(X_i, \hat{A}(X_{n+1}, y))$ becomes equivalent to $\hat{C}^{\text{m}}(X_{n+1})$ if $\hat{A}(X_{n+1}, y) = \emptyset$. Finally, the AFCP prediction set for $Y_{n+1}$ is given by:

$$\hat{C}(X_{n+1}) = \hat{C}^{\text{m}}(X_{n+1}) \cup \left\{ \cup_{y=1}^{L} \hat{C}(X_{n+1}, \hat{A}(X_{n+1}, y)) \right\}. \tag{8}$$

See Algorithm 2 for an outline of this procedure.

Note that the AFCP set $\hat{C}(X_{n+1})$ given by (8) always contains the marginal set $\hat{C}^{\text{m}}(X_{n+1})$; this is essential to prove the validity of our approach. Second, in practice the selection $\hat{A}(X_{n+1}, y)$ tends to be very consistent for different values of the placeholder label $y$, as long as the sample size $n$ is large enough; therefore, the union in (8) will typically not lead to a very large prediction set.

---

**Algorithm 2** Adaptively Fair Conformal Prediction (AFCP).

---

1: **Input**: calibration data $\mathcal{D}$; test point with features $X_{n+1}$; list of $K$ sensitive attributes;
2:        pre-trained classifier $\hat{f}$; fixed rule for computing nonconformity scores; level $\alpha \in (0,1)$.
3: **for** $y \in [L]$ **do**
4:    Select an attribute $\hat{A}(X_{n+1}, y)$ by applying Algorithm 1 with placeholder label $y$.
5:    Construct $\hat{C}(X_{n+1}, A)$ by applying Algorithm A1 with the attribute $A = \hat{A}(X_{n+1}, y)$.
6: **end for**
7: Construct $\hat{C}^{\text{m}}(X_{n+1})$ by applying Algorithm A1 without protected attributes.
8: **Output**: selected attribute $\hat{A}(X_{n+1})$ given by (7) and prediction set $\hat{C}(X_{n+1})$ given by (8).

---

The following result, proved in Appendix A5, establishes that the prediction sets $\hat{C}(X_{n+1})$ output by AFCP guarantee adaptive equalized coverage (3) with respect to the adaptively selected attribute $\hat{A}(X_{n+1})$. It is worth emphasizing this result is not straightforward and involves an innovative proof technique to address the lack of exchangeability introduced by the adaptive selection step.

**Theorem 1.** *If* $\{(X_i, Y_i)\}_{i=1}^{n+1}$ *are exchangeable, the prediction set* $\hat{C}(X_{n+1})$ *and the selected attribute* $\hat{A}(X_{n+1})$ *output by Algorithm 2 satisfy the* adaptive equalized coverage *defined in* (3).

# 3 Numerical Experiments

## 3.1 Setup and Benchmarks

This section demonstrates the empirical performance of AFCP, focusing on the implementation described in Section 2, which selects at most one sensitive attribute. Our method is compared with three existing approaches, which utilize the same data, ML model, and conformity scores but produce prediction sets with different guarantees. The first is the *marginal* benchmark, which constructs prediction sets guaranteeing (1) by applying Algorithm A1 without protected attributes. The second is the *exhaustive* equalized benchmark, which constructs prediction sets guaranteeing (2) by applying Algorithm A1 with all $K$ sensitive attributes simultaneously protected. The third is a *partial* equalized benchmark that separately applies Algorithm A1 with each possible protected attribute $k \in [K]$, and then takes the union of all such prediction sets. This is an intuitive approach that can be easily verified to provide a coverage guarantee intermediate between (2) and (3), namely:

$$\mathbb{P}[Y_{n+1} \in \hat{C}(X_{n+1}) \mid \phi(X_{n+1}, \{k\})] \geq 1 - \alpha, \quad \forall k \in [K]. \tag{9}$$

However, we will see that these prediction sets are often still too conservative in practice.

In addition, we apply a variation of AFCP that always selects one sensitive attribute, regardless of the outcome of the significance test. This method is denoted as AFCP1 in our experiments.

For all methods considered, the classifier is based on a five-layer neural network with linear layers interconnected via a ReLU activation function. The output layer uses a softmax function to estimate the conditional label probabilities. The Adam optimizer and cross-entropy loss function are used in the training process, with a learning rate set at 0.0001. The loss values demonstrate convergence after 100 epochs of training. For all methods, the miscoverage target level is set at $\alpha = 0.1$.

## 3.2 Synthetic Data

We generate synthetic classification data to mimic a medical diagnosis task with six possible labels: Skin cancer, Diabetes, Asthma, Stroke, Flu, and Epilepsy. The available features include three sensitive attributes—Age Group, Region, and Color—and six additional non-sensitive covariates. Color is categorized as Blue or Grey, with 10% and 90% marginal frequencies, respectively. The Age Group is cyclically repeated as $< 18, 18 - 24, 25 - 40, 41 - 65, > 65$, and Region is sampled from an i.i.d. multinomial distribution across {West, East, North, South} with equal probabilities. The six non-sensitive features are i.i.d. random samples from a uniform distribution on $[0, 1]$. For simplicity, Color is denoted as $X_0$ and the first non-sensitive feature as $X_1$. Conditional on $X$, the label $Y$ is generated based on a decision tree model that depends only on $X_0$ and $X_1$, as detailed in Appendix A7. This model is designed so that the diagnosis label for individuals with Color equal to Blue is intrinsically harder to predict, mimicking the presence of algorithmic bias.

Figure 3 shows the performance of all methods as a function of the total sample size, ranging from 200 to 2000. In each case, $50\%$ of the samples are used for training and the remaining $50\%$ for calibration. Results are averaged over 500 test points and 100 independent experiments.

While the marginal benchmark produces the smallest prediction sets on average, it leads to significant empirical undercoverage within the Blue group. In contrast, the exhaustive benchmark, which achieves the highest coverage overall, tends to lead to overly conservative and thus uninformative prediction sets, especially for the Blue group. The partial benchmark, though less conservative than the exhaustive method, still generates prediction sets that are too large when the sample size is small.

Our AFCP method and its simpler variation, AFCP1, not only achieve valid coverage for the Blue group but do so with prediction sets that, on average, are not much larger than the marginal ones. AFCP1 is slightly more robust than AFCP when the sample size is very small, as it never fails to select a sensitive attribute. This is advantageous in scenarios where we know there is a sensitive attribute worth equalizing coverage for, though this may not always be the case in practice. See Figure A1 and Table A1 for detailed results with standard errors.

Figure 4 provides additional insight into our method's performance by plotting the selection frequencies of each sensitive attribute as a function of sample size, within the same experiments described in Figure 3. These results show that our method behaves as anticipated. When the sample size and algorithmic bias are both small, AFCP shows more variability in selecting the sensitive attribute,

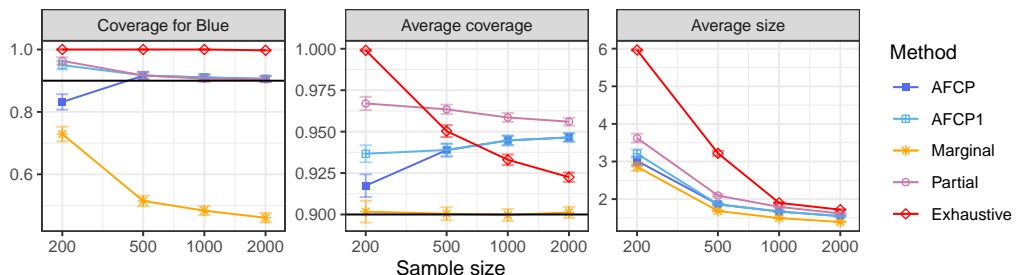

Figure 3: Performance of conformal prediction sets constructed by different methods on synthetic medical diagnosis data, as a function of the total number of training and calibration data points. Our method (AFCP) leads to more informative prediction sets (smaller average size) with more effective mitigation of algorithmic bias (higher conditional coverage). The error bars indicate 2 standard errors.

often selecting no attributes. However, as the sample size grows and the undercoverage affecting the Blue group becomes more pronounced, AFCP consistently selects Color as the most sensitive attribute, correctly identifying the main manifestation of algorithmic bias in these data.

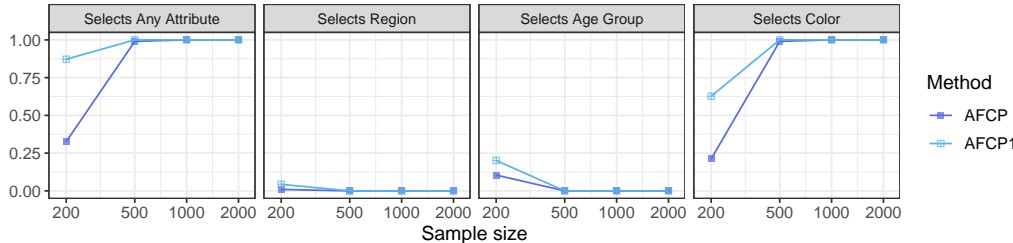

Figure 4: Selection frequency of different attributes using our AFCP method and its variation, AFCP1, in the experiments of Figure 3. As the sample size increases, AFCP becomes more consistent in selecting the most relevant attribute, Color.

Additional results are presented in Appendix A7.1.1. Figures A1–A3 and Tables A2–A4 summarize the average coverage and prediction set size conditional on each protected attribute. Figures A4–A7 and Tables A5–A8 study the performance of an extension of our method that also provides valid coverage conditional on the true label of the test point.

### 3.3 Nursery data

We apply AFCP and its benchmarks to the open-domain *Nursery* data set [49], which was derived from a hierarchical decision model originally developed to rank applications for nursery schools for social science studies. The data encompass 12,960 instances with eight categorical features: Parents' occupation (3 levels), Child's nursery (5 levels), Family form (4 levels), Number of children (1, 2, 3, or more), Housing conditions (3 levels), Financial standing (2 levels), Social conditions (3 levels), and Health status (3 levels). These features are used to predict application ranks across five categories. The variables "Parents' occupation", "Number of children", "Financial standing", "Social conditions", and "Health status" are marked as possible sensitive attributes.

To prepare for model training, we executed several pre-processing steps. First, two instances labeled "recommend" were removed due to the minimal occurrence of this outcome label. Subsequently, we utilized the LabelEncoder function in the sklearn Python package to numerically encode all features and labels. To make the problem more interesting and allow control over the strength of algorithmic bias, we added independent, uniformly distributed noise to the labels of samples with Parents' occupation in the first category, rounding these perturbed labels to the nearest integer. This makes the group corresponding to the first category of Parents' occupation intrinsically more unpredictable and hence more prone to algorithmic bias. To enhance the challenge of making accurate predictions for this group, it was further down-sampled to 10% of its original size.

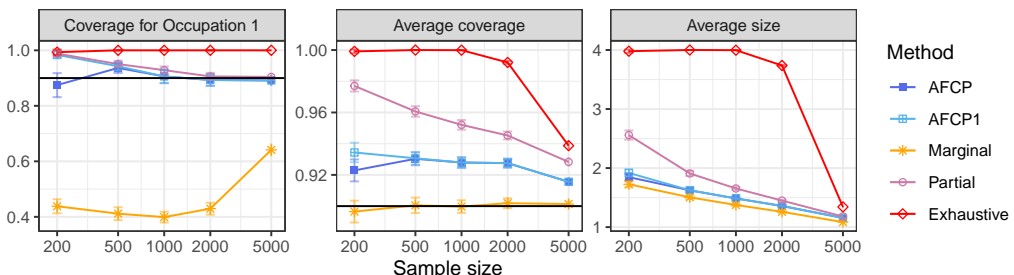

Figure 5: Performance of prediction sets constructed by different methods on the Nursery data, as a function of the sample size. AFCP leads to more informative predictions with higher coverage conditional on the sensitive attribute, Parents' occupation (shown explicitly for level one).

Figure 5 summarizes the performance of all approaches as a function of the total number of training and calibration data points, which varies between 200 and 5000. The results are averaged over 500 randomly chosen test points and 100 repeated experiments. In each experiment, 50% of the samples are randomly assigned for training and the remaining 50% for calibration. The marginal benchmark is heavily biased, leading to prediction sets with very low coverage for samples with Parents' occupation in the first category. The exhaustive benchmark is too conservative and results in very large prediction sets unless the sample size is very large. The partial benchmark performs better than the other two benchmarks, but our AFCP method still outperforms it, producing smaller prediction sets with valid coverage even for the hardest-to-predict group. See Table A9 for detailed results.

The results of additional experiments are presented in Appendix A7.1.2. Figures A8–A12 and Tables A10–A14 detail the average coverage and prediction set size for each sensitive attribute. Additionally, Figures A13–A18 and Tables A15–A20 report on the performance of an extended version of AFCP which also ensured valid coverage conditional on the true test label.

## 3.4 Additional Numerical Experiments

Figures A19–A21 and Table A21 in Appendix A7.1.3 summarize additional experimental results using the open-source COMPAS dataset [50]. Moreover, Appendix A7.2 demonstrates the empirical performance of our AFCP extension for outlier detection; see Figures A22–A36 and Tables A22–A34. These demonstrations involve both synthetic data and the open-domain Adult Income dataset [51]. The experiments with the real-world Adult Income data also include an AFCP extension that allows for the selection of multiple sensitive attributes at the same time.

## 3.5 Performance of AFCP with Different Sample Sizes

Constructing informative prediction sets that achieve high conditional coverage is inherently more challenging when dealing with smaller sample sizes. By experimenting with different sample sizes, we demonstrate that our AFCP method consistently performs well across different scenarios. For instance, in the experiments depicted in Figures 3–4, when the sample size is as small as 200, it is difficult to fit an accurate predictive model, assess conditional coverage, and reliably identify the sensitive attribute associated with the lowest coverage. This difficulty is reflected in the relatively large sizes of the prediction sets produced by all methods and the substantial discrepancies between the nominal and empirical conditional coverage. Despite these challenges, our method often succeeds in selecting the correct sensitive attribute, achieving significantly higher conditional coverage compared to the Marginal benchmark, with only a slight increase in average prediction set size.

Moreover, as the sample size increases, our method becomes highly effective at identifying the attribute associated with the lowest conditional coverage, as illustrated in Figure 4. Consequently, our method is able to achieve high conditional coverage with relatively small prediction sets. Overall, these experiments demonstrate that our method offers distinct advantages over existing approaches in both large-sample and small-sample settings.

### 3.6 Comparison between AFCP and AFCP1

Both AFCP and AFCP1 outperform the benchmark approaches when applied to datasets with small sample sizes, each excelling in different scenarios. AFCP is better suited to situations where there is uncertainty regarding the presence of significant algorithmic bias, while AFCP1 is more effective when prior knowledge suggests that at least one attribute may be biased. For example, in Figure 3, which illustrates a case where one group (Color-Blue) is consistently biased, AFCP1 achieves slightly higher conditional coverage than AFCP. While AFCP exhibits slight undercoverage for the blue group with small sample sizes, it still outperforms the Marginal approach. The occasional inability of AFCP to select a sensitive attribute in small samples reflects the inherent challenges posed by limited datasets. When the method does not select an attribute, it often signifies a lack of sufficient evidence of algorithmic bias, making it reasonable to calibrate the prediction sets solely for marginal coverage.

## 4   Discussion

This paper presents a practical and statistically principled method to construct informative conformal prediction sets with valid coverage conditional on adaptively selected features. This approach balances efficiency and equalized coverage, which may be particularly useful in applications involving multiple sensitive attributes. While we believe it offers substantial benefits, a potential limitation of this method is that it does not always identify the most relevant sensitive attribute, particularly when working with limited sample sizes. Nevertheless, our empirical results are quite encouraging, demonstrating that AFCP effectively mitigates significant instances of algorithmic bias when the sample size is adequate. Moreover, our method is flexible, allowing for the integration of prior knowledge about which sensitive attributes might require protection against algorithmic bias.

This paper creates several opportunities for further work. Future research could focus on theoretically studying the conditions under which our method can be guaranteed to select the correct sensitive attribute with high probability. Additionally, future extensions might explore implementing different attribute selection procedures, such as those inspired by [47] - within our flexible AFCP framework to delve into the subtle trade-offs associated with different selection algorithms. Moreover, adapting our approach to accommodate different fairness criteria by adaptively adjusting the coverage rate target for each subgroup is another promising area of study. Extending our method to more efficiently handle scenarios with an extremely high number of possible classes is also worthwhile, potentially drawing inspiration from [43]. Furthermore, investigating extensions for classification tasks where the target variable is ordered could be both intriguing and practically useful. In such cases, a naive modification of our method would involve utilizing the discrete convex hull of all components instead of unions of subintervals on the right-hand side of Equation (8). However, developing a more refined approach would be a valuable contribution for future work. Future extensions of our work could focus on adapting to distributional shifts or enhancing the robustness and efficiency of our method under adversarial attacks or contaminated data, potentially drawing connections with [52–56]. Finally, extending our method to accommodate regression tasks with continuous outcomes presents additional computational challenges, but potential solutions could be inspired by [33].

The numerical experiments described in this paper were carried out on a computing cluster. Individual experiments, involving 1000 calibration samples and 500 test samples, required less than 25 minutes and 5GB of memory on a single CPU. The entire project took approximately 100 hours of computing time, and did not involve preliminary or failed experiments.

Software implementing the algorithms and data experiments are available online at `https://github.com/FionaZ3696/Adaptively-Fair-Conformal-Prediction`.

### Acknowledgements

The authors thank anonymous reviewers for helpful comments, and the Center for Advanced Research Computing at the University of Southern California for providing computing resources. M. S. and Y. Z. were partly supported by NSF grant DMS 2210637. M. S. was also partly supported by an Amazon Research Award.

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

## A1  Review of Existing Conformal Classification Methods

Algorithm A1 outlines the standard approach for constructing conformal prediction sets with equalized coverage with respect to a fixed list of protected attributes [17]. In the special case where the list of protected attributes is empty, this method reduces to the standard approach for constructing prediction sets with marginal coverage.

---

**Algorithm A1** Conformal classification with equalized coverage for fixed protected attributes.

---

1: **Input**: calibration data $\mathcal{D}$; test point with features $X_{n+1}$; list of protected attributes $A$;
2:       pre-trained classifier $\hat{f}$; pre-defined rule for computing nonconformity scores;
3:       nominal level $\alpha \in (0, 1)$.
4: Define the calibration subset

$$\mathcal{I}(X_{n+1}, A) = \{i \in [n] : \phi(X_i, A) = \phi(X_{n+1}, A)\}.$$

5: **for** $y \in [L]$ **do**
6:     Compute the nonconformity scores $\hat{S}_i^y$ for $i \in \mathcal{I}(X_{n+1}, A) \cup \{(X_{n+1}, y)\}$ using $\hat{f}$.
7:     Compute the conformal p-value:

$$\hat{u}^y(X_{n+1}) = \frac{1 + |i \in \mathcal{I}(X_{n+1}, A) : \hat{S}_i^y \leq \hat{S}_{n+1}^y|}{1 + |\mathcal{I}(X_{n+1}, A)|}.$$

8: **end for**
9: Construct a prediction set using $\hat{C}(X_{n+1}) = \{y \in [L] : \hat{u}^y(X_{n+1}) \geq \alpha\}$.
10: **Output**: a prediction set $\hat{C}(X_{n+1})$.

---

In the context of outlier detection, Algorithm A2 reviews the standard approach for computing conformal p-values achieving valid false positive rate (FPR) control conditional a fixed list of protected attributes.

---

**Algorithm A2** Conformal p-value with equalized FPR for fixed protected attributes.

---

1: **Input**: calibration data $\mathcal{D}$; test point $Z_{n+1}$; list of protected attributes $A$;
2:       pre-trained one-class classifier $\hat{f}$; pre-defined rule for computing nonconformity scores;
3:       nominal level $\alpha \in (0, 1)$.
4: Define the calibration subset

$$\mathcal{I}(Z_{n+1}, A) = \{i \in [n] : \phi(Z_i, A) = \phi(Z_{n+1}, A)\}.$$

5: Compute the nonconformity scores $\hat{S}_i$ for $i \in \mathcal{I}(Z_{n+1}, A) \cup \{Z_{n+1}\}$ using $\hat{f}$.
6: Compute the conformal p-value:

$$\hat{u}(Z_{n+1}) = \frac{1 + |i \in \mathcal{I}(Z_{n+1}, A) : \hat{S}_i \leq \hat{S}_{n+1}|}{1 + |\mathcal{I}(Z_{n+1}, A)|}.$$

7: **Output**: a conformal p-value $\hat{u}(Z_{n+1})$.

---

## A2  Methodology Extension: AFCP with Multiple Selected Attributes

This section introduces an extension of AFCP that enables the selection of more than one sensitive attribute. For simplicity, we focus on the selection of up to two attributes. The methodology for selecting more than two attributes can be extended in a similar manner, as explained later.

### A2.1  Automatic Multiple Attribute Selections

Given a pre-trained classification model, an independent calibration data set $\mathcal{D}$ with size $n$, and a test point $Z_{n+1} = (X_{n+1}, Y_{n+1})$ with an unknown label $Y_{n+1}$, Algorithm 1 in Section 2.1 introduces

the AFCP component to select one sensitive attribute according to the *leave-one-out* procedure. Intuitively, selecting two sensitive attributes requires executing Algorithm 1 twice. However, during the second iteration, the sensitive attribute list is restricted to exclude the most critical protected attribute selected in the first round.

Specifically, for each placeholder label $y \in [L]$, assuming $Y_{n+1} = y$, Algorithm 1 is run with all $K$ sensitive attributes for the first iteration to obtain the first selected attribute $\hat{A}^1(X_{n+1}, y) \in \{\emptyset, \{1\}, \ldots, \{K\}\}$. If $\hat{A}^1(X_{n+1}, y) \neq \emptyset$, Algorithm 1 is run again using the sensitive attributes $[K] \setminus \hat{A}^1(X_{n+1}, y)$ and one can get the second selected attribute $\hat{A}^2(X_{n+1}, y) \in \{\emptyset, \{1\}, \ldots, \{K\}\} \setminus \hat{A}^1(X_{n+1}, y)$. Therefore, the identified attributes for test feature $X_{n+1}$ with placeholder $y$ for the test label is the union of the two $\hat{A}(X_{n+1}, y) = \hat{A}^1(X_{n+1}, y) \cup \hat{A}^2(X_{n+1}, y)$. This procedure is outlined in Algorithm A3.

After repeating this procedure for each $y \in [L]$, the final selected attribute $\hat{A}(X_{n+1})$ is:

$$\hat{A}(X_{n+1}) = \cap_{y \in [L]} \hat{A}(X_{n+1}, y). \tag{A10}$$

---

**Algorithm A3** Two attributes selection using a placeholder test label.

1: **Input**: calibration data $\mathcal{D}$; test point with features $X_{n+1}$; list of $K$ sensitive attributes;
2:        pre-trained classifier $\hat{f}$; pre-defined rule for computing nonconformity scores;
3:        nominal level $\alpha \in (0, 1)$, placeholder label $y \in [L]$.
4: Select the first attribute $\hat{A}^1(X_{n+1}, y)$ by applying Algorithm 1 with placeholder label $y$ and sensitive attributes $[K]$.
5: **if** $\hat{A}^1(X_{n+1}, y) \neq \emptyset$ **then**
6:    Select the second attribute $\hat{A}^2(X_{n+1}, y)$ by applying Algorithm 1 with placeholder label $y$ and sensitive attributes $[K] \setminus \hat{A}^1(X_{n+1}, y)$.
7: **end if**
8: **Output**: $\hat{A}(X_{n+1}, y) = \hat{A}^1(X_{n+1}, y) \cup \hat{A}^2(X_{n+1}, y)$, which is a set of an empty set, or a set including one or two selected sensitive attribute(s).

---

### A2.2   Constructing the Adaptive Prediction Sets

After selecting a subset of attributes $\hat{A}(X_{n+1})$ using (A10), which may be empty, or include one or two attributes, AFCP constructs an adaptive prediction set for $Y_{n+1}$ that satisfies (3) as follows.

First, it constructs a marginal conformal prediction set $\hat{C}^{\mathrm{m}}(X_{n+1})$ targeting (1), by applying Algorithm A1 without protected attributes. Then, for each $y \in [L]$, it constructs a conformal prediction set $\hat{C}(X_{n+1}, \hat{A}(X_{n+1}, y))$ with equalized coverage for the group *jointly* identified by attributes $\hat{A}(X_{n+1}, y)$. This can be achieved by applying Algorithm A1 with protected attributes $\hat{A}(X_{n+1}, y)$. Lastly, it constructs a conformal prediction set $\hat{C}^{\mathrm{eq}}(X_{n+1}, \ell)$ with equalized coverage separately for each protected attribute $\ell \in \cup_{y \in [L]} \hat{A}(X_{n+1}, y)$, by applying the standard approach in Algorithm A1 on the subsets indexed by $\{i \in [n] : \phi(X_i, \{\ell\}) = \phi(X_{n+1}, \{\ell\})\}$.

Finally, the AFCP prediction set constructed using up to two selected attributes is given as:

$$\hat{C}(X_{n+1}) = \hat{C}^{\mathrm{m}}(X_{n+1}) \cup \left\{ \cup_{y=1}^{L} \hat{C}(X_{n+1}, \hat{A}(X_{n+1}, y)) \right\} \cup \left\{ \cup_{\ell \in \cup_y \hat{A}(X_{n+1}, y)} \hat{C}^{\mathrm{eq}}(X_{n+1}, \ell) \right\}. \tag{A11}$$

See Algorithm A4 for an outline of this AFCP extension.

**Theorem A1.** *If $\{(X_i, Y_i)\}_{i=1}^{n+1}$ are exchangeable random samples, then the conformal prediction set $\hat{C}(X_{n+1})$ and the selected attributes $\hat{A}(X_{n+1})$ output by Algorithm A4 satisfy the adaptive equalized coverage defined in* (3).

---

**Algorithm A4** AFCP with two selected attributes.

---

1: **Input**: calibration data $\mathcal{D}$; test point with features $X_{n+1}$; list of $K$ sensitive attributes;
2:        pre-trained classifier $\hat{f}$; pre-defined rule for computing nonconformity scores;
3:        nominal level $\alpha \in (0, 1)$.
4: **for** $y \in [L]$ **do**
5:     Select attribute(s) $\hat{A}(X_{n+1}, y)$ by applying Algorithm A3 with placeholder label $y$.
6:     Construct $\hat{C}(X_{n+1}, A)$ by applying Algorithm A1 with protected attribute(s) $A = \hat{A}(X_{n+1}, y)$.
7: **end for**
8: **for** $\ell \in \cup_{y \in [L]} \hat{A}(X_{n+1}, y)$ **do**
9:     Construct $\hat{C}^{\mathrm{eq}}(X_{n+1}, \ell)$ by applying Algorithm A1 with protected attribute $\{\ell\}$.
10: **end for**
11: Construct $\hat{C}^{\mathrm{m}}(X_{n+1})$ by applying Algorithm A1 without protected attributes.
12: Define the final selected attribute(s) $\hat{A}(X_{n+1})$ using Equation (A10).
13: Define the final prediction set $\hat{C}(X_{n+1})$ using Equation (A11).
14: **Output**: $\hat{A}(X_{n+1})$ and $\hat{C}(X_{n+1})$.

---

## A3   Methodology Extension: AFCP with Label Conditional Coverage

This section extends the AFCP method with adaptive equalized coverage that is also conditional on the true test label. We focus on the main implementation of AFCP, where it can select up to one sensitive attribute.

First, the label conditional counterparts of the marginal coverage, the exhaustive equalized coverage, and the adaptive equalized coverage are defined. The label-conditional counterpart of marginal coverage is defined as:

$$\mathbb{P}[Y_{n+1} \in \hat{C}(X_{n+1}) \mid Y_{n+1} = y] \geq 1 - \alpha, \quad \forall y \in [L]. \tag{A12}$$

Intuitively, this coverage ensures that the prediction sets constructed are valid for each group with the test label $y$ for all possible values of $y \in [L]$. This coverage offers a stronger assurance than marginal coverage in classification contexts. However, it overlooks scenarios where groups, identified by feature attributes, may suffer adverse effects from prediction biases. To address these concerns, one can aim for label-conditional exhaustive equalized coverage [17], defined as:

$$\mathbb{P}[Y_{n+1} \in \hat{C}(X_{n+1}) \mid Y_{n+1} = y, \phi(X_{n+1}, [K])] \geq 1 - \alpha, \quad \forall y \in [L]. \tag{A13}$$

Achieving this label conditional exhaustive equalized coverage involves applying the standard conformal classification method (outlined in Algorithm A1) separately within each of the groups characterized by every possible combination of sensitive attributes and test labels. Hence, this approach can become overly conservative when a large number of sensitive attributes or response labels are presented.

Our AFCP method strikes a balance between the two approaches to achieve label-conditional adaptive equalized coverage:

$$\mathbb{P}[Y_{n+1} \in \hat{C}(X_{n+1}) \mid Y_{n+1} = y, \phi(X_{n+1}, \hat{A}(X_{n+1}))] \geq 1 - \alpha, \quad \forall y \in [L], \tag{A14}$$

which guarantees that the true test label is contained within the prediction sets with high probability for the groups defined by the selected attribute $\hat{A}(X_{n+1})$ and the test label $y$ for every $y \in [L]$.

### A3.1   Automatic Attribute Selection

Given a pre-trained classification model and an independent calibration data set $\mathcal{D}$ with size $n$, for each placeholder label $y \in [L]$ for $Y_{n+1}$, the AFCP method with label-conditional adaptive equalized coverage (A14) selects the sensitive attribute $\hat{A}(X_{n+1}, y)$ by simply applying Algorithm 1 using the *label-restricted calibration data* $\mathcal{D}_y = \{i \in [n] : Y_i = y\}$. After repeating this process for each $y \in [L]$, the final selected attribute $\hat{A}(X_{n+1})$ is again given by

$$\hat{A}(X_{n+1}) = \cap_{y \in [L]} \hat{A}(X_{n+1}, y). \tag{A15}$$

### A3.2 Constructing the Adaptive Prediction Sets

After selecting $\hat{A}(X_{n+1}, y)$ by applying Algorithm 1 with placeholder label $y$ based on the label-restricted calibration data $\mathcal{D}_y = \{i \in [n] : Y_i = y\}$, and selecting a sensitive attribute $\hat{A}(X_{n+1})$, AFCP constructs an adaptive prediction set for $Y_{n+1}$ that satisfies (A14) as follows.

For each $y \in [L]$, it firstly construct a conformal prediction set $\hat{C}^{\mathrm{lc}}(X_{n+1}, y)$ by applying Algorithm A1 using the label-restricted calibration set $\mathcal{D}_y$ without considering protected attributes. Then, it constructs another conformal prediction set $\hat{C}(X_{n+1}, \hat{A}(X_{n+1}, y))$ with equalized coverage for the group identified by both the selected attribute $\hat{A}(X_{n+1}, y)$ *and label* $y$. This can be achieved by applying Algorithm A1 based on a subset of the calibration samples indexed by $\mathcal{I}(X_{n+1}, y) = \{i \in \mathcal{D}_y : \phi(X_i, \hat{A}(X_{n+1}, y)) = \phi(X_{n+1}, \hat{A}(X_{n+1}, y))\}$. Lemma A1 shows that, for any given placeholder label, the prediction set constructed in this step satisfies the label conditional adaptive equalized coverage as long as the selected variable using that placeholder label is fixed.

**Lemma A1.** *If $\{(X_i, Y_i)\}_{i=1}^{n+1}$ are exchangeable and the selected attribute $\hat{A}(X_{n+1}, y)$ is fixed for some placeholder label $y$, then, the prediction set $\hat{C}(X_{n+1}, \hat{A}(X_{n+1}, y))$ constructed by calibrating on $\mathcal{I}(X_{n+1}, y)$ satisfies*

$$\mathbb{P}[Y_{n+1} \in \hat{C}(X_{n+1}, \hat{A}(X_{n+1}, y)) \mid Y_{n+1} = \tilde{y}, \phi(X_{n+1}, \hat{A}(X_{n+1}, y))] \geq 1 - \alpha,$$

*for any $\tilde{y} \in [L]$.*

Lastly, the final AFCP prediction set is obtained by:

$$\hat{C}(X_{n+1}) = \left\{\cup_{y=1}^{L} \hat{C}^{\mathrm{lc}}(X_{n+1}, y)\right\} \cup \left\{\cup_{y=1}^{L} \hat{C}(X_{n+1}, \hat{A}(X_{n+1}, y))\right\}. \tag{A16}$$

The procedures to form AFCP prediction sets with the label-conditional adaptive equalized coverage (A14) is summarized in Algorithm A5.

---

**Algorithm A5** AFCP with label-conditional adaptive equalized coverage (A14).

---

1: **Input**: calibration data $\mathcal{D}$; test point with features $X_{n+1}$; list of $K$ sensitive attributes;
2:        pre-trained classifier $\hat{f}$; pre-defined rule for computing nonconformity scores;
3:        nominal level $\alpha \in (0, 1)$.
4: **for** $y \in [L]$ **do**
5:      Define the label-restricted calibration set $\mathcal{D}_y = \{i \in [n] : Y_i = y\}$.
6:      Select an attribute $\hat{A}(X_{n+1}, y)$ by applying Algorithm 1 with placeholder label $y$ on $\mathcal{D}_y$.
7:      Construct $\hat{C}(X_{n+1}, \hat{A}(X_{n+1}, y))$ by applying Algorithm A1 with protected attributes $\hat{A}(X_{n+1}, y)$ on $\mathcal{D}_y$.
8:      Construct $\hat{C}^{\mathrm{lc}}(X_{n+1}, y)$ by applying Algorithm A1 on $\mathcal{D}_y$.
9: **end for**
10: Define the final selected attribute $\hat{A}(X_{n+1})$ using Equation (A15).
11: Define the final prediction set $\hat{C}(X_{n+1})$ using Equation (A16).
12: **Output**: $\hat{A}(X_{n+1})$ and $\hat{C}(X_{n+1})$.

---

**Theorem A2.** *If $\{(X_i, Y_i)\}_{i=1}^{n+1}$ are exchangeable random samples, then the conformal prediction set $\hat{C}(X_{n+1})$ and the selected attribute $\hat{A}(X_{n+1})$ output by Algorithm A5 satisfy the label-conditional adaptive equalized coverage defined in (A14).*

## A4 Methodology Extension: AFCP for Outlier Detection

Consider a dataset $\mathcal{D} = \{Z_i\}_{i=1}^{n}$ containing $n$ sample points drawn exchangeably from an unknown distribution $P_Z$. Consider an additional test point $Z_{n+1}$. In the outlier detection problems, our AFCP method aims to study whether $Z_{n+1} \sim P_Z$ by constructing a valid conformal p-value $\hat{u}(Z_{n+1})$ conditional on the group identified by the selected attribute $\hat{A}(Z_{n+1})$, that is:

$$\mathbb{P}[\hat{u}(Z_{n+1}) \leq \alpha \mid \phi(Z_{n+1}, \hat{A}(Z_{n+1}))] \leq \alpha, \tag{A17}$$

for any $\alpha \in (0,1)$. Intuitively, this guarantees that, on groups defined by the selected attribute $\hat{A}(Z_{n+1})$, the conformal p-value is super-uniform, therefore controlling the FPR (the probability of rejecting the null hypothesis that $Z_{n+1}$ is an inlier when it is true) below $\alpha$.

### A4.1   Automatic Attribute Selection

Given a pre-trained one-class classifier $\hat{f}$, an independent calibration dataset $\mathcal{D}$, and a test point $Z_{n+1}$, our method selects a sensitive attribute $\hat{A}(Z_{n+1}) \in \{\emptyset, \{1\}, \ldots, \{K\}\}$ according to the following *leave-one-out* procedure. Sometimes, no attribute may be selected, as denoted by $\hat{A}(Z_{n+1}) = \emptyset$.

Define an *augmented* calibration set $\mathcal{D}' := \mathcal{D} \cup \{Z_{n+1}\}$. For each $i \in [n+1]$, define also the leave-one-out set $\mathcal{D}_i' := \mathcal{D}' \setminus \{Z_i\}$. Then, for each $i \in [n+1]$, we compute a conformal p-value $\hat{u}^{\mathrm{loo}}(Z_i)$ for $Z_i$ using the data in $\mathcal{D}_i'$ to test if $Z_i$ is an outlier. This can be accomplished by running Algorithm A2, using $\mathcal{D}_i'$ as the calibration data and $Z_i$ as the test point, and with the convention that smaller nonconformity scores suggest $Z_i$ is more likely to be an outlier. Any nonconformity scores can be utilized here. For concreteness, we focus on using the adaptive conformity scores proposed by [27]. Small $\hat{u}^{\mathrm{loo}}(Z_i)$ provides stronger evidence to reject the null hypothesis $H_{0,i} : Z_i$ is an inlier. Let $E_i$ denote the binary indicator of whether $H_{0,i}$ is rejected:

$$E_i := \mathbf{1}\{\hat{u}^{\mathrm{loo}}(Z_i) \leq \alpha\}. \tag{A18}$$

After evaluating $E_i$ for all $i \in [n+1]$, we will assess the leave-one-out FPR for the worst-off group identified by each sensitive attribute $k \in [K]$. That is, we evaluate

$$\delta_k := \max_{m \in [M_k]} \frac{\sum_{i=1}^{n+1} E_i \cdot \mathbf{1}\{\phi(Z_i, \{k\}) = m\}}{\sum_{i=1}^{n+1} \mathbf{1}\{\phi(Z_i, \{k\}) = m\}}. \tag{A19}$$

Intuitively, $\delta_k$ denotes the maximum FPR across all groups identified by the $k$-th attribute, as estimated by the leave-one-out simulation carried out under the assumption that $Z_{n+1}$ is an inlier. Large values of $\delta_k$ suggest that the $k$-th attribute may be a sensitive attribute corresponding to at least one group suffering from algorithmic bias.

To assess whether there is evidence of significant algorithmic bias, we can perform a statistical test for the null hypothesis that no algorithmic bias exists. We define:

$$\hat{q} := \max_{k \in [K]} \delta_k, \tag{A20}$$

and carry out a one-sided t-test for the null hypothesis $H_0 : \hat{q} \leq \alpha$ against $H_1 : \hat{q} > \alpha$.

If $H_0$ is rejected (at any desired level, such as 5%), we conclude there exists a group suffering from significant algorithmic bias, and we identify the corresponding attribute through

$$\hat{A}(Z_{n+1}) = \{\arg\max_{k \in [K]} \delta_k\}. \tag{A21}$$

Otherwise, we set $\hat{A}(Z_{n+1}) = \emptyset$, which corresponds to selecting no attribute. See Algorithm A6 for an outline of this procedure.

### A4.2   Evaluating the Adaptive Conformal P-Value

After selecting either a single attribute or an empty set $\hat{A}(X_{n+1})$ that corresponds to at least one group suffering from algorithmic bias, the next step of our AFCP method is to compute an adaptive conformal p-value for testing whether $Z_{n+1}$ is an outlier that satisfies (A17). This can be simply achieved by applying the standard conformal method outlined in Algorithm A2 based on a restricted calibration sample indexed by $\mathcal{I}(\hat{A}(Z_{n+1})) = \{i \in [n] : \phi(Z_i, \hat{A}(Z_{n+1})) = \phi(Z_{n+1}, \hat{A}(Z_{n+1}))\}$, See Algorithm A7 for a summary of AFCP for outlier detection tasks.

**Theorem A3.** *If $\{Z_i\}_{i=1}^{n+1}$ are exchangeable random samples, the conformal p-value $\hat{u}(Z_{n+1})$ and the selected attribute $\hat{A}(Z_{n+1})$ output by Algorithm A7 satisfy* (A17).

---

**Algorithm A6** Automatic attribute selection for outlier detection.

---

1: **Input**: calibration data $\mathcal{D}$; test point $Z_{n+1}$; list of $K$ sensitive attributes;
2:         pre-trained one-class classifier $\hat{f}$; pre-defined rule for computing nonconformity scores;
3:         nominal level $\alpha \in (0, 1)$.
4: Define the augmented data set $\mathcal{D}' = \mathcal{D} \cup \{Z_{n+1}\}$.
5: **for** $i \in [n+1]$ **do**
6:     Pretend that $Z_i$ is the test point and $\mathcal{D}' \setminus \{Z_i\}$ is the calibration set.
7:     Compute a conformal p-value $\hat{u}^{\mathrm{loo}}(Z_i)$.
8:     Evaluate the false positive indicator $E_i$ using (A18).
9: **end for**
10: Compute $\hat{q}$ using (A20).
11: Perform a one-sided test for $H_0 : \hat{q} \le \alpha$ vs. $H_1 : \hat{q} > \alpha$.
12: Select the attribute $\hat{A}(Z_{n+1})$ using (A21) if $H_0$ is rejected, else set $\hat{A}(Z_{n+1}) = \emptyset$.
13: **Output**: $\hat{A}(Z_{n+1})$, either a selected sensitive attribute or an empty set.

---

---

**Algorithm A7** AFCP for outlier detection.

---

1: **Input**: calibration data $\mathcal{D}$; test point $Z_{n+1}$; list of $K$ sensitive attributes;
2:         pre-trained one-class classifier $\hat{f}$; pre-defined rule for computing nonconformity scores;
3:         nominal level $\alpha \in (0, 1)$.
4: Select a sensitive attribute $\hat{A}(Z_{n+1})$ by applying Algorithm A6.
5: Evaluate $\hat{u}(Z_{n+1})$ by applying Algorithm A2 with protected attribute $\hat{A}(Z_{n+1})$.
6: **Output**: $\hat{u}(Z_{n+1})$.

---

### A4.3 AFCP for Outlier Detection with Multiple Selected Attributes

AFCP for outlier detection problems can be readily extended to select $J$ sensitive attributes where $J > 1$. This is achieved by repeatedly applying the single attribute selection procedure described in Algorithm A6 $J$ times. Each time, the algorithm selects a (possibly empty) subset of attributes $\hat{A}(Z_{n+1})^j$ from the list of sensitive attributes excluding the previously selected attribute $\hat{A}(Z_{n+1})^{j-1}$. The final set of selected attributes is given by $\hat{A}(Z_{n+1}) = \cup_{j=1}^{J} \hat{A}(Z_{n+1})^j$. See Algorithm A8 for an outline of this procedure.

---

**Algorithm A8** Multiple attributes selection for outlier detection.

---

1: **Input**: calibration data $\mathcal{D}$; test point $Z_{n+1}$; list of $K$ sensitive attributes;
2:         pre-trained one-class classifier $\hat{f}$; pre-defined rule for computing nonconformity scores;
3:         nominal level $\alpha \in (0, 1)$; number of selected attributes $J$.
4: Denote $K^0 = [K]$.
5: **for** $j \in \{1, \ldots, J\}$ **do**
6:     Select an attribute $\hat{A}(Z_{n+1})^j$ by applying Algorithm A6 with sensitive attributes $K^{j-1}$.
7:     Update the list of sensitive attributes $K^j = K^{j-1} \setminus \hat{A}(Z_{n+1})^j$.
8: **end for**
9: **Output**: $\hat{A}(Z_{n+1}) = \cup_{j=1}^{J} \hat{A}(Z_{n+1})^j$, an empty set or a set of selected attributes.

---

After selecting a set of attributes $\hat{A}(Z_{n+1})$, which might be empty or include one or more sensitive attributes, AFCP constructs an adaptive conformal p-value satisfying (A17). This can be easily achieved by applying Algorithm A2 with protected attributes $\hat{A}(Z_{n+1})$. Algorithm A9 summarizes the AFCP implementation for outlier detection that allows selecting multiple protected attributes.

**Theorem A4.** *If $\{Z_i\}_{i=1}^{n+1}$ are exchangeable random samples, the conformal p-value $\hat{u}(Z_{n+1})$ and selected attributes $\hat{A}(Z_{n+1})$ output by Algorithm A9 satisfy* (A17).

---
**Algorithm A9** AFCP for outlier detection with multiple selected attributes.
___
 1: **Input**: calibration data $\mathcal{D}$; test point $Z_{n+1}$; list of $K$ sensitive attributes;
 2:       pre-trained one-class classifier $\hat{f}$; pre-defined rule for computing nonconformity scores;
 3:       nominal level $\alpha \in (0,1)$; number of selected attributes $J$.
 4: Select up to $J$ sensitive attributes $\hat{A}(Z_{n+1})$ by applying Algorithm A8.
 5: Evaluate $\hat{u}(Z_{n+1})$ by applying Algorithm A2 with protected attribute $\hat{A}(Z_{n+1})$.
 6: **Output**: $\hat{u}(Z_{n+1})$.
___

## A5   Mathematical Proofs

*Proof of Theorem 1.* Consider an imaginary oracle that has access to the true value of $Y_{n+1}$. Denote $\hat{A}^{\mathrm{o}}(X_{n+1}, Y_{n+1})$ as the sensitive attribute selected by this oracle by applying Algorithm 1 with the true $Y_{n+1}$ instead of a placeholder label. Let $\hat{C}^{\mathrm{o}}(X_{n+1}, \hat{A}^{\mathrm{o}}(X_{n+1}, Y_{n+1}))$ represent the corresponding output prediction set by applying Algorithm A1 with the protected attribute $\hat{A}^{\mathrm{o}}(X_{n+1}, Y_{n+1})$. Consider also $\hat{C}^{\mathrm{m}}(X_{n+1})$, the standard prediction set with marginal coverage (1).

The main idea of our proof is to connect the output prediction set $\hat{C}(X_{n+1})$ and selected attribute $\hat{A}(X_{n+1})$ from Algorithm 2 to those of the imaginary oracle described above. Throughout this proof, we adopt the convention that $\phi(X_{n+1}, \emptyset) = 0$.

To establish this connection, note that the attribute $\hat{A}(X_{n+1})$ selected by Algorithm 2 is either empty, $\hat{A}(X_{n+1}) = \emptyset$, or a singleton, $\hat{A}(X_{n+1}) = \{k\}$ for some $k \in [K]$. In the latter case, $\hat{A}(X_{n+1}) = \hat{A}(X_{n+1}, \tilde{y}), \forall \tilde{y} \in [L]$, and thus $\hat{A}(X_{n+1}) = \hat{A}^{\mathrm{o}}(X_{n+1}, Y_{n+1})$ almost-surely. Therefore,

$$
\begin{aligned}
&\mathbb{P}[Y_{n+1} \in \hat{C}(X_{n+1}) \mid \phi(X_{n+1}, \hat{A}(X_{n+1}))] \\
&\geq \min\Bigg\{ \mathbb{P}[Y_{n+1} \in \hat{C}(X_{n+1}) \mid \phi(X_{n+1}, \emptyset)], \\
&\qquad\qquad \mathbb{P}[Y_{n+1} \in \hat{C}(X_{n+1}) \mid \phi(X_{n+1}, \hat{A}^{\mathrm{o}}(X_{n+1}, Y_{n+1}))] \Bigg\} \\
&= \min\Bigg\{ \mathbb{P}[Y_{n+1} \in \hat{C}(X_{n+1})], \\
&\qquad\qquad \mathbb{P}[Y_{n+1} \in \hat{C}(X_{n+1}) \mid \phi(X_{n+1}, \hat{A}^{\mathrm{o}}(X_{n+1}, Y_{n+1}))] \Bigg\} \\
&\geq \min\Bigg\{ \mathbb{P}[Y_{n+1} \in \hat{C}^{\mathrm{m}}(X_{n+1})], \\
&\qquad\qquad \mathbb{P}[Y_{n+1} \in \hat{C}^{\mathrm{o}}(X_{n+1}, \hat{A}^{\mathrm{o}}(X_{n+1}, Y_{n+1})) \mid \phi(X_{n+1}, \hat{A}^{\mathrm{o}}(X_{n+1}, Y_{n+1}))] \Bigg\},
\end{aligned}
\tag{A22}
$$

where the last inequality follows from the facts that $\hat{C}^{\mathrm{m}}(X_{n+1}) \subseteq \hat{C}(X_{n+1})$ and $\hat{C}^{\mathrm{o}}(X_{n+1}, \hat{A}^{\mathrm{o}}(X_{n+1}, Y_{n+1})) \subseteq \hat{C}(X_{n+1})$ almost-surely. Next, we only need to separately lower-bound by $1 - \alpha$ the two terms on the right-hand-side of (A22).

The first part of the remaining task is trivial. It is already well-known that $\mathbb{P}(Y_{n+1} \in \hat{C}^{\mathrm{m}}(X_{n+1})) \geq 1 - \alpha$; see [13, 57].

To complete the second part of the remaining task, note that the oracle-selected attribute $\hat{A}^{\mathrm{o}}(X_{n+1}, Y_{n+1})$ is invariant to any permutations of the exchangeable data indexed by $[n+1]$. Therefore, the data points are also exchangeable conditional on the groups defined by $\hat{A}^{\mathrm{o}}(X_{n+1}, Y_{n+1})$. This means we can imagine the protected attribute $\hat{A}^{\mathrm{o}}(X_{n+1}, Y_{n+1})$ is fixed, and the oracle prediction set $\hat{C}^{\mathrm{o}}(X_{n+1}, \hat{A}^{\mathrm{o}}(X_{n+1}, Y_{n+1}))$ is simply obtained by applying Algorithm A1 to exchangeable data

using a fixed protected attribute. This procedure is the same as the main algorithm in [17] and has guaranteed coverage above $1 - \alpha$; see Theorem 1 in [17]. $\qquad\square$

*Proof of Theorem A1.* Similar to the proof of Theorem 1, consider an imaginary oracle that has access to the true value of $Y_{n+1}$. Denote $\hat{A}^{\text{o}}(X_{n+1}, Y_{n+1})$ as the (possibly empty, one, or two) sensitive attribute(s) selected by this oracle by applying Algorithm A3 with the true $Y_{n+1}$. Let $\hat{C}^{\text{o}}(X_{n+1}, \hat{A}^{\text{o}}(X_{n+1}, Y_{n+1}))$ represent the output prediction set by applying Algorithm A1 with the protected attribute(s) $\hat{A}^{\text{o}}(X_{n+1}, Y_{n+1})$. Consider also $\hat{C}^{\text{m}}(X_{n+1})$, the standard prediction set with marginal coverage (1), and $\hat{C}^{\text{eq}}(X_{n+1}, \ell)$ the prediction set with valid coverage conditional on groups identified by a fixed attribute $\{\ell\}$.

The key idea of our proof is again to connect the practical prediction set $\hat{C}(X_{n+1})$ and selected protected attributes $\hat{A}(X_{n+1})$ of Algorithm A4 to those of the imaginary oracle described above. Throughout this proof, we adopt the convention that $\phi(X_{n+1}, \emptyset) = 0$.

To establish this connection, note that the attribute(s) $\hat{A}(X_{n+1})$ selected by applying Algorithm A3 falls into one of the three possible cases almost surely: (a) $\hat{A}(X_{n+1}) = \emptyset$, (b) $\hat{A}(X_{n+1}) = \hat{A}^{\text{o}}(X_{n+1}, Y_{n+1})$, and (c) $\dot{A}(X_{n+1}) = \hat{A}(X_{n+1}) \subset \hat{A}^{\text{o}}(X_{n+1}, Y_{n+1})$. The last scenario happens when the oracle selects two attributes, and the $\hat{A}(X_{n+1})$ contains only one of them. For clarify of the notation, we denote the last case using $\dot{A}(X_{n+1})$.

Therefore,

$$\mathbb{P}[Y_{n+1} \in \hat{C}(X_{n+1}) \mid \phi(X_{n+1}, \hat{A}(X_{n+1}))]$$

$$\geq \min\Bigg\{ \mathbb{P}[Y_{n+1} \in \hat{C}(X_{n+1}) \mid \phi(X_{n+1}, \emptyset)],$$

$$\mathbb{P}[Y_{n+1} \in \hat{C}(X_{n+1}) \mid \phi(X_{n+1}, \hat{A}^{\text{o}}(X_{n+1}, Y_{n+1}))],$$

$$\mathbb{P}[Y_{n+1} \in \hat{C}(X_{n+1}) \mid \phi(X_{n+1}, \dot{A}(X_{n+1}))] \Bigg\}$$

$$= \min\Bigg\{ \mathbb{P}[Y_{n+1} \in \hat{C}(X_{n+1})],$$

$$\mathbb{P}[Y_{n+1} \in \hat{C}(X_{n+1}) \mid \phi(X_{n+1}, \hat{A}^{\text{o}}(X_{n+1}, Y_{n+1}))], \qquad \text{(A23)}$$

$$\mathbb{P}[Y_{n+1} \in \hat{C}(X_{n+1}) \mid \phi(X_{n+1}, \dot{A}(X_{n+1}))] \Bigg\}$$

$$\geq \min\Bigg\{ \mathbb{P}[Y_{n+1} \in \hat{C}^{\text{m}}(X_{n+1})],$$

$$\mathbb{P}[Y_{n+1} \in \hat{C}^{\text{o}}(X_{n+1}, \hat{A}^{\text{o}}(X_{n+1}, Y_{n+1})) \mid \phi(X_{n+1}, \hat{A}^{\text{o}}(X_{n+1}, Y_{n+1}))],$$

$$\mathbb{P}[Y_{n+1} \in \cup_{\ell \in \hat{A}^{\text{o}}(X_{n+1}, Y_{n+1})} \hat{C}^{\text{eq}}(X_{n+1}, \ell) \mid \phi(X_{n+1}, \dot{A}(X_{n+1}))] \Bigg\},$$

where the last inequality follows from the facts that $\hat{C}^{\text{m}}(X_{n+1}) \subseteq \hat{C}(X_{n+1})$, $\hat{C}^{\text{o}}(X_{n+1}, \hat{A}^{\text{o}}(X_{n+1}, Y_{n+1})) \subseteq \hat{C}(X_{n+1})$, and $\cup_{\ell \in \hat{A}^{\text{o}}(X_{n+1}, Y_{n+1})} \hat{C}^{\text{eq}}(X_{n+1}, \ell) \subseteq \hat{C}(X_{n+1})$ almost-surely.

Next, we show how to separately lower-bound by $1 - \alpha$ the three terms on the right-hand-side of (A23).

The lower bounds of the first and second terms have been proved in theorem 1. We will focus on lower-bounding the coverage of the third term. Because the oracle-selected attribute(s) $\hat{A}^{\text{o}}(X_{n+1}, Y_{n+1})$ are invariant to any permutation of the exchangeable data indexed by $[n + 1]$, we can treat each of the two elements in $\hat{A}^{\text{o}}(X_{n+1}, Y_{n+1})$ as fixed. Without loss of generality, let $\ell_1$ and $\ell_2$ denote the first

and the second element respectively, $\dot{A}(X_{n+1}) = \ell_1$ or $\dot{A}(X_{n+1}) = \ell_2$ with probability 1. Then,

$$\mathbb{P}[Y_{n+1} \in \cup_{\ell \in \hat{A}^\circ(X_{n+1}, Y_{n+1})} \hat{C}^{eq}(X_{n+1}, \ell) \mid \phi(X_{n+1}, \dot{k}(X_{n+1}))]$$

$$\geq \min\left\{ \mathbb{P}[Y_{n+1} \in \cup_{\ell \in \hat{A}^\circ(X_{n+1}, Y_{n+1})} \hat{C}^{eq}(X_{n+1}, \ell) \mid \phi(X_{n+1}, \ell_1)], \right.$$

$$\left. \mathbb{P}[Y_{n+1} \in \cup_{\ell \in \hat{A}^\circ(X_{n+1}, Y_{n+1})} \hat{C}^{eq}(X_{n+1}, \ell) \mid \phi(X_{n+1}, \ell_2)] \right\}$$

$$\geq \min\left\{ \mathbb{P}[Y_{n+1} \in \hat{C}^{eq}(X_{n+1}, \ell_1) \mid \phi(X_{n+1}, \ell_1)], \right.$$

$$\left. \mathbb{P}[Y_{n+1} \in \hat{C}^{eq}(X_{n+1}, \ell_2) \mid \phi(X_{n+1}, \ell_2)] \right\}$$

$$\geq \min\{1 - \alpha, 1 - \alpha\},$$

where the inequality of the last line of (A24) is proved in [17] with fixed attribute $\ell_1$ and $\ell_2$ respectively. Lastly, realize that $\cup_{\ell \in \hat{A}^\circ(X_{n+1}, Y_{n+1})} \hat{C}^{eq}(X_{n+1}, \ell) \subseteq \cup_{\ell \in \cup_{y=1}^{L} \hat{A}(X_{n+1}, y)} \hat{C}^{eq}(X_{n+1}, \ell) \subseteq \hat{C}(X_{n+1})$ almost-surely, the proof is completed. $\qquad \square$

*Proof of Theorem A2.* Similar to the proof of Theorem 1, consider an imaginary oracle that has access to the true value of $Y_{n+1}$. Denote $\hat{A}^\circ(X_{n+1}, Y_{n+1})$ as the sensitive attribute selected by this oracle by applying Algorithm 1 based on a subset of the calibration data indexed by $\mathcal{D}_{Y_{n+1}} = \{i \in [n] : Y_i = Y_{n+1}\}$. Let $\hat{C}^\circ(X_{n+1}, \hat{A}^\circ(X_{n+1}, Y_{n+1}))$ represent the output prediction set by applying Algorithm A1 with the protected attribute $\hat{A}^\circ(X_{n+1}, Y_{n+1})$. Consider also $\hat{C}^{lc}(X_{n+1}, Y_{n+1})$, the prediction set with label-conditional coverage obtained by running Algorithm A1 using $\mathcal{D}_{Y_{n+1}}$ without any protected attributes.

To establish this connection between the output prediction set $\hat{C}(X_{n+1})$ and the selected attribute $\hat{A}(X_{n+1})$ of Algorithm A5 and these of the imaginary oracle, note that the selected attribute $\hat{A}(X_{n+1})$ must be $\hat{A}(X_{n+1}) = \emptyset$ or $\hat{A}(X_{n+1}) = \hat{A}^\circ(X_{n+1}, Y_{n+1})$.

Therefore, for any $y \in [L]$,

$$\mathbb{P}[Y_{n+1} \in \hat{C}(X_{n+1}) \mid Y_{n+1} = y, \phi(X_{n+1}, \hat{A}(X_{n+1}))]$$

$$\geq \min\left\{ \mathbb{P}(Y_{n+1} \in \hat{C}(X_{n+1}) \mid Y_{n+1} = y, \phi(X_{n+1}, \emptyset)), \right.$$

$$\left. \mathbb{P}(Y_{n+1} \in \hat{C}(X_{n+1}) \mid Y_{n+1} = y, \phi(X_{n+1}, \hat{A}^\circ(X_{n+1}, Y_{n+1}))) \right\}$$

$$= \min\left\{ \mathbb{P}(Y_{n+1} \in \hat{C}(X_{n+1}) \mid Y_{n+1} = y), \right.$$

$$\left. \mathbb{P}(Y_{n+1} \in \hat{C}(X_{n+1}) \mid Y_{n+1} = y, \phi(X_{n+1}, \hat{A}^\circ(X_{n+1}, Y_{n+1}))) \right\}$$

$$\geq \min\left\{ \mathbb{P}(Y_{n+1} \in \hat{C}^{lc}(X_{n+1}, Y_{n+1}) \mid Y_{n+1} = y), \right.$$

$$\left. \mathbb{P}(Y_{n+1} \in \hat{C}^\circ(X_{n+1}, \hat{A}^\circ(X_{n+1}, Y_{n+1})) \mid Y_{n+1} = y, \phi(X_{n+1}, \hat{A}^\circ(X_{n+1}, Y_{n+1}))) \right\},$$

$$(A25)$$

where the last inequality follows from the facts that $\hat{C}^{lc}(X_{n+1}, Y_{n+1}) \subseteq \hat{C}(X_{n+1})$ and $\hat{C}^\circ(X_{n+1}, \hat{A}^\circ(X_{n+1}, Y_{n+1})) \subseteq \hat{C}(X_{n+1})$ almost-surely.

Next, we only need to separately lower-bound by $1 - \alpha$ the two terms on the right-hand-side of (A25).

The first part of the remaining task is trivial. It is already well-known that $\mathbb{P}(Y_{n+1} \in \hat{C}^{\text{lc}}(X_{n+1}, Y_{n+1}) \mid Y_{n+1} = y) \geq 1 - \alpha$; see [13, 47, 58].

Next, we prove the lower bound for the second term. Fix any $y \in [L]$, and assume $Y_{n+1} = y$. First, note that the oracle-selected attribute $\hat{A}^{\text{o}}(X_{n+1}, Y_{n+1})$ is invariant to any permutations of the exchangeable data indexed by $\mathcal{D}_y \cup \{X_{n+1}, y\}$. Therefore, the data points are exchangeable conditional on the group identified by the oracle-selected attribute $\hat{A}^{\text{o}}(X_{n+1}, Y_{n+1})$. This means that we can imagine the protected attribute $\hat{A}^{\text{o}}(X_{n+1}, Y_{n+1})$ is fixed, and the oracle prediction set $\hat{C}^{\text{o}}(X_{n+1}, \hat{A}^{\text{o}}(X_{n+1}, Y_{n+1}))$ is simply obtained by applying Algorithm A1 to exchangeable data using a fixed protected attribute. This procedure has guaranteed coverage above $1 - \alpha$; see Lemma A1. $\square$

*Proof of Theorem A3.* The strategy of this proof is very standard in the conformal inference literature. We add this proof for completeness.

Recall that $\mathcal{I}(\hat{A}(Z_{n+1}))$ denotes a subset of the calibration data $\mathcal{D}$ that has the same value of the selected attribute as the test point. To prove (A17), it suffices to show that the nonconformity scores $\{\hat{S}_i : i \in \mathcal{I}(\hat{A}(Z_{n+1})) \cup \{Z_{n+1}\}\}$ are exchangeable. Indeed, if the scores are exchangeable and almost surely distinct (which can be easily achieved by adding continuous random noises), then the rank of $\hat{S}_{n+1}$ is uniformly distributed over the discrete values $\{1, 2, \ldots, \mathcal{I}(\hat{A}(Z_{n+1})) + 1\}$. Consequently, the conformal p-value $\hat{u}(Z_{n+1})$ constructed by Algorithm A7 follows a uniform distribution $\text{Unif}(\{\frac{1}{|\mathcal{I}(\hat{A}(Z_{n+1}))|+1}, \frac{2}{|\mathcal{I}(\hat{A}(Z_{n+1}))|+1}, \ldots, 1\})$. This implies that $\mathbb{P}(\hat{u}(Z_{n+1}) \leq \alpha \mid \phi(Z_{n+1}, \hat{A}(Z_{n+1}))) = \alpha$. Even if the nonconformity scores are not almost surely distinct, one can still verify that the distribution of $\hat{u}(Z_{n+1})$ is super-uniform. Combining both cases, we have $\mathbb{P}(\hat{u}(Z_{n+1}) \leq \alpha | \phi(Z_{n+1}, \hat{A}(Z_{n+1}))) \leq \alpha$ for any $\alpha \in (0, 1)$.

We complete the proof by showing that the nonconformity scores $\{\hat{S}_i : i \in \mathcal{I}(\hat{A}(Z_{n+1})) \cup \{Z_{n+1}\}\}$ are exchangeable. Define $\sigma$ as an arbitrary permutation function applied on $\mathcal{D} \cup \{Z_{n+1}\}$ and denote the permuted dataset as $\sigma(\mathcal{D})$. We first run Algorithm A7 based on $\mathcal{D}$ to select the sensitive attribute $\hat{A}(Z_{n+1})$. Next, assume in a parallel world, we repeat Algorithm A7 with the same parameters and seed settings but based on the permuted data $\sigma(\mathcal{D})$. Denote the selected attribute in the parallel world as $\hat{A}'(Z_{n+1})$. Essentially, $\hat{A}'(Z_{n+1}) = \hat{A}(Z_{n+1})$. This is because the computation of conformal p-values and the procedure of selecting the attribute with the worst FPR in Algorithm A6 are not affected by the order of the calibration and test data. Therefore, the attribute selection process is invariant to the order of $\mathcal{D} \cup \{Z_{n+1}\}$. This implies that the attribute selected by Algorithm A6 $\hat{A}(Z_{n+1})$ can be treated as fixed, and the nonconformity scores computed on the subset $\mathcal{I}(\hat{A}(Z_{n+1})) \cup \{Z_{n+1}\}$ are simply reordered in the parallel world, i.e.,

$$\{\hat{S}'_{\sigma(i)} : i \in \mathcal{I}(\hat{A}(Z_{n+1})) \cup \{Z_{n+1}\}\} = \bar{\sigma}(\{\hat{S}_i : i \in \mathcal{I}(\hat{A}(Z_{n+1})) \cup \{Z_{n+1}\}\}),$$

where $\bar{\sigma}$ is the permutation obtained by restricting $\sigma$ on $\mathcal{I}(\hat{A}(Z_{n+1})) \cup \{Z_{n+1}\}$. Hence, we have

$$\{\hat{S}_i : i \in \mathcal{I}(\hat{A}(Z_{n+1})) \cup \{Z_{n+1}\}\} \stackrel{d}{=} \{\hat{S}'_{\sigma(i)} : i \in \mathcal{I}(\hat{A}(Z_{n+1})) \cup \{Z_{n+1}\}\}$$
$$= \bar{\sigma}(\{\hat{S}_i : i \in \mathcal{I}(\hat{A}(Z_{n+1})) \cup \{Z_{n+1}\}\}),$$

where the equality in distribution is implied by $\mathcal{D} \stackrel{d}{=} \sigma(\mathcal{D})$. $\square$

*Proof of Theorem A4.* This proof is the same as the proof of Theorem A3 since the multiple selected attributes $\hat{A}(Z_{n+1})$ are invariant to any permutation of the calibration and test data. $\square$

*Proof of Lemma A1.* This proof is a minor extension of the proof of Theorem 1 in [17], with the difference that we additionally condition on the true test label.

Fix any $\tilde{y} \in [L]$, and suppose $Y_{n+1} = \tilde{y}$. For a placeholder label $y$, consider a subset of the calibration data $\mathcal{D}$ indexed by $\mathcal{I}(X_{n+1}, y) = \{i \in \mathcal{D} : Y_i = \tilde{y}, \phi(X_i, \hat{A}(X_{n+1}, y)) = \phi(X_{n+1}, \hat{A}(X_{n+1}, y))\}$.

Since $\{(X_i, Y_i)\}_{i=1}^{n+1}$ are exchangeable and the protected attribute $\hat{A}(X_{n+1}, y)$ is fixed, the nonconformity scores $\hat{S}_i$ evaluated on the subset $\mathcal{I}(X_{n+1}, y) \cup \{(X_{n+1}, y)\}$ are also exchangeable.

Further, denote $\hat{Q}(X_{n+1}, y)$ as the $\lceil (1-\alpha) \cdot |1 + \mathcal{I}(X_{n+1}, y)| \rceil$-th smallest value of $\{\hat{S}_i\}_{i \in \mathcal{I}(X_{n+1}, y)}$. By the quantile lemma [13, 17, 57], for any $\alpha \in (0, 1)$,

$$
\begin{aligned}
\mathbb{P}[Y_{n+1} &\in \hat{C}(X_{n+1}, \hat{A}(X_{n+1}, y)) \mid Y_{n+1} = \tilde{y}, \phi(X_{n+1}, \hat{A}(X_{n+1}, y))] \\
&= \mathbb{P}[\hat{S}_{n+1} \le \hat{Q}(X_{n+1}, y) \mid Y_{n+1} = \tilde{y}, \phi(X_{n+1}, \hat{A}(X_{n+1}, y))] \\
&\ge 1 - \alpha.
\end{aligned}
\tag{A26}
$$

$\square$

## A6 Computational Shortcuts and Efficient Implementation

### A6.1 Outlier Detection

Given a pre-trained one-class classifier, consider a calibration dataset $\mathcal{D}$ of size $n$. Let $K$ denote the number of sensitive attributes, and for each attribute $k \in [K]$, let $M_k \in \mathbb{N}$ denote the count of its possible values. Denote $M = \max_k M_k$ as the maximum count across all attributes.

AFCP for outlier detection outlined in Algorithm A7 has the following computational cost.

**Analysis for a single test point**

- The cost of computing the false positive indicators for every sample in $\mathcal{D} \cup \{Z_{n+1}\}$ takes $\mathcal{O}(n \cdot \log n)$. Breaking into steps, for every data in $\mathcal{D} \cup \{Z_{n+1}\}$, compute their nonconformity scores takes $\mathcal{O}(n)$. Their associated conformal p-value can be computed at once by sorting all scores and keeping track of their ranks, which takes $\mathcal{O}(n \cdot \log n)$.
- Then, selecting the sensitive attributes requires $\mathcal{O}(n \cdot K \cdot M)$.
- Once the attribute is selected, the cost of applying conformal prediction conditional on the group identified by the selected attribute is $\mathcal{O}(1)$ because the subset of the group has been found in the last step.

Hence, the total cost of running Algorithm A7 for a single test point is $\mathcal{O}(n \log n + nKM)$.

**Analysis for $m$ test points**

- The cost of computing the false positive indicators for every sample in $\mathcal{D} \cup \{Z_{n+t}\}_{t=1}^{m}$ takes $\mathcal{O}(n \cdot (m + \log n))$. This can be derived by rewriting the conformal p-values for all $j \in \mathcal{D} \cup \{Z_{n+t}\}$ and for all $t \in [m]$ as follows:

$$
\begin{aligned}
\hat{u}_{j,t} &= \frac{1}{1+n} \Big( \sum_{i \in \mathcal{D} \cup \{Z_{n+t}\}} \mathbf{1}(\hat{S}_i \le \hat{S}_j) \Big) \\
&= \frac{1}{1+n} \Big( \text{rank}(\hat{S}_j) \text{ among } \{\hat{S}_i\}_{i \in \mathcal{D}} + \mathbf{1}(\hat{S}_{n+t} \le \hat{S}_j) \Big).
\end{aligned}
$$

  Computing the nonconformity scores for all samples in $\mathcal{D} \cup \{Z_{n+t}\}_{t=1}^{m}$ costs $\mathcal{O}(n + m)$, evaluating the ranks takes $\mathcal{O}(n \log n)$, and comparing $\hat{S}_{n+t}$ and $\hat{S}_j$ costs $\mathcal{O}(n \cdot m)$.
- Selecting the sensitive attribute costs $\mathcal{O}(n \cdot m + M \cdot K \cdot (n + m))$. Breaking in steps, for each test sample $Z_{n+t}$, the worst FPR for attribute $k$, as defined in (A20), can be rewritten as follows:

$$
\begin{aligned}
\delta_{k,t} &:= \max_{m \in [M_k]} \frac{\sum_{i \in \mathcal{D} \cup \{Z_{n+t}\}} E_{i,t} \cdot \mathbf{1}\{\phi(Z_i, \{k\}) = m\}}{\sum_{i \in \mathcal{D} \cup \{Z_{n+t}\}} \mathbf{1}\{\phi(Z_i, \{k\}) = m\}} \\
&= \max_{m \in [M_k]} \frac{\sum_{i \in \mathcal{D}} E_{i,t} \cdot \mathbf{1}\{\phi(Z_i, \{k\}) = m\} + E_{t,t} \cdot \mathbf{1}\{\phi(Z_i, \{k\}) = m\}}{\sum_{i \in \mathcal{D}} \mathbf{1}\{\phi(Z_i, \{k\}) = m\} + \mathbf{1}\{\phi(Z_i, \{k\}) = m\}} \\
&= \max_{m \in [M_k]} \frac{\sum_{i \in \mathcal{D}(k,m)} E_{i,t} + E_{t,t} \cdot \mathbf{1}\{\phi(Z_i, \{k\}) = m\}}{|\mathcal{D}(k,m)| + \mathbf{1}\{\phi(Z_i, \{k\}) = m\}},
\end{aligned}
$$

where $\mathcal{D}(k, m)$ represents a subset of the calibration data indexed by $\{i \in \mathcal{D} : \phi(Z_i, \{k\}) = m\}$. The trick we use is that the identification of subset $\mathcal{D}(k, m)$ does not depend on the test sample $t$ and can be *reused* to calculate the FPR for every test sample $\{Z_{n+t}\}_{t=1}^m$. In specific, identifying $\mathcal{D}(k, m), \forall k \in [K], m \in [M_k]$ takes $\mathcal{O}(n \cdot M \cdot K)$. For each $\mathcal{D}(k, m)$, computing $\sum_{i \in \mathcal{D}(k,m)} E_{i,t}, \forall t \in [m]$ takes $\mathcal{O}(|\mathcal{D}(k, m)| \cdot m)$, and computing $E_{t,t} \mathbf{1}\{\phi(Z_i, \{k\}) = m\}$ takes $\mathcal{O}(m)$. Therefore, repeating this process for all $\mathcal{D}(k, m), k \in [K], m \in [M_k]$ in total takes $\mathcal{O}(n \cdot m)$. Lastly, finding the maximum FPR across all attributes takes $\mathcal{O}(m \cdot M \cdot K)$.

Hence, the total cost of running Algorithm A7 for $m$ test samples is $\mathcal{O}(n \log n + nm + MK(n+m))$.

### A6.2 Multi-Class Classification

Given a pre-trained multi-class classifier, consider a calibration dataset $\mathcal{D}$ of size $n$. Let $L$ denote the total number of possible labels to predict, and let $K$ denote the number of sensitive attributes. For each attribute $k \in [K]$, let $M_k \in \mathbb{N}$ denote the count of its possible values. Denote $M = \max_k M_k$ as the maximum count across all attributes.

AFCP for multi-class classification outlined in Algorithm 2 has the following computational cost.

**Analysis for a single test point**

- The cost of constructing prediction sets and computing miscoverage indicators within the leave-one-out procedure is $\mathcal{O}(L \cdot n \cdot \log n)$.
- Then, selecting the sensitive attributes requires $\mathcal{O}(n \cdot K \cdot M + L \cdot (n + M \cdot K))$.
- Once the attribute is selected, the cost of applying conformal prediction conditional on the group identified by the selected attribute is $\mathcal{O}(1)$ because the subset of the group has been found in the last step.

Hence, the total cost of running Algorithm 2 for a single test point is $\mathcal{O}(L(n \log n + KM) + nKM)$.

**Analysis for $m$ test points**

- The cost of computing miscoverage indicators is $\mathcal{O}(L \cdot n \cdot (m + \log n))$. This can be derived by rewriting the conformal p-values for all $j \in \mathcal{D} \cup \{(X_{n+t}, y)\} \ \forall y \in [L]$, and $\forall t \in [m]$ as follows:

$$
\hat{u}_{j,t}^y = \frac{1}{1+n} \Big( \sum_{i \in \mathcal{D} \cup \{Z_{n+t}\}} \mathbf{1}(\hat{S}_i^y \leq \hat{S}_j^y) \Big)
$$

$$
= \frac{1}{1+n} \Big( \text{rank}(\hat{S}_j^y) \text{ among } \{\hat{S}_i^y\}_{i \in \mathcal{D}} + \mathbf{1}(\hat{S}_{n+t}^y \leq \hat{S}_j^y) \Big).
$$

For each placeholder label $y \in [L]$, computing the nonconformity scores for all samples in $\mathcal{D} \cup \{(X_{n+t}, y)\}_{t=1}^m$ costs $\mathcal{O}(n + m)$, evaluating the ranks takes $\mathcal{O}(n \log n)$, and comparing $\hat{S}_{n+t}^y$ with $\hat{S}_j^y$ costs $\mathcal{O}(n \cdot m)$. This process needs to be conducted for each $y \in [L]$, therefore the total cost of this step is $\mathcal{O}(L \cdot (n + m + n \log n + nm)) = \mathcal{O}(L \cdot n \cdot (\log n + m))$.

- Selecting the sensitive attribute costs $\mathcal{O}(n \cdot M \cdot K + L \cdot m(n + M \cdot K)$. Breaking in steps, for each test sample $(X_{n+t}, y), y \in [L]$, the worst miscoverage rate for attribute $k$, as defined in (5), can be rewritten as follows:

$$
\delta_{y,k,t} := \max_{m \in [M_k]} \frac{\sum_{i \in \mathcal{D} \cup \{(X_{n+t}, y)\}} E_{y,i,t} \cdot \mathbf{1}\{\phi(X_i, \{k\}) = m\}}{\sum_{i \in \mathcal{D} \cup \{(X_{n+t}, y)\}} \mathbf{1}\{\phi(X_i, \{k\}) = m\}}
$$

$$
= \max_{m \in [M_k]} \frac{\sum_{i \in \mathcal{D}} E_{y,i,t} \cdot \mathbf{1}\{\phi(X_i, \{k\}) = m\} + E_{y,t,t} \cdot \mathbf{1}\{\phi(X_i, \{k\}) = m\}}{\sum_{i \in \mathcal{D}} \mathbf{1}\{\phi(X_i, \{k\}) = m\} + \mathbf{1}\{\phi(X_i, \{k\}) = m\}}
$$

$$
= \max_{m \in [M_k]} \frac{\sum_{i \in \mathcal{D}(k,m)} E_{y,i,t} + E_{y,t,t} \cdot \mathbf{1}\{\phi(X_i, \{k\}) = m\}}{|\mathcal{D}(k, m)| + \mathbf{1}\{\phi(X_i, \{k\}) = m\}},
$$

where $\mathcal{D}(k, m)$ represents a subset of the calibration data indexed by $\{i \in \mathcal{D} : \phi(Z_i, \{k\}) = m\}$. The trick we use is that the identification of subset $\mathcal{D}(k, m)$ does not depend on the test sample $t$ and the placeholder label $y$, therefore can be *reused* to calculate the miscoverage

rate for every test sample $\{(X_{n+t}, y)\}_{t=1}^m$. In specific, identifying $\mathcal{D}(k, m), \forall k \in [K], m \in [M_k]$ takes $\mathcal{O}(n \cdot M \cdot K)$. For each $\mathcal{D}(k, m)$, computing $\sum_{i \in \mathcal{D}(k,m)} E_{y,i,t}, \forall t \in [m], \forall y \in [L]$ takes $\mathcal{O}(L \cdot |\mathcal{D}(k,m)| \cdot m)$, and computing $E_{y,t,t} \mathbf{1}\{\phi(Z_i, \{k\}) = m\}$ takes $\mathcal{O}(L \cdot m)$. Therefore, repeating this process for all $\mathcal{D}(k, m), k \in [K], m \in [M_k]$ and for all $y \in [L]$ in total takes $\mathcal{O}(L \cdot n \cdot m)$. Lastly, finding the maximum miscoverage across all attributes takes $\mathcal{O}(L \cdot m \cdot M \cdot K)$.

Hence, the total cost of running Algorithm 2 for $m$ test samples is $\mathcal{O}(n \log n + Lnm + MK(n + Lm))$.

## A7    Additional Results from Numerical Experiments

### A7.1    AFCP for Multiclass Classification

#### A7.1.1    Synthetic Data

Recall from Section 3.2 that Color is denoted as $X_0$ and the first one of the non-sensitive features as $X_1$. The distribution of $Y$ conditional on $X$ is modeled by a simple decision tree, where $X_0$ and $X_1$ are the only useful predictors for $Y$, formulated as the following:

$$\mathbb{P}[Y \mid X] = \begin{cases} \left(\frac{1}{3}, \frac{1}{3}, \frac{1}{3}, 0, 0, 0\right), & \text{if } X_0 = \text{Blue and } X_1 < 0.5, \\ \left(0, 0, 0, \frac{1}{3}, \frac{1}{3}, \frac{1}{3}\right), & \text{if } X_0 = \text{Blue and }, X_1 \geq 0.5, \\ (1, 0, 0, 0, 0, 0), & \text{if } X_0 = \text{Grey and }, 0 \leq X_1 \leq \frac{1}{6}, \\ (0, 1, 0, 0, 0, 0), & \text{if } X_0 = \text{Grey and }, \frac{1}{6} \leq X_1 \leq \frac{2}{6}, \\ \vdots & \vdots \\ (0, 0, 0, 0, 0, 1), & \text{if } X_0 = \text{Grey and }, \frac{5}{6} \leq X_1 \leq 1. \end{cases} \quad \text{(A27)}$$

The detailed numerical experiments results of AFCP with the adaptive equalized coverage (see Equation (3)) are presented and compared with other benchmark methods.

Table A1: Average coverage and average size of prediction sets for all test samples constructed by different methods as a function of the training and calibration size. All methods obtain coverage beyond 0.9, while our AFCP and AFCP1 methods, along with the Marginal method, produce the smallest, thus, the most informative, prediction sets. Red numbers indicate the small size of prediction sets. See corresponding plots in Figure 3.

| Sample size | AFCP | | AFCP1 | | Marginal | | Partial | | Exhaustive | |
|---|---|---|---|---|---|---|---|---|---|---|
| | Coverage | Size | Coverage | Size | Coverage | Size | Coverage | Size | Coverage | Size |
| 200 | 0.917 (0.003) | 3.016 (0.058) | 0.937 (0.003) | 3.206 (0.059) | 0.902 (0.003) | 2.861 (0.056) | 0.967 (0.002) | 3.624 (0.059) | 0.999 (0.000) | 5.965 (0.007) |
| 500 | 0.939 (0.002) | 1.860 (0.012) | 0.939 (0.002) | 1.861 (0.012) | 0.900 (0.002) | 1.685 (0.011) | 0.963 (0.001) | 2.094 (0.016) | 0.950 (0.002) | 3.219 (0.034) |
| 1000 | 0.945 (0.001) | 1.670 (0.009) | 0.945 (0.001) | 1.670 (0.009) | 0.900 (0.002) | 1.495 (0.009) | 0.959 (0.001) | 1.794 (0.010) | 0.933 (0.002) | 1.900 (0.009) |
| 2000 | 0.946 (0.001) | 1.548 (0.007) | 0.946 (0.001) | 1.548 (0.007) | 0.901 (0.002) | 1.391 (0.007) | 0.956 (0.001) | 1.618 (0.008) | 0.923 (0.001) | 1.719 (0.008) |

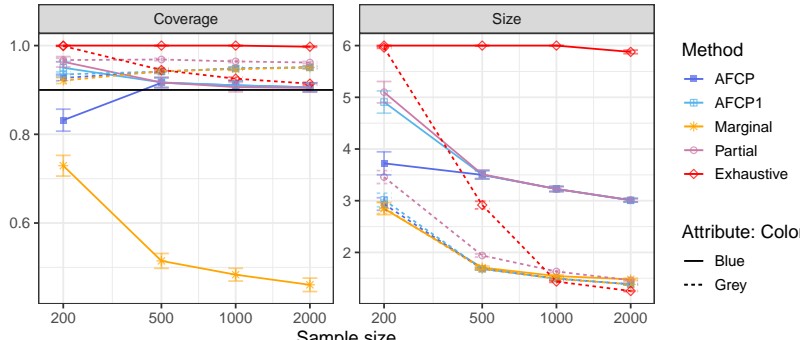

Figure A1: Coverage and size of prediction sets constructed with different methods for groups formed by Color. For the Blue group, the Marginal method (dashed orange lines) fails to detect and correct for its undercoverage, and the Exhaustive method produces prediction sets that are too conservative to be helpful. In contrast, our AFCP and AFCP1 methods correct the undercoverage and maintain small prediction sets. See Table A2 for numerical details and standard errors.

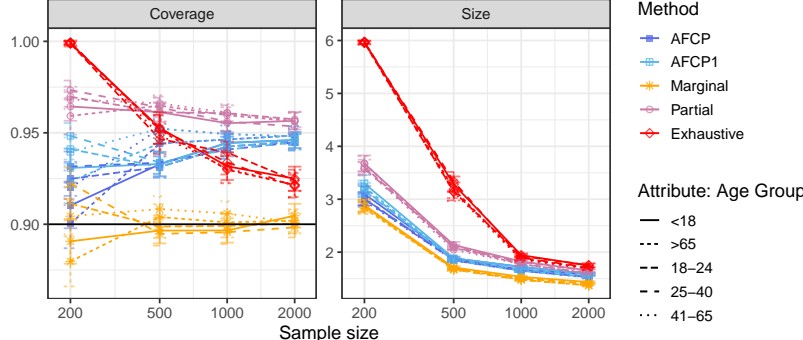

Figure A2: Coverage and size of prediction sets constructed with different methods for groups formed by Age group. By design, all groups have similar performance, and none of them are subject to unfairness/undercoverage. See Table A3 for numerical details and standard errors.

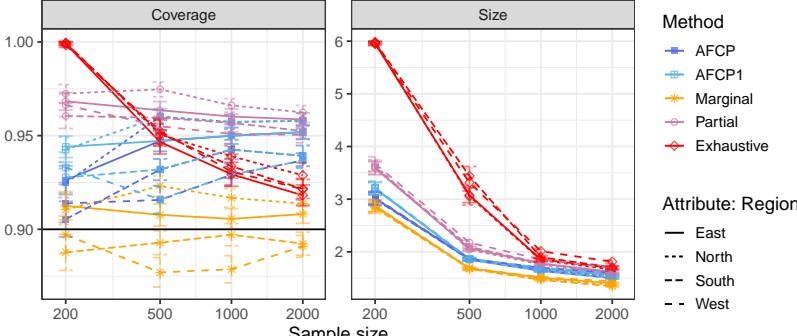

Figure A3: Coverage and size of prediction sets constructed with different methods for groups formed by Region. By design, all groups have similar performance, and none of them are subject to unfairness/undercoverage. See Table A4 for numerical details and standard errors.

Table A2: Coverage and size of prediction sets constructed with different methods for groups formed by Color. Green numbers indicate low coverage and red numbers indicate the small size of prediction sets. See corresponding plots in Figure A1.

| Group | Sample size | AFCP Coverage | AFCP Size | AFCP1 Coverage | AFCP1 Size | Marginal Coverage | Marginal Size | Partial Coverage | Partial Size | Exhaustive Coverage | Exhaustive Size |
|---|---|---|---|---|---|---|---|---|---|---|---|
| **Grey** | | | | | | | | | | | |
| Grey | 200 | 0.927 (0.003) | 2.936 (0.061) | 0.935 (0.003) | 3.015 (0.065) | 0.921 (0.003) | 2.862 (0.055) | 0.967 (0.002) | 3.457 (0.062) | 0.999 (0.000) | 5.961 (0.007) |
| Grey | 500 | 0.941 (0.002) | 1.682 (0.011) | 0.941 (0.002) | 1.682 (0.011) | 0.942 (0.002) | 1.683 (0.011) | 0.969 (0.001) | 1.941 (0.015) | 0.945 (0.002) | 2.917 (0.037) |
| Grey | 1000 | 0.949 (0.001) | 1.494 (0.009) | 0.949 (0.001) | 1.494 (0.009) | 0.947 (0.002) | 1.490 (0.009) | 0.964 (0.001) | 1.632 (0.010) | 0.925 (0.002) | 1.436 (0.008) |
| Grey | 2000 | 0.951 (0.001) | 1.384 (0.007) | 0.951 (0.001) | 1.384 (0.007) | 0.951 (0.001) | 1.381 (0.007) | 0.962 (0.001) | 1.462 (0.008) | 0.914 (0.002) | 1.254 (0.005) |
| **Blue** | | | | | | | | | | | |
| Blue | 200 | 0.832 (0.012) | 3.723 (0.111) | 0.950 (0.007) | 4.908 (0.108) | 0.729 (0.012) | 2.847 (0.059) | 0.963 (0.006) | 5.099 (0.103) | 1.000 (0.000) | 6.000 (0.000) |
| Blue | 500 | 0.916 (0.006) | 3.500 (0.040) | 0.917 (0.006) | 3.504 (0.040) | 0.515 (0.008) | 1.707 (0.014) | 0.917 (0.006) | 3.512 (0.041) | 1.000 (0.000) | 6.000 (0.000) |
| Blue | 1000 | 0.910 (0.005) | 3.228 (0.024) | 0.910 (0.005) | 3.228 (0.024) | 0.484 (0.007) | 1.542 (0.011) | 0.906 (0.005) | 3.225 (0.024) | 1.000 (0.000) | 6.000 (0.000) |
| Blue | 2000 | 0.906 (0.005) | 3.008 (0.017) | 0.906 (0.005) | 3.008 (0.017) | 0.461 (0.008) | 1.480 (0.010) | 0.905 (0.005) | 3.013 (0.017) | 0.998 (0.001) | 5.878 (0.017) |

Table A3: Coverage and size of prediction sets constructed with different methods for groups formed by Age Group. By design, all groups have similar performance, and none of them are subject to unfairness/undercoverage. See corresponding plots in Figure A2.

| Group | Sample size | AFCP Coverage | AFCP Size | AFCP1 Coverage | AFCP1 Size | Marginal Coverage | Marginal Size | Partial Coverage | Partial Size | Exhaustive Coverage | Exhaustive Size |
|---|---|---|---|---|---|---|---|---|---|---|---|
| **<18** | | | | | | | | | | | |
| <18 | 200 | 0.910 (0.006) | 3.098 (0.075) | 0.931 (0.005) | 3.298 (0.079) | 0.891 (0.006) | 2.887 (0.056) | 0.964 (0.004) | 3.674 (0.077) | 0.999 (0.001) | 5.967 (0.015) |
| <18 | 500 | 0.933 (0.003) | 1.884 (0.015) | 0.933 (0.003) | 1.885 (0.015) | 0.897 (0.003) | 1.710 (0.013) | 0.961 (0.003) | 2.136 (0.029) | 0.953 (0.004) | 3.283 (0.074) |
| <18 | 1000 | 0.944 (0.003) | 1.722 (0.010) | 0.944 (0.003) | 1.722 (0.010) | 0.897 (0.003) | 1.535 (0.010) | 0.955 (0.002) | 1.824 (0.014) | 0.932 (0.004) | 1.943 (0.017) |
| <18 | 2000 | 0.946 (0.002) | 1.586 (0.011) | 0.946 (0.002) | 1.586 (0.011) | 0.905 (0.003) | 1.434 (0.009) | 0.957 (0.002) | 1.653 (0.012) | 0.925 (0.003) | 1.747 (0.018) |
| **18-24** | | | | | | | | | | | |
| 18-24 | 200 | 0.925 (0.005) | 3.010 (0.063) | 0.941 (0.004) | 3.185 (0.063) | 0.911 (0.005) | 2.870 (0.058) | 0.970 (0.003) | 3.613 (0.068) | 0.999 (0.001) | 5.958 (0.016) |
| 18-24 | 500 | 0.931 (0.003) | 1.859 (0.014) | 0.931 (0.003) | 1.859 (0.014) | 0.899 (0.003) | 1.686 (0.012) | 0.961 (0.002) | 2.091 (0.024) | 0.947 (0.004) | 3.147 (0.084) |
| 18-24 | 1000 | 0.942 (0.003) | 1.673 (0.011) | 0.942 (0.003) | 1.673 (0.011) | 0.899 (0.003) | 1.504 (0.010) | 0.961 (0.002) | 1.815 (0.014) | 0.939 (0.003) | 1.925 (0.017) |
| 18-24 | 2000 | 0.945 (0.002) | 1.573 (0.009) | 0.945 (0.002) | 1.573 (0.009) | 0.900 (0.003) | 1.404 (0.008) | 0.957 (0.002) | 1.645 (0.013) | 0.924 (0.003) | 1.754 (0.018) |
| **25-40** | | | | | | | | | | | |
| 25-40 | 200 | 0.932 (0.004) | 2.995 (0.057) | 0.948 (0.004) | 3.159 (0.059) | 0.922 (0.004) | 2.868 (0.058) | 0.973 (0.003) | 3.577 (0.064) | 0.999 (0.001) | 5.965 (0.018) |
| 25-40 | 500 | 0.934 (0.003) | 1.845 (0.014) | 0.934 (0.003) | 1.845 (0.014) | 0.895 (0.003) | 1.672 (0.012) | 0.964 (0.003) | 2.116 (0.032) | 0.952 (0.004) | 3.336 (0.090) |
| 25-40 | 1000 | 0.941 (0.003) | 1.646 (0.011) | 0.941 (0.003) | 1.646 (0.011) | 0.896 (0.003) | 1.479 (0.010) | 0.956 (0.003) | 1.782 (0.018) | 0.934 (0.003) | 1.895 (0.020) |
| 25-40 | 2000 | 0.945 (0.002) | 1.530 (0.009) | 0.945 (0.002) | 1.530 (0.009) | 0.898 (0.003) | 1.372 (0.008) | 0.953 (0.002) | 1.600 (0.012) | 0.921 (0.003) | 1.710 (0.016) |
| **41-65** | | | | | | | | | | | |
| 41-65 | 200 | 0.920 (0.005) | 2.969 (0.057) | 0.940 (0.004) | 3.162 (0.060) | 0.904 (0.005) | 2.851 (0.056) | 0.968 (0.003) | 3.580 (0.061) | 0.999 (0.000) | 5.955 (0.018) |
| 41-65 | 500 | 0.952 (0.003) | 1.847 (0.013) | 0.952 (0.003) | 1.847 (0.013) | 0.908 (0.004) | 1.668 (0.011) | 0.966 (0.002) | 2.047 (0.026) | 0.949 (0.004) | 3.138 (0.086) |
| 41-65 | 1000 | 0.949 (0.002) | 1.654 (0.012) | 0.949 (0.002) | 1.654 (0.012) | 0.906 (0.003) | 1.479 (0.010) | 0.961 (0.002) | 1.772 (0.017) | 0.930 (0.004) | 1.858 (0.023) |
| 41-65 | 2000 | 0.948 (0.002) | 1.519 (0.009) | 0.948 (0.002) | 1.519 (0.009) | 0.901 (0.003) | 1.369 (0.008) | 0.956 (0.002) | 1.590 (0.010) | 0.921 (0.003) | 1.684 (0.015) |
| **>65** | | | | | | | | | | | |
| >65 | 200 | 0.900 (0.007) | 3.008 (0.059) | 0.923 (0.006) | 3.228 (0.064) | 0.880 (0.007) | 2.826 (0.054) | 0.959 (0.005) | 3.675 (0.072) | 0.999 (0.000) | 5.979 (0.010) |
| >65 | 500 | 0.944 (0.003) | 1.867 (0.014) | 0.944 (0.003) | 1.867 (0.014) | 0.904 (0.003) | 1.689 (0.012) | 0.965 (0.002) | 2.082 (0.020) | 0.952 (0.004) | 3.191 (0.084) |
| >65 | 1000 | 0.946 (0.002) | 1.653 (0.011) | 0.946 (0.002) | 1.653 (0.011) | 0.901 (0.003) | 1.479 (0.010) | 0.960 (0.002) | 1.774 (0.014) | 0.930 (0.004) | 1.880 (0.020) |
| >65 | 2000 | 0.949 (0.003) | 1.529 (0.010) | 0.949 (0.003) | 1.529 (0.010) | 0.902 (0.003) | 1.377 (0.009) | 0.956 (0.002) | 1.600 (0.012) | 0.921 (0.003) | 1.699 (0.016) |

Table A4: Coverage and size of prediction sets constructed with different methods for groups formed by Region. By design, all groups have similar performance, and none of them are subject to unfairness/undercoverage. See corresponding plots in Figure A3.

| Group | Sample size | AFCP Coverage | AFCP Size | AFCP1 Coverage | AFCP1 Size | Marginal Coverage | Marginal Size | Partial Coverage | Partial Size | Exhaustive Coverage | Exhaustive Size |
|---|---|---|---|---|---|---|---|---|---|---|---|
| **West** | | | | | | | | | | | |
| West | 200 | 0.914 (0.005) | 3.030 (0.059) | 0.933 (0.004) | 3.218 (0.060) | 0.897 (0.005) | 2.878 (0.057) | 0.966 (0.003) | 3.671 (0.068) | 0.998 (0.001) | 5.948 (0.017) |
| West | 500 | 0.916 (0.004) | 1.869 (0.015) | 0.916 (0.004) | 1.869 (0.015) | 0.877 (0.004) | 1.691 (0.012) | 0.955 (0.003) | 2.176 (0.022) | 0.952 (0.003) | 3.315 (0.080) |
| West | 1000 | 0.929 (0.003) | 1.699 (0.011) | 0.929 (0.003) | 1.699 (0.011) | 0.879 (0.004) | 1.516 (0.010) | 0.951 (0.003) | 1.854 (0.015) | 0.931 (0.004) | 2.012 (0.021) |
| West | 2000 | 0.937 (0.002) | 1.602 (0.009) | 0.937 (0.002) | 1.602 (0.009) | 0.891 (0.003) | 1.442 (0.009) | 0.950 (0.002) | 1.693 (0.011) | 0.921 (0.003) | 1.811 (0.015) |
| **East** | | | | | | | | | | | |
| East | 200 | 0.926 (0.004) | 3.012 (0.059) | 0.944 (0.003) | 3.194 (0.059) | 0.913 (0.004) | 2.874 (0.057) | 0.968 (0.002) | 3.586 (0.062) | 0.999 (0.001) | 5.958 (0.016) |
| East | 500 | 0.947 (0.003) | 1.862 (0.013) | 0.947 (0.003) | 1.863 (0.013) | 0.908 (0.003) | 1.686 (0.012) | 0.964 (0.002) | 2.065 (0.020) | 0.947 (0.003) | 3.075 (0.071) |
| East | 1000 | 0.950 (0.002) | 1.677 (0.010) | 0.950 (0.002) | 1.677 (0.010) | 0.906 (0.003) | 1.508 (0.010) | 0.960 (0.002) | 1.778 (0.014) | 0.929 (0.003) | 1.856 (0.019) |
| East | 2000 | 0.952 (0.002) | 1.564 (0.009) | 0.952 (0.002) | 1.564 (0.009) | 0.908 (0.002) | 1.406 (0.008) | 0.959 (0.002) | 1.621 (0.010) | 0.918 (0.003) | 1.694 (0.015) |
| **North** | | | | | | | | | | | |
| North | 200 | 0.925 (0.004) | 3.022 (0.059) | 0.943 (0.003) | 3.204 (0.060) | 0.911 (0.004) | 2.855 (0.055) | 0.972 (0.002) | 3.630 (0.058) | 0.999 (0.001) | 5.979 (0.010) |
| North | 500 | 0.960 (0.002) | 1.859 (0.012) | 0.960 (0.002) | 1.859 (0.012) | 0.923 (0.003) | 1.687 (0.010) | 0.975 (0.002) | 2.040 (0.016) | 0.951 (0.003) | 3.054 (0.083) |
| North | 1000 | 0.957 (0.002) | 1.665 (0.011) | 0.957 (0.002) | 1.665 (0.011) | 0.917 (0.002) | 1.489 (0.010) | 0.966 (0.002) | 1.760 (0.012) | 0.939 (0.003) | 1.841 (0.018) |
| North | 2000 | 0.958 (0.002) | 1.523 (0.009) | 0.958 (0.002) | 1.523 (0.009) | 0.914 (0.003) | 1.373 (0.008) | 0.962 (0.002) | 1.572 (0.010) | 0.929 (0.003) | 1.664 (0.015) |
| **South** | | | | | | | | | | | |
| South | 200 | 0.905 (0.005) | 2.998 (0.060) | 0.928 (0.004) | 3.209 (0.064) | 0.888 (0.005) | 2.836 (0.054) | 0.961 (0.003) | 3.604 (0.065) | 1.000 (0.000) | 5.976 (0.011) |
| South | 500 | 0.932 (0.003) | 1.851 (0.012) | 0.932 (0.003) | 1.852 (0.012) | 0.893 (0.003) | 1.676 (0.012) | 0.960 (0.002) | 2.097 (0.022) | 0.951 (0.003) | 3.440 (0.090) |
| South | 1000 | 0.943 (0.003) | 1.639 (0.010) | 0.943 (0.003) | 1.639 (0.010) | 0.897 (0.003) | 1.468 (0.010) | 0.957 (0.002) | 1.784 (0.015) | 0.934 (0.003) | 1.898 (0.016) |
| South | 2000 | 0.939 (0.003) | 1.501 (0.008) | 0.939 (0.003) | 1.501 (0.008) | 0.892 (0.003) | 1.345 (0.007) | 0.952 (0.002) | 1.584 (0.011) | 0.922 (0.003) | 1.704 (0.015) |

Next, we provide numerical experimental results comparing AFCP with the **label-conditional** adaptive equalized coverage (see Equation (A14)) with other benchmark methods.

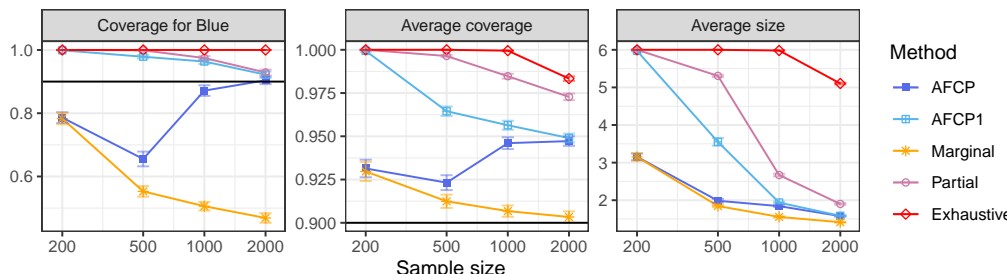

Figure A4: Performance on prediction sets constructed by different methods on synthetic medical diagnosis data as a function of the total number of training and calibration points. Our method (AFCP) leads to more informative sets with lower average width and higher conditional coverage on the Blue group. The error bars indicate 2 standard errors.

Table A5: Average coverage and average size of prediction sets for all test samples constructed by different methods as a function of the training and calibration size. All methods obtain coverage beyond 0.9, while our AFCP and AFCP1 methods, along with the Marginal method, produce the smallest, thus, the most informative, prediction sets. Red numbers indicate the small size of prediction sets. See corresponding plots in Figure A4.

| Sample size | AFCP | | AFCP1 | | Marginal | | Partial | | Exhaustive | |
|---|---|---|---|---|---|---|---|---|---|---|
| | Coverage | Size | Coverage | Size | Coverage | Size | Coverage | Size | Coverage | Size |
| 200 | 0.931 (0.003) | 3.153 (0.046) | 0.999 (0.000) | 5.972 (0.008) | 0.930 (0.003) | 3.151 (0.047) | 1.000 (0.000) | 6.000 (0.000) | 1.000 (0.000) | 6.000 (0.000) |
| 500 | 0.923 (0.002) | 1.986 (0.025) | 0.965 (0.001) | 3.552 (0.051) | 0.912 (0.002) | 1.846 (0.016) | 0.996 (0.000) | 5.307 (0.015) | 1.000 (0.000) | 6.000 (0.000) |
| 1000 | 0.946 (0.002) | 1.843 (0.012) | 0.956 (0.001) | 1.938 (0.012) | 0.907 (0.002) | 1.557 (0.010) | 0.985 (0.001) | 2.669 (0.017) | 1.000 (0.000) | 5.982 (0.002) |
| 2000 | 0.947 (0.001) | 1.578 (0.008) | 0.949 (0.001) | 1.585 (0.008) | 0.903 (0.002) | 1.415 (0.008) | 0.973 (0.001) | 1.900 (0.011) | 0.983 (0.001) | 5.104 (0.011) |

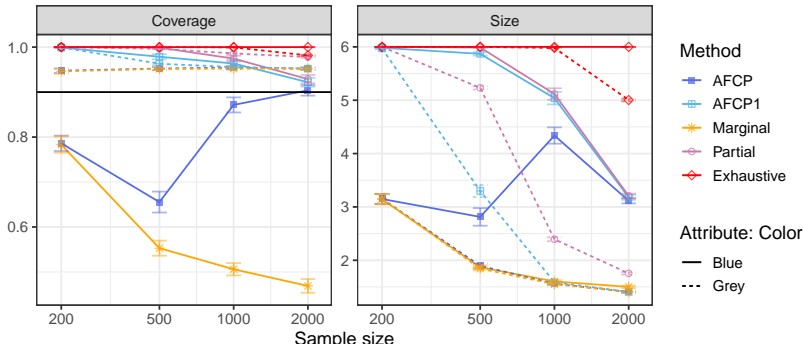

Figure A5: Coverage and size of prediction sets constructed with different methods for groups formed by Color. For the Blue group, the Marginal method (dashed orange lines) fails to detect and correct for its undercoverage, and the Exhaustive method produces prediction sets that are too conservative to be helpful. In contrast, our AFCP and AFCP1 methods correct the undercoverage and maintain small prediction sets. See Table A6 for numerical details and standard errors.

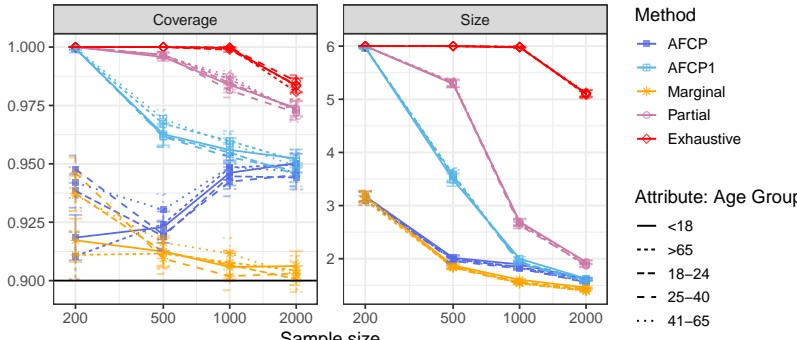

Figure A6: Coverage and size of prediction sets constructed with different methods for groups formed by Age group. By design, all groups have similar performance, and none of them are subject to unfairness/undercoverage. See Table A7 for numerical details and standard errors.

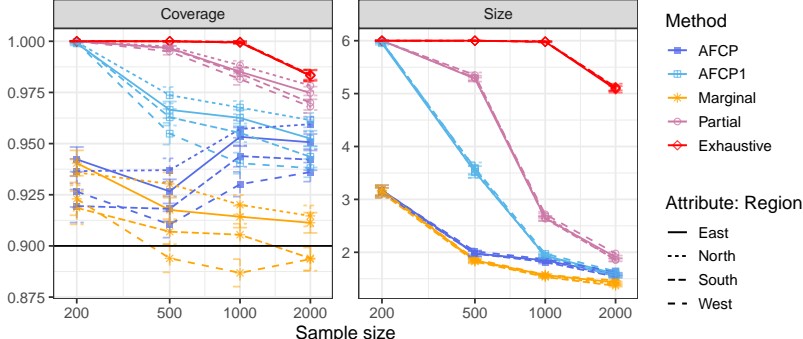

Figure A7: Coverage and size of prediction sets constructed with different methods for groups formed by Region. By design, all groups have similar performance, and none of them are subject to unfairness/undercoverage. See Table A8 for numerical details and standard errors.

Table A6: Coverage and size of prediction sets constructed with different methods for groups formed by Color. Green numbers indicate low coverage and red numbers indicate the small size of prediction sets. See corresponding plots in Figure A5.

| Group | Sample size | AFCP Coverage | AFCP Size | AFCP1 Coverage | AFCP1 Size | Marginal Coverage | Marginal Size | Partial Coverage | Partial Size | Exhaustive Coverage | Exhaustive Size |
|---|---|---|---|---|---|---|---|---|---|---|---|
| **Grey** | | | | | | | | | | | |
| Grey | 200 | 0.948 (0.003) | 3.153 (0.046) | 0.999 (0.000) | 5.971 (0.008) | 0.946 (0.003) | 3.151 (0.047) | 1.000 (0.000) | 6.000 (0.000) | 1.000 (0.000) | 6.000 (0.000) |
| Grey | 500 | 0.952 (0.002) | 1.896 (0.022) | 0.963 (0.002) | 3.299 (0.057) | 0.952 (0.002) | 1.843 (0.016) | 0.996 (0.000) | 5.232 (0.016) | 1.000 (0.000) | 6.000 (0.000) |
| Grey | 1000 | 0.955 (0.001) | 1.561 (0.011) | 0.956 (0.001) | 1.588 (0.013) | 0.952 (0.002) | 1.551 (0.011) | 0.986 (0.001) | 2.393 (0.019) | 0.999 (0.000) | 5.980 (0.002) |
| Grey | 2000 | 0.952 (0.001) | 1.406 (0.008) | 0.952 (0.001) | 1.407 (0.007) | 0.952 (0.001) | 1.405 (0.008) | 0.978 (0.001) | 1.755 (0.011) | 0.981 (0.001) | 5.004 (0.012) |
| **Blue** | | | | | | | | | | | |
| Blue | 200 | 0.786 (0.009) | 3.149 (0.047) | 0.999 (0.000) | 5.983 (0.005) | 0.783 (0.009) | 3.145 (0.049) | 1.000 (0.000) | 6.000 (0.000) | 1.000 (0.000) | 6.000 (0.000) |
| Blue | 500 | 0.655 (0.012) | 2.813 (0.083) | 0.979 (0.003) | 5.869 (0.016) | 0.553 (0.008) | 1.869 (0.018) | 0.999 (0.001) | 5.989 (0.005) | 1.000 (0.000) | 6.000 (0.000) |
| Blue | 1000 | 0.872 (0.008) | 4.339 (0.077) | 0.964 (0.004) | 5.038 (0.058) | 0.506 (0.007) | 1.603 (0.013) | 0.974 (0.003) | 5.117 (0.054) | 1.000 (0.000) | 6.000 (0.000) |
| Blue | 2000 | 0.904 (0.006) | 3.114 (0.024) | 0.922 (0.005) | 3.180 (0.023) | 0.469 (0.008) | 1.501 (0.012) | 0.929 (0.005) | 3.206 (0.022) | 1.000 (0.000) | 6.000 (0.000) |

Table A7: Coverage and size of prediction sets constructed with different methods for groups formed by Age Group. By design, all groups have similar performance, and none of them are subject to unfairness/undercoverage. See corresponding plots in Figure A6.

| Group | Sample size | AFCP Coverage | AFCP Size | AFCP1 Coverage | AFCP1 Size | Marginal Coverage | Marginal Size | Partial Coverage | Partial Size | Exhaustive Coverage | Exhaustive Size |
|---|---|---|---|---|---|---|---|---|---|---|---|
| **<18** | | | | | | | | | | | |
| <18 | 200 | 0.918 (0.005) | 3.172 (0.046) | 0.999 (0.000) | 5.972 (0.008) | 0.917 (0.005) | 3.168 (0.048) | 1.000 (0.000) | 6.000 (0.000) | 1.000 (0.000) | 6.000 (0.000) |
| <18 | 500 | 0.923 (0.003) | 2.017 (0.028) | 0.963 (0.002) | 3.483 (0.059) | 0.912 (0.003) | 1.874 (0.019) | 0.996 (0.001) | 5.289 (0.034) | 1.000 (0.000) | 5.999 (0.001) |
| <18 | 1000 | 0.946 (0.003) | 1.893 (0.016) | 0.956 (0.003) | 1.999 (0.019) | 0.906 (0.003) | 1.604 (0.012) | 0.984 (0.002) | 2.696 (0.030) | 1.000 (0.000) | 5.980 (0.006) |
| <18 | 2000 | 0.950 (0.002) | 1.616 (0.012) | 0.952 (0.002) | 1.624 (0.012) | 0.906 (0.003) | 1.458 (0.009) | 0.974 (0.002) | 1.940 (0.016) | 0.983 (0.002) | 5.110 (0.031) |
| **18-24** | | | | | | | | | | | |
| 18-24 | 200 | 0.938 (0.004) | 3.178 (0.050) | 1.000 (0.000) | 5.971 (0.008) | 0.937 (0.004) | 3.180 (0.050) | 1.000 (0.000) | 6.000 (0.000) | 1.000 (0.000) | 6.000 (0.000) |
| 18-24 | 500 | 0.919 (0.003) | 1.994 (0.029) | 0.962 (0.002) | 3.555 (0.059) | 0.912 (0.003) | 1.846 (0.017) | 0.997 (0.001) | 5.303 (0.036) | 1.000 (0.000) | 6.000 (0.000) |
| 18-24 | 1000 | 0.945 (0.003) | 1.843 (0.015) | 0.955 (0.002) | 1.934 (0.016) | 0.907 (0.003) | 1.561 (0.012) | 0.985 (0.001) | 2.685 (0.031) | 0.999 (0.000) | 5.975 (0.006) |
| 18-24 | 2000 | 0.944 (0.003) | 1.605 (0.010) | 0.945 (0.002) | 1.611 (0.010) | 0.901 (0.003) | 1.429 (0.009) | 0.973 (0.002) | 1.930 (0.017) | 0.984 (0.001) | 5.104 (0.035) |
| **25-40** | | | | | | | | | | | |
| 25-40 | 200 | 0.948 (0.003) | 3.162 (0.050) | 0.999 (0.000) | 5.971 (0.008) | 0.946 (0.003) | 3.164 (0.051) | 1.000 (0.000) | 6.000 (0.000) | 1.000 (0.000) | 6.000 (0.000) |
| 25-40 | 500 | 0.920 (0.003) | 1.957 (0.026) | 0.962 (0.002) | 3.554 (0.059) | 0.909 (0.003) | 1.827 (0.017) | 0.997 (0.001) | 5.299 (0.037) | 1.000 (0.000) | 6.000 (0.000) |
| 25-40 | 1000 | 0.942 (0.003) | 1.814 (0.015) | 0.953 (0.003) | 1.908 (0.017) | 0.902 (0.003) | 1.537 (0.012) | 0.981 (0.002) | 2.628 (0.030) | 1.000 (0.000) | 5.985 (0.005) |
| 25-40 | 2000 | 0.945 (0.002) | 1.558 (0.010) | 0.947 (0.002) | 1.566 (0.009) | 0.903 (0.002) | 1.398 (0.009) | 0.972 (0.002) | 1.891 (0.017) | 0.985 (0.001) | 5.099 (0.027) |
| **41-65** | | | | | | | | | | | |
| 41-65 | 200 | 0.942 (0.003) | 3.160 (0.048) | 1.000 (0.000) | 5.974 (0.007) | 0.937 (0.003) | 3.148 (0.049) | 1.000 (0.000) | 6.000 (0.000) | 1.000 (0.000) | 6.000 (0.000) |
| 41-65 | 500 | 0.930 (0.003) | 1.969 (0.026) | 0.969 (0.002) | 3.575 (0.059) | 0.916 (0.003) | 1.836 (0.016) | 0.996 (0.001) | 5.328 (0.038) | 1.000 (0.000) | 6.000 (0.000) |
| 41-65 | 1000 | 0.949 (0.003) | 1.826 (0.016) | 0.959 (0.002) | 1.917 (0.016) | 0.912 (0.003) | 1.537 (0.011) | 0.988 (0.001) | 2.676 (0.031) | 1.000 (0.000) | 5.981 (0.005) |
| 41-65 | 2000 | 0.947 (0.002) | 1.552 (0.009) | 0.949 (0.002) | 1.558 (0.009) | 0.902 (0.003) | 1.391 (0.009) | 0.972 (0.002) | 1.873 (0.015) | 0.983 (0.002) | 5.126 (0.032) |
| **>65** | | | | | | | | | | | |
| >65 | 200 | 0.910 (0.005) | 3.092 (0.043) | 0.999 (0.001) | 5.972 (0.008) | 0.911 (0.005) | 3.093 (0.043) | 1.000 (0.000) | 6.000 (0.000) | 1.000 (0.000) | 6.000 (0.000) |
| >65 | 500 | 0.924 (0.003) | 1.992 (0.028) | 0.967 (0.002) | 3.590 (0.059) | 0.912 (0.003) | 1.846 (0.017) | 0.997 (0.001) | 5.316 (0.034) | 1.000 (0.000) | 6.000 (0.000) |
| >65 | 1000 | 0.948 (0.003) | 1.839 (0.016) | 0.960 (0.002) | 1.930 (0.016) | 0.908 (0.003) | 1.544 (0.011) | 0.986 (0.001) | 2.662 (0.028) | 1.000 (0.000) | 5.988 (0.004) |
| >65 | 2000 | 0.950 (0.002) | 1.562 (0.011) | 0.951 (0.002) | 1.568 (0.011) | 0.904 (0.003) | 1.400 (0.010) | 0.973 (0.002) | 1.868 (0.016) | 0.981 (0.002) | 5.082 (0.032) |

Table A8: Coverage and size of prediction sets constructed with different methods for groups formed by Region. By design, all groups have similar performance, and none of them are subject to unfairness/undercoverage. See corresponding plots in Figure A7.

| Group | Sample size | AFCP Coverage | AFCP Size | AFCP1 Coverage | AFCP1 Size | Marginal Coverage | Marginal Size | Partial Coverage | Partial Size | Exhaustive Coverage | Exhaustive Size |
|---|---|---|---|---|---|---|---|---|---|---|---|
| **West** | | | | | | | | | | | |
| West | 200 | 0.927 (0.004) | 3.165 (0.047) | 0.999 (0.000) | 5.979 (0.009) | 0.923 (0.004) | 3.164 (0.048) | 1.000 (0.000) | 6.000 (0.000) | 1.000 (0.000) | 6.000 (0.000) |
| West | 500 | 0.910 (0.003) | 2.018 (0.028) | 0.955 (0.003) | 3.594 (0.055) | 0.894 (0.003) | 1.872 (0.016) | 0.995 (0.001) | 5.303 (0.023) | 1.000 (0.000) | 5.999 (0.001) |
| West | 1000 | 0.930 (0.003) | 1.869 (0.015) | 0.940 (0.003) | 1.972 (0.018) | 0.887 (0.003) | 1.575 (0.012) | 0.981 (0.001) | 2.718 (0.024) | 0.999 (0.000) | 5.976 (0.005) |
| West | 2000 | 0.936 (0.002) | 1.632 (0.009) | 0.938 (0.002) | 1.640 (0.009) | 0.894 (0.003) | 1.465 (0.009) | 0.968 (0.002) | 1.978 (0.013) | 0.983 (0.001) | 5.102 (0.025) |
| **East** | | | | | | | | | | | |
| East | 200 | 0.942 (0.003) | 3.174 (0.048) | 0.999 (0.000) | 5.952 (0.014) | 0.940 (0.003) | 3.172 (0.049) | 1.000 (0.000) | 6.000 (0.000) | 1.000 (0.000) | 6.000 (0.000) |
| East | 500 | 0.927 (0.003) | 1.980 (0.025) | 0.966 (0.002) | 3.520 (0.054) | 0.918 (0.003) | 1.848 (0.016) | 0.997 (0.000) | 5.283 (0.028) | 1.000 (0.000) | 6.000 (0.000) |
| East | 1000 | 0.953 (0.002) | 1.846 (0.016) | 0.963 (0.002) | 1.934 (0.015) | 0.914 (0.003) | 1.569 (0.012) | 0.985 (0.001) | 2.643 (0.022) | 0.999 (0.000) | 5.980 (0.005) |
| East | 2000 | 0.951 (0.002) | 1.588 (0.010) | 0.952 (0.002) | 1.595 (0.010) | 0.911 (0.002) | 1.430 (0.009) | 0.975 (0.002) | 1.909 (0.015) | 0.983 (0.001) | 5.081 (0.031) |
| **North** | | | | | | | | | | | |
| North | 200 | 0.937 (0.003) | 3.146 (0.046) | 1.000 (0.000) | 5.979 (0.010) | 0.936 (0.003) | 3.144 (0.046) | 1.000 (0.000) | 6.000 (0.000) | 1.000 (0.000) | 6.000 (0.000) |
| North | 500 | 0.937 (0.003) | 1.972 (0.026) | 0.974 (0.002) | 3.519 (0.057) | 0.930 (0.003) | 1.840 (0.016) | 0.997 (0.001) | 5.292 (0.027) | 1.000 (0.000) | 6.000 (0.000) |
| North | 1000 | 0.957 (0.002) | 1.844 (0.015) | 0.968 (0.002) | 1.937 (0.015) | 0.920 (0.002) | 1.548 (0.011) | 0.988 (0.001) | 2.686 (0.023) | 1.000 (0.000) | 5.986 (0.004) |
| North | 2000 | 0.959 (0.002) | 1.554 (0.010) | 0.961 (0.002) | 1.560 (0.010) | 0.915 (0.003) | 1.397 (0.009) | 0.978 (0.001) | 1.854 (0.014) | 0.983 (0.001) | 5.107 (0.031) |
| **South** | | | | | | | | | | | |
| South | 200 | 0.919 (0.004) | 3.125 (0.047) | 1.000 (0.000) | 5.982 (0.009) | 0.919 (0.004) | 3.120 (0.047) | 1.000 (0.000) | 6.000 (0.000) | 1.000 (0.000) | 6.000 (0.000) |
| South | 500 | 0.918 (0.003) | 1.974 (0.030) | 0.963 (0.002) | 3.580 (0.059) | 0.907 (0.003) | 1.826 (0.018) | 0.996 (0.001) | 5.351 (0.024) | 1.000 (0.000) | 6.000 (0.000) |
| South | 1000 | 0.944 (0.003) | 1.815 (0.014) | 0.955 (0.002) | 1.911 (0.016) | 0.905 (0.003) | 1.533 (0.011) | 0.984 (0.001) | 2.634 (0.022) | 1.000 (0.000) | 5.985 (0.004) |
| South | 2000 | 0.942 (0.003) | 1.537 (0.009) | 0.944 (0.003) | 1.544 (0.009) | 0.894 (0.003) | 1.366 (0.008) | 0.970 (0.002) | 1.860 (0.015) | 0.984 (0.001) | 5.124 (0.031) |

### A7.1.2 Nursery Data

Table A9: Average coverage and average size of prediction sets for all test samples constructed by different methods as a function of the training and calibration size. All methods obtain coverage beyond 0.9, while our AFCP and AFCP1 methods, along with the Marginal method, produce the smallest, thus, the most informative, prediction sets. Red numbers indicate the small size of prediction sets. See corresponding plots in Figure 5.

| Sample size | AFCP Coverage | AFCP Size | AFCP1 Coverage | AFCP1 Size | Marginal Coverage | Marginal Size | Partial Coverage | Partial Size | Exhaustive Coverage | Exhaustive Size |
|---|---|---|---|---|---|---|---|---|---|---|
| 200 | 0.923 (0.004) | 1.847 (0.029) | 0.934 (0.003) | 1.918 (0.030) | 0.896 (0.003) | 1.726 (0.027) | 0.977 (0.002) | 2.559 (0.039) | 0.999 (0.001) | 3.980 (0.014) |
| 500 | 0.930 (0.002) | 1.623 (0.014) | 0.931 (0.002) | 1.625 (0.014) | 0.900 (0.002) | 1.504 (0.015) | 0.961 (0.002) | 1.909 (0.022) | 1.000 (0.000) | 4.000 (0.000) |
| 1000 | 0.928 (0.002) | 1.486 (0.009) | 0.928 (0.002) | 1.486 (0.009) | 0.900 (0.002) | 1.375 (0.009) | 0.952 (0.002) | 1.655 (0.014) | 1.000 (0.000) | 3.997 (0.001) |
| 2000 | 0.928 (0.001) | 1.359 (0.006) | 0.928 (0.001) | 1.359 (0.006) | 0.902 (0.002) | 1.259 (0.005) | 0.945 (0.001) | 1.450 (0.007) | 0.992 (0.000) | 3.738 (0.006) |
| 5000 | 0.916 (0.001) | 1.156 (0.005) | 0.916 (0.001) | 1.156 (0.005) | 0.901 (0.002) | 1.085 (0.003) | 0.928 (0.002) | 1.180 (0.005) | 0.939 (0.001) | 1.342 (0.007) |

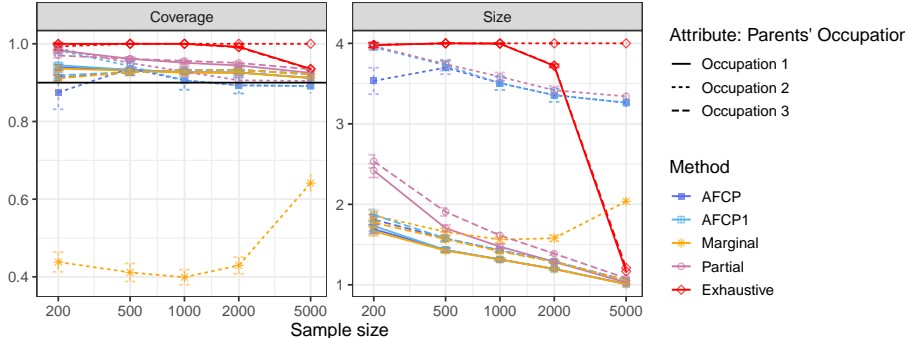

Figure A8: Coverage and size of prediction sets constructed with different methods for groups formed by Parents' Occupation. For the Occupation 1 group, the Marginal method (dashed orange lines) fails to detect and correct for its undercoverage, and the Exhaustive method produces prediction sets that are too conservative to be helpful. In contrast, our AFCP and AFCP1 methods correct the undercoverage and maintain small prediction sets. See Table A10 for numerical details and standard errors.

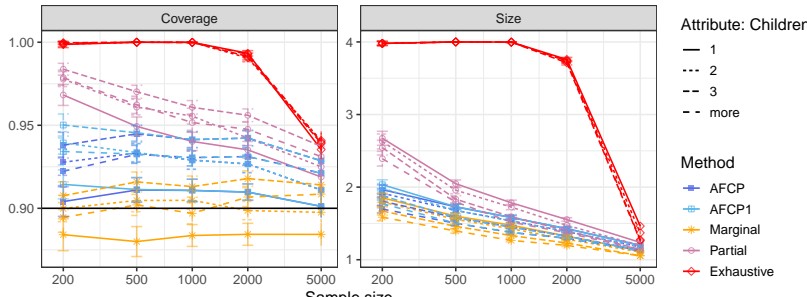

Figure A9: Coverage and size of prediction sets constructed with different methods for groups formed by Children. All groups have similar performance, and none of them are subject to unfairness/undercoverage. See Table A11 for numerical details and standard errors.

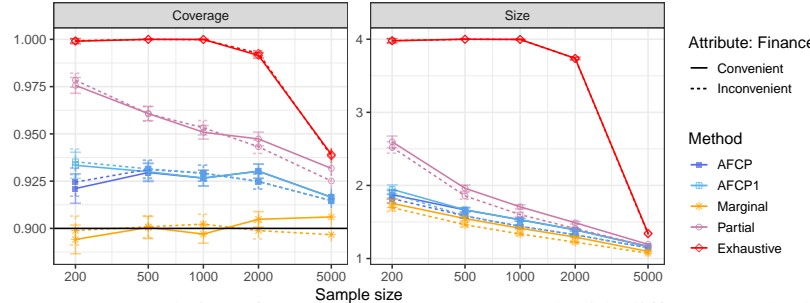

Figure A10: Coverage and size of prediction sets constructed with different methods for groups formed by Finance. All groups have similar performance, and none of them are subject to unfairness/undercoverage. See Table A12 for numerical details and standard errors.

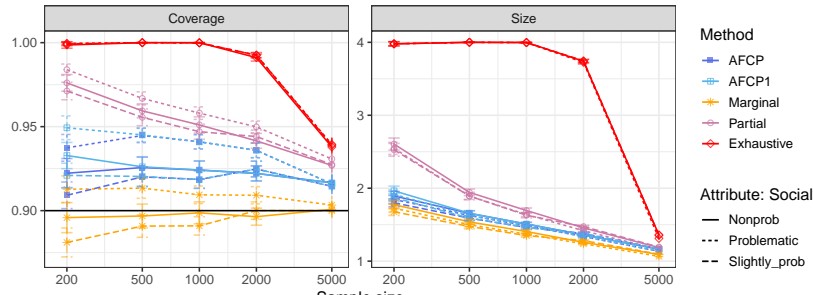

Figure A11: Coverage and size of prediction sets constructed with different methods for groups formed by Social. All groups have similar performance, and none of them are subject to unfairness/undercoverage. See Table A13 for numerical details and standard errors.

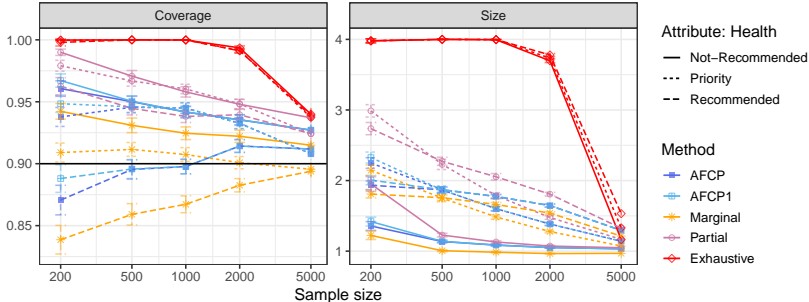

Figure A12: Coverage and size of prediction sets constructed with different methods for groups formed by Health. All groups have similar performance, and none of them are subject to unfairness/undercoverage. See Table A14 for numerical details and standard errors.

Table A10: Coverage and size of prediction sets constructed with different methods for groups formed by Parents' Occupation. Green numbers indicate low coverage and red numbers indicate the small size of prediction sets. See corresponding plots in Figure A8.

| Group | Sample size | AFCP Coverage | AFCP Size | AFCP1 Coverage | AFCP1 Size | Marginal Coverage | Marginal Size | Partial Coverage | Partial Size | Exhaustive Coverage | Exhaustive Size |
|---|---|---|---|---|---|---|---|---|---|---|---|
| **Occupation 0** | | | | | | | | | | | |
| Occupation 0 | 200 | 0.940 (0.003) | 1.692 (0.028) | 0.945 (0.003) | 1.731 (0.031) | 0.936 (0.004) | 1.665 (0.029) | 0.983 (0.002) | 2.420 (0.044) | 1.000 (0.000) | 3.980 (0.014) |
| Occupation 0 | 500 | 0.933 (0.002) | 1.433 (0.012) | 0.933 (0.002) | 1.433 (0.012) | 0.931 (0.003) | 1.424 (0.013) | 0.961 (0.002) | 1.703 (0.021) | 1.000 (0.000) | 4.000 (0.000) |
| Occupation 0 | 1000 | 0.927 (0.002) | 1.315 (0.009) | 0.927 (0.002) | 1.315 (0.009) | 0.925 (0.002) | 1.314 (0.009) | 0.951 (0.002) | 1.475 (0.015) | 1.000 (0.000) | 3.996 (0.001) |
| Occupation 0 | 2000 | 0.926 (0.002) | 1.197 (0.007) | 0.926 (0.002) | 1.197 (0.007) | 0.926 (0.002) | 1.196 (0.006) | 0.944 (0.002) | 1.286 (0.008) | 0.992 (0.001) | 3.720 (0.010) |
| Occupation 0 | 5000 | 0.912 (0.002) | 1.010 (0.004) | 0.912 (0.002) | 1.010 (0.004) | 0.912 (0.002) | 1.007 (0.003) | 0.925 (0.002) | 1.028 (0.004) | 0.936 (0.002) | 1.167 (0.009) |
| **Occupation 1** | | | | | | | | | | | |
| Occupation 1 | 200 | 0.875 (0.022) | 3.533 (0.082) | 0.984 (0.006) | 3.952 (0.017) | 0.438 (0.013) | 1.862 (0.031) | 0.988 (0.006) | 3.965 (0.016) | 0.993 (0.005) | 3.980 (0.014) |
| Occupation 1 | 500 | 0.937 (0.009) | 3.699 (0.041) | 0.943 (0.008) | 3.727 (0.034) | 0.411 (0.012) | 1.657 (0.024) | 0.950 (0.006) | 3.742 (0.027) | 1.000 (0.000) | 4.000 (0.000) |
| Occupation 1 | 1000 | 0.906 (0.012) | 3.505 (0.042) | 0.906 (0.012) | 3.505 (0.042) | 0.399 (0.010) | 1.561 (0.020) | 0.928 (0.007) | 3.585 (0.026) | 1.000 (0.000) | 4.000 (0.000) |
| Occupation 1 | 2000 | 0.893 (0.010) | 3.356 (0.041) | 0.893 (0.010) | 3.356 (0.041) | 0.429 (0.011) | 1.579 (0.020) | 0.906 (0.007) | 3.416 (0.027) | 1.000 (0.000) | 4.000 (0.000) |
| Occupation 1 | 5000 | 0.891 (0.010) | 3.263 (0.038) | 0.891 (0.010) | 3.263 (0.038) | 0.641 (0.012) | 2.037 (0.021) | 0.904 (0.006) | 3.341 (0.025) | 1.000 (0.000) | 4.000 (0.000) |
| **Occupation 2** | | | | | | | | | | | |
| Occupation 2 | 200 | 0.913 (0.004) | 1.810 (0.033) | 0.919 (0.004) | 1.868 (0.035) | 0.911 (0.004) | 1.772 (0.029) | 0.970 (0.003) | 2.534 (0.041) | 0.999 (0.001) | 3.980 (0.014) |
| Occupation 2 | 500 | 0.927 (0.003) | 1.577 (0.018) | 0.927 (0.003) | 1.577 (0.018) | 0.926 (0.003) | 1.566 (0.018) | 0.961 (0.002) | 1.906 (0.026) | 1.000 (0.000) | 4.000 (0.000) |
| Occupation 2 | 1000 | 0.932 (0.002) | 1.428 (0.011) | 0.932 (0.002) | 1.428 (0.011) | 0.931 (0.002) | 1.415 (0.011) | 0.956 (0.002) | 1.614 (0.016) | 1.000 (0.000) | 3.998 (0.001) |
| Occupation 2 | 2000 | 0.933 (0.002) | 1.289 (0.007) | 0.933 (0.002) | 1.289 (0.007) | 0.933 (0.002) | 1.285 (0.007) | 0.951 (0.002) | 1.387 (0.009) | 0.991 (0.001) | 3.726 (0.010) |
| Occupation 2 | 5000 | 0.922 (0.002) | 1.056 (0.005) | 0.922 (0.002) | 1.056 (0.005) | 0.921 (0.002) | 1.053 (0.004) | 0.934 (0.002) | 1.082 (0.005) | 0.935 (0.002) | 1.210 (0.009) |

Table A11: Coverage and size of prediction sets constructed with different methods for groups formed by Children. All groups have similar performance, and none of them are subject to unfairness/undercoverage. See corresponding plots in Figure A9.

| Group | Sample size | AFCP Coverage | AFCP Size | AFCP1 Coverage | AFCP1 Size | Marginal Coverage | Marginal Size | Partial Coverage | Partial Size | Exhaustive Coverage | Exhaustive Size |
|---|---|---|---|---|---|---|---|---|---|---|---|
| **1** | | | | | | | | | | | |
| 1 | 200 | 0.904 (0.004) | 1.966 (0.033) | 0.914 (0.004) | 2.034 (0.034) | 0.884 (0.005) | 1.853 (0.032) | 0.968 (0.003) | 2.673 (0.048) | 0.999 (0.001) | 3.980 (0.014) |
| 1 | 500 | 0.911 (0.004) | 1.728 (0.019) | 0.911 (0.004) | 1.730 (0.019) | 0.880 (0.005) | 1.600 (0.019) | 0.949 (0.003) | 2.047 (0.026) | 1.000 (0.000) | 4.000 (0.000) |
| 1 | 1000 | 0.911 (0.003) | 1.581 (0.014) | 0.911 (0.003) | 1.581 (0.014) | 0.884 (0.003) | 1.475 (0.013) | 0.940 (0.003) | 1.784 (0.018) | 1.000 (0.000) | 3.998 (0.001) |
| 1 | 2000 | 0.910 (0.002) | 1.428 (0.009) | 0.910 (0.002) | 1.428 (0.009) | 0.884 (0.003) | 1.326 (0.007) | 0.935 (0.003) | 1.554 (0.012) | 0.993 (0.001) | 3.758 (0.014) |
| 1 | 5000 | 0.901 (0.003) | 1.198 (0.009) | 0.901 (0.003) | 1.198 (0.009) | 0.884 (0.003) | 1.122 (0.006) | 0.919 (0.003) | 1.238 (0.009) | 0.935 (0.003) | 1.463 (0.016) |
| **2** | | | | | | | | | | | |
| 2 | 200 | 0.928 (0.004) | 1.917 (0.032) | 0.939 (0.003) | 1.983 (0.033) | 0.900 (0.004) | 1.794 (0.030) | 0.978 (0.002) | 2.629 (0.051) | 0.999 (0.001) | 3.980 (0.014) |
| 2 | 500 | 0.933 (0.003) | 1.679 (0.017) | 0.934 (0.003) | 1.680 (0.017) | 0.905 (0.003) | 1.556 (0.017) | 0.961 (0.002) | 1.957 (0.024) | 1.000 (0.000) | 4.000 (0.000) |
| 2 | 1000 | 0.929 (0.003) | 1.535 (0.012) | 0.929 (0.003) | 1.535 (0.012) | 0.905 (0.003) | 1.427 (0.011) | 0.956 (0.002) | 1.725 (0.018) | 1.000 (0.000) | 3.997 (0.002) |
| 2 | 2000 | 0.927 (0.003) | 1.387 (0.008) | 0.927 (0.003) | 1.387 (0.008) | 0.899 (0.003) | 1.283 (0.007) | 0.942 (0.002) | 1.473 (0.010) | 0.991 (0.001) | 3.739 (0.013) |
| 2 | 5000 | 0.911 (0.003) | 1.175 (0.008) | 0.911 (0.003) | 1.175 (0.008) | 0.898 (0.004) | 1.105 (0.005) | 0.925 (0.003) | 1.198 (0.008) | 0.941 (0.003) | 1.370 (0.012) |
| **3** | | | | | | | | | | | |
| 3 | 200 | 0.938 (0.004) | 1.806 (0.031) | 0.950 (0.003) | 1.882 (0.032) | 0.908 (0.004) | 1.670 (0.027) | 0.984 (0.002) | 2.544 (0.051) | 1.000 (0.000) | 3.980 (0.014) |
| 3 | 500 | 0.945 (0.002) | 1.579 (0.015) | 0.945 (0.002) | 1.581 (0.015) | 0.916 (0.003) | 1.460 (0.016) | 0.970 (0.002) | 1.834 (0.024) | 1.000 (0.000) | 4.000 (0.000) |
| 3 | 1000 | 0.941 (0.002) | 1.451 (0.011) | 0.941 (0.002) | 1.451 (0.011) | 0.913 (0.003) | 1.332 (0.011) | 0.961 (0.002) | 1.592 (0.015) | 1.000 (0.000) | 3.998 (0.001) |
| 3 | 2000 | 0.942 (0.002) | 1.324 (0.008) | 0.942 (0.002) | 1.324 (0.008) | 0.918 (0.002) | 1.230 (0.007) | 0.956 (0.002) | 1.401 (0.010) | 0.991 (0.001) | 3.713 (0.015) |
| 3 | 5000 | 0.929 (0.003) | 1.128 (0.008) | 0.929 (0.003) | 1.128 (0.008) | 0.914 (0.003) | 1.056 (0.005) | 0.937 (0.002) | 1.142 (0.008) | 0.939 (0.002) | 1.262 (0.012) |
| **more** | | | | | | | | | | | |
| more | 200 | 0.922 (0.004) | 1.698 (0.026) | 0.934 (0.004) | 1.772 (0.029) | 0.894 (0.004) | 1.586 (0.025) | 0.979 (0.002) | 2.392 (0.051) | 0.999 (0.000) | 3.980 (0.014) |
| more | 500 | 0.932 (0.003) | 1.502 (0.015) | 0.933 (0.003) | 1.503 (0.015) | 0.902 (0.003) | 1.396 (0.014) | 0.962 (0.003) | 1.795 (0.028) | 1.000 (0.000) | 4.000 (0.000) |
| more | 1000 | 0.931 (0.002) | 1.376 (0.010) | 0.931 (0.002) | 1.376 (0.010) | 0.897 (0.003) | 1.266 (0.010) | 0.952 (0.002) | 1.519 (0.017) | 1.000 (0.000) | 3.996 (0.002) |
| more | 2000 | 0.931 (0.003) | 1.296 (0.008) | 0.931 (0.003) | 1.296 (0.008) | 0.907 (0.003) | 1.197 (0.007) | 0.948 (0.002) | 1.371 (0.009) | 0.993 (0.001) | 3.739 (0.015) |
| more | 5000 | 0.921 (0.003) | 1.121 (0.008) | 0.921 (0.003) | 1.121 (0.008) | 0.909 (0.003) | 1.055 (0.005) | 0.931 (0.003) | 1.139 (0.008) | 0.940 (0.003) | 1.274 (0.013) |

Table A12: Coverage and size of prediction sets constructed with different methods for groups formed by Finance. All groups have similar performance, and none of them are subject to unfairness/undercoverage. See corresponding plots in Figure A10.

| Group | Sample size | AFCP Coverage | AFCP Size | AFCP1 Coverage | AFCP1 Size | Marginal Coverage | Marginal Size | Partial Coverage | Partial Size | Exhaustive Coverage | Exhaustive Size |
|---|---|---|---|---|---|---|---|---|---|---|---|
| **Convenient** | | | | | | | | | | | |
| Convenient | 200 | 0.921 (0.004) | 1.873 (0.029) | 0.933 (0.003) | 1.946 (0.031) | 0.894 (0.004) | 1.755 (0.028) | 0.976 (0.002) | 2.596 (0.040) | 0.999 (0.001) | 3.980 (0.014) |
| Convenient | 500 | 0.930 (0.002) | 1.665 (0.015) | 0.930 (0.002) | 1.667 (0.015) | 0.901 (0.003) | 1.547 (0.015) | 0.961 (0.002) | 1.961 (0.022) | 1.000 (0.000) | 4.000 (0.000) |
| Convenient | 1000 | 0.927 (0.002) | 1.531 (0.010) | 0.927 (0.002) | 1.531 (0.010) | 0.897 (0.002) | 1.414 (0.009) | 0.951 (0.002) | 1.707 (0.015) | 1.000 (0.000) | 3.997 (0.001) |
| Convenient | 2000 | 0.930 (0.002) | 1.393 (0.008) | 0.930 (0.002) | 1.393 (0.008) | 0.905 (0.002) | 1.293 (0.007) | 0.947 (0.002) | 1.490 (0.009) | 0.992 (0.001) | 3.739 (0.008) |
| Convenient | 5000 | 0.917 (0.002) | 1.165 (0.006) | 0.917 (0.002) | 1.165 (0.006) | 0.906 (0.002) | 1.097 (0.004) | 0.932 (0.002) | 1.194 (0.006) | 0.939 (0.002) | 1.347 (0.010) |
| **Inconvenient** | | | | | | | | | | | |
| Inconvenient | 200 | 0.924 (0.004) | 1.821 (0.028) | 0.935 (0.003) | 1.889 (0.029) | 0.899 (0.004) | 1.698 (0.027) | 0.978 (0.002) | 2.521 (0.040) | 0.999 (0.001) | 3.980 (0.014) |
| Inconvenient | 500 | 0.931 (0.002) | 1.582 (0.015) | 0.931 (0.002) | 1.583 (0.015) | 0.901 (0.003) | 1.462 (0.015) | 0.961 (0.002) | 1.857 (0.022) | 1.000 (0.000) | 4.000 (0.000) |
| Inconvenient | 1000 | 0.929 (0.002) | 1.440 (0.010) | 0.929 (0.002) | 1.440 (0.010) | 0.902 (0.003) | 1.336 (0.010) | 0.953 (0.002) | 1.601 (0.015) | 1.000 (0.000) | 3.997 (0.001) |
| Inconvenient | 2000 | 0.925 (0.002) | 1.325 (0.006) | 0.925 (0.002) | 1.325 (0.006) | 0.899 (0.002) | 1.224 (0.006) | 0.943 (0.002) | 1.410 (0.008) | 0.993 (0.001) | 3.736 (0.010) |
| Inconvenient | 5000 | 0.915 (0.002) | 1.146 (0.006) | 0.915 (0.002) | 1.146 (0.006) | 0.897 (0.002) | 1.072 (0.004) | 0.925 (0.002) | 1.166 (0.006) | 0.938 (0.002) | 1.337 (0.010) |

Table A13: Coverage and size of prediction sets constructed with different methods for groups formed by Social. All groups have similar performance, and none of them are subject to unfairness/undercoverage. See corresponding plots in Figure A11.

| Group | Sample size | AFCP Coverage | AFCP Size | AFCP1 Coverage | AFCP1 Size | Marginal Coverage | Marginal Size | Partial Coverage | Partial Size | Exhaustive Coverage | Exhaustive Size |
|---|---|---|---|---|---|---|---|---|---|---|---|
| **Nonprob** | | | | | | | | | | | |
| Nonprob | 200 | 0.922 (0.004) | 1.894 (0.031) | 0.933 (0.004) | 1.964 (0.032) | 0.896 (0.004) | 1.775 (0.030) | 0.976 (0.002) | 2.605 (0.041) | 0.999 (0.001) | 3.980 (0.014) |
| Nonprob | 500 | 0.926 (0.003) | 1.655 (0.018) | 0.926 (0.003) | 1.657 (0.017) | 0.897 (0.004) | 1.538 (0.018) | 0.959 (0.002) | 1.943 (0.023) | 1.000 (0.000) | 4.000 (0.000) |
| Nonprob | 1000 | 0.924 (0.003) | 1.509 (0.012) | 0.924 (0.003) | 1.509 (0.012) | 0.899 (0.003) | 1.406 (0.012) | 0.951 (0.002) | 1.694 (0.017) | 1.000 (0.000) | 3.998 (0.001) |
| Nonprob | 2000 | 0.922 (0.002) | 1.360 (0.008) | 0.922 (0.002) | 1.360 (0.008) | 0.896 (0.003) | 1.258 (0.008) | 0.942 (0.002) | 1.456 (0.011) | 0.991 (0.001) | 3.740 (0.012) |
| Nonprob | 5000 | 0.917 (0.003) | 1.164 (0.007) | 0.917 (0.003) | 1.164 (0.007) | 0.901 (0.003) | 1.092 (0.005) | 0.927 (0.002) | 1.186 (0.007) | 0.938 (0.003) | 1.360 (0.013) |
| **Problematic** | | | | | | | | | | | |
| Problematic | 200 | 0.937 (0.004) | 1.846 (0.030) | 0.949 (0.004) | 1.916 (0.030) | 0.913 (0.004) | 1.726 (0.028) | 0.984 (0.002) | 2.546 (0.041) | 1.000 (0.000) | 3.980 (0.014) |
| Problematic | 500 | 0.945 (0.002) | 1.626 (0.014) | 0.945 (0.002) | 1.628 (0.014) | 0.913 (0.003) | 1.500 (0.015) | 0.967 (0.002) | 1.892 (0.023) | 1.000 (0.000) | 4.000 (0.000) |
| Problematic | 1000 | 0.941 (0.002) | 1.486 (0.010) | 0.941 (0.002) | 1.486 (0.010) | 0.909 (0.003) | 1.369 (0.009) | 0.958 (0.002) | 1.639 (0.014) | 1.000 (0.000) | 3.996 (0.002) |
| Problematic | 2000 | 0.936 (0.002) | 1.337 (0.008) | 0.936 (0.002) | 1.337 (0.008) | 0.909 (0.003) | 1.238 (0.007) | 0.950 (0.002) | 1.417 (0.009) | 0.993 (0.001) | 3.726 (0.012) |
| Problematic | 5000 | 0.916 (0.003) | 1.135 (0.006) | 0.916 (0.003) | 1.135 (0.006) | 0.903 (0.003) | 1.066 (0.004) | 0.931 (0.002) | 1.160 (0.006) | 0.939 (0.002) | 1.312 (0.010) |
| **Slightly_prob** | | | | | | | | | | | |
| Slightly_prob | 200 | 0.909 (0.004) | 1.800 (0.027) | 0.921 (0.004) | 1.871 (0.029) | 0.881 (0.004) | 1.677 (0.026) | 0.971 (0.003) | 2.526 (0.044) | 0.999 (0.001) | 3.980 (0.014) |
| Slightly_prob | 500 | 0.920 (0.003) | 1.590 (0.016) | 0.920 (0.003) | 1.590 (0.016) | 0.891 (0.003) | 1.476 (0.016) | 0.956 (0.002) | 1.893 (0.026) | 1.000 (0.000) | 4.000 (0.000) |
| Slightly_prob | 1000 | 0.919 (0.003) | 1.462 (0.011) | 0.919 (0.003) | 1.462 (0.011) | 0.891 (0.003) | 1.351 (0.011) | 0.947 (0.002) | 1.629 (0.018) | 1.000 (0.000) | 3.997 (0.001) |
| Slightly_prob | 2000 | 0.925 (0.002) | 1.378 (0.008) | 0.925 (0.002) | 1.378 (0.008) | 0.900 (0.003) | 1.282 (0.006) | 0.944 (0.002) | 1.475 (0.009) | 0.993 (0.001) | 3.746 (0.011) |
| Slightly_prob | 5000 | 0.914 (0.002) | 1.167 (0.006) | 0.914 (0.002) | 1.167 (0.006) | 0.900 (0.003) | 1.096 (0.005) | 0.927 (0.002) | 1.193 (0.007) | 0.939 (0.002) | 1.354 (0.011) |

Table A14: Coverage and size of prediction sets constructed with different methods for groups formed by Health. All groups have similar performance, and none of them are subject to unfairness/undercoverage. See corresponding plots in Figure A12.

| Group | Sample size | AFCP | | AFCP1 | | Marginal | | Partial | | Exhaustive | |
|---|---|---|---|---|---|---|---|---|---|---|---|
| | | Coverage | Size | Coverage | Size | Coverage | Size | Coverage | Size | Coverage | Size |
| **Not-Recommended** | | | | | | | | | | | |
| Not-Recommended | 200 | 0.961 (0.003) | 1.357 (0.031) | 0.967 (0.003) | 1.419 (0.031) | 0.942 (0.003) | 1.223 (0.029) | 0.990 (0.001) | 1.951 (0.052) | 1.000 (0.000) | 3.980 (0.014) |
| Not-Recommended | 500 | 0.950 (0.002) | 1.136 (0.008) | 0.950 (0.002) | 1.139 (0.008) | 0.931 (0.003) | 1.006 (0.007) | 0.971 (0.002) | 1.228 (0.013) | 1.000 (0.000) | 4.000 (0.000) |
| Not-Recommended | 1000 | 0.942 (0.002) | 1.086 (0.007) | 0.942 (0.002) | 1.086 (0.007) | 0.925 (0.003) | 0.985 (0.005) | 0.958 (0.002) | 1.130 (0.008) | 1.000 (0.000) | 3.995 (0.002) |
| Not-Recommended | 2000 | 0.935 (0.002) | 1.051 (0.005) | 0.935 (0.002) | 1.051 (0.005) | 0.922 (0.002) | 0.966 (0.003) | 0.948 (0.002) | 1.072 (0.005) | 0.994 (0.001) | 3.697 (0.013) |
| Not-Recommended | 5000 | 0.927 (0.002) | 1.033 (0.005) | 0.927 (0.002) | 1.033 (0.005) | 0.915 (0.002) | 0.969 (0.003) | 0.937 (0.002) | 1.048 (0.005) | 0.940 (0.002) | 1.164 (0.009) |
| **Priority** | | | | | | | | | | | |
| Priority | 200 | 0.938 (0.004) | 2.249 (0.038) | 0.948 (0.003) | 2.326 (0.039) | 0.909 (0.004) | 2.145 (0.038) | 0.979 (0.002) | 2.987 (0.044) | 1.000 (0.000) | 3.980 (0.014) |
| Priority | 500 | 0.946 (0.002) | 1.866 (0.026) | 0.946 (0.002) | 1.867 (0.026) | 0.912 (0.003) | 1.751 (0.026) | 0.967 (0.002) | 2.229 (0.036) | 1.000 (0.000) | 4.000 (0.000) |
| Priority | 1000 | 0.945 (0.002) | 1.601 (0.015) | 0.945 (0.002) | 1.601 (0.015) | 0.907 (0.003) | 1.484 (0.016) | 0.960 (0.002) | 1.788 (0.021) | 1.000 (0.000) | 3.998 (0.001) |
| Priority | 2000 | 0.933 (0.002) | 1.384 (0.010) | 0.933 (0.002) | 1.384 (0.010) | 0.901 (0.002) | 1.278 (0.009) | 0.948 (0.002) | 1.476 (0.013) | 0.991 (0.001) | 3.734 (0.011) |
| Priority | 5000 | 0.908 (0.003) | 1.140 (0.008) | 0.908 (0.003) | 1.140 (0.008) | 0.896 (0.003) | 1.073 (0.005) | 0.924 (0.003) | 1.167 (0.007) | 0.937 (0.002) | 1.330 (0.010) |
| **Recommended** | | | | | | | | | | | |
| Recommended | 200 | 0.871 (0.006) | 1.933 (0.031) | 0.888 (0.006) | 2.006 (0.033) | 0.839 (0.006) | 1.809 (0.028) | 0.962 (0.004) | 2.736 (0.043) | 0.998 (0.001) | 3.980 (0.014) |
| Recommended | 500 | 0.895 (0.004) | 1.869 (0.020) | 0.896 (0.004) | 1.870 (0.020) | 0.859 (0.004) | 1.757 (0.020) | 0.945 (0.003) | 2.273 (0.029) | 1.000 (0.000) | 4.000 (0.000) |
| Recommended | 1000 | 0.898 (0.003) | 1.775 (0.014) | 0.898 (0.003) | 1.775 (0.014) | 0.867 (0.003) | 1.662 (0.014) | 0.938 (0.002) | 2.053 (0.023) | 1.000 (0.000) | 3.998 (0.001) |
| Recommended | 2000 | 0.914 (0.003) | 1.646 (0.011) | 0.914 (0.003) | 1.646 (0.011) | 0.883 (0.003) | 1.539 (0.010) | 0.940 (0.002) | 1.807 (0.013) | 0.991 (0.001) | 3.780 (0.010) |
| Recommended | 5000 | 0.912 (0.003) | 1.293 (0.010) | 0.912 (0.003) | 1.293 (0.010) | 0.894 (0.002) | 1.213 (0.007) | 0.924 (0.002) | 1.324 (0.010) | 0.939 (0.002) | 1.532 (0.014) |

Next, we provide numerical experimental results comparing AFCP with the **label-conditional** adaptive equalized coverage (see Equation (A14)) with other benchmark methods.

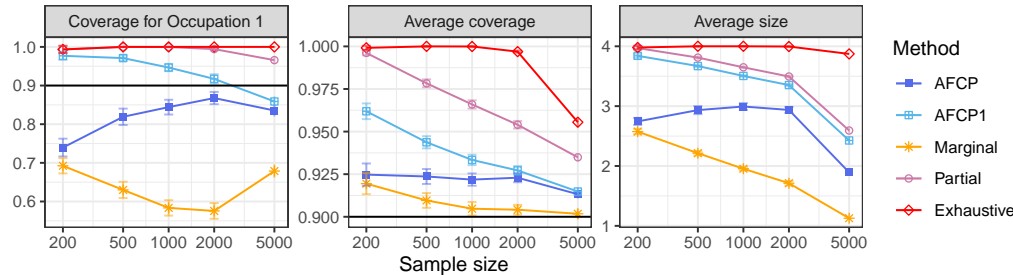

Figure A13: Performance on prediction sets constructed by different methods on the nursery data as a function of the total number of training and calibration points. Our method (AFCP) leads to more informative sets with a lower average width and higher conditional coverage on the Pretentious group. The error bars indicate 2 standard errors.

Table A15: Average coverage and average size of prediction sets for all test samples constructed by different methods as a function of the training and calibration size. All methods obtain coverage beyond 0.9, while our AFCP and AFCP1 methods, along with the Marginal method, produce the smallest, thus, the most informative, prediction sets. Red numbers indicate the small size of prediction sets. See corresponding plots in Figure A13.

| Sample size | AFCP Coverage | AFCP Size | AFCP1 Coverage | AFCP1 Size | Marginal Coverage | Marginal Size | Partial Coverage | Partial Size | Exhaustive Coverage | Exhaustive Size |
|---|---|---|---|---|---|---|---|---|---|---|
| 200 | 0.925 (0.003) | 2.745 (0.029) | 0.962 (0.002) | 3.841 (0.017) | 0.919 (0.003) | 2.575 (0.023) | 0.996 (0.001) | 3.970 (0.014) | 0.999 (0.001) | 3.980 (0.014) |
| 500 | 0.924 (0.002) | 2.934 (0.034) | 0.944 (0.002) | 3.670 (0.008) | 0.910 (0.002) | 2.214 (0.019) | 0.978 (0.001) | 3.811 (0.007) | 1.000 (0.000) | 4.000 (0.000) |
| 1000 | 0.922 (0.002) | 2.994 (0.029) | 0.933 (0.002) | 3.505 (0.006) | 0.905 (0.002) | 1.954 (0.015) | 0.966 (0.001) | 3.649 (0.006) | 1.000 (0.000) | 4.000 (0.000) |
| 2000 | 0.923 (0.001) | 2.937 (0.025) | 0.927 (0.001) | 3.352 (0.010) | 0.904 (0.001) | 1.712 (0.019) | 0.954 (0.001) | 3.495 (0.007) | 0.997 (0.000) | 3.996 (0.000) |
| 5000 | 0.913 (0.001) | 1.893 (0.022) | 0.915 (0.001) | 2.426 (0.022) | 0.902 (0.001) | 1.127 (0.005) | 0.935 (0.001) | 2.593 (0.019) | 0.956 (0.001) | 3.872 (0.002) |

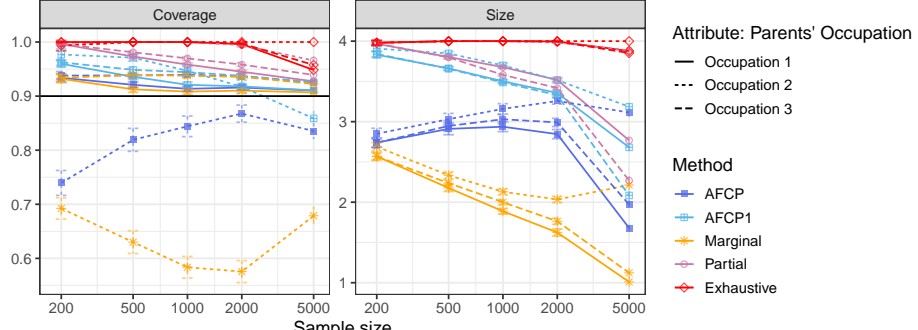

Figure A14: Coverage and size of prediction sets constructed with different methods for groups formed by Parents' Occupation. For the Occupation 1 group, the Marginal method (dashed orange lines) fails to detect and correct for its undercoverage, and the Exhaustive method produces prediction sets that are too conservative to be helpful. In contrast, our AFCP and AFCP1 methods correct the undercoverage and maintain small prediction sets. See Table A16 for details and standard errors.

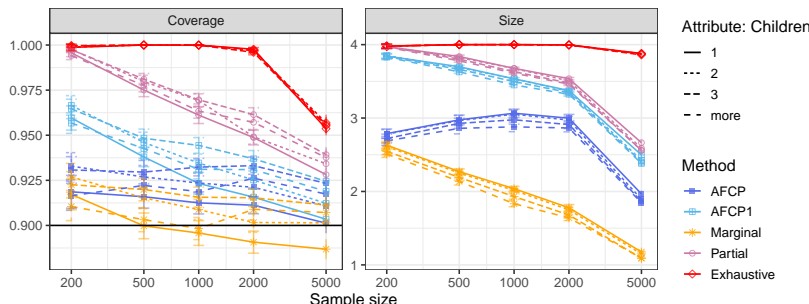

Figure A15: Coverage and size of prediction sets constructed with different methods for groups formed by Children. All groups have similar performance, and none of them are subject to unfairness/undercoverage. See Table A17 for numerical details and standard errors.

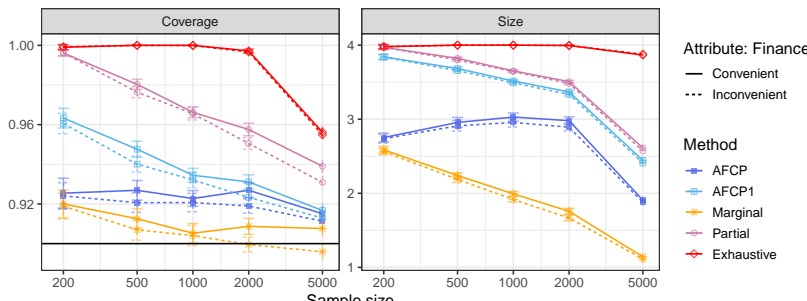

Figure A16: Coverage and size of prediction sets constructed with different methods for groups formed by Finance. All groups have similar performance, and none of them are subject to unfairness/undercoverage. See Table A18 for numerical details and standard errors.

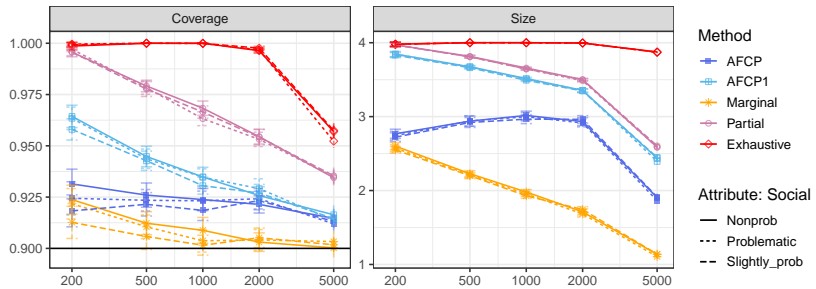

Figure A17: Coverage and size of prediction sets constructed with different methods for groups formed by Social. All groups have similar performance, and none of them are subject to unfairness/undercoverage. See Table A19 for numerical details and standard errors.

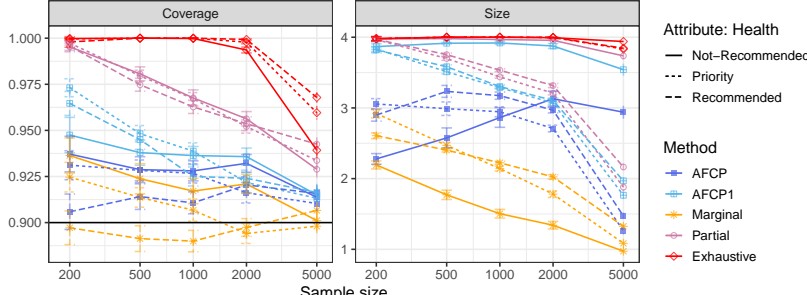

Figure A18: Coverage and size of prediction sets constructed with different methods for groups formed by Health. All groups have similar performances, and none of them are subject to unfairness/undercoverage. See Table A20 for numerical details and standard errors.

Table A16: Coverage and size of prediction sets constructed with different methods for groups formed by Parents' Occupation. Green numbers indicate low coverage and red numbers indicate the small size of prediction sets. See corresponding plots in Figure A14.

| Group | Sample size | AFCP Coverage | AFCP Size | AFCP1 Coverage | AFCP1 Size | Marginal Coverage | Marginal Size | Partial Coverage | Partial Size | Exhaustive Coverage | Exhaustive Size |
|---|---|---|---|---|---|---|---|---|---|---|---|
| **Occupation 0** | | | | | | | | | | | |
| Occupation 0 | 200 | 0.934 (0.004) | 2.740 (0.029) | 0.960 (0.003) | 3.831 (0.018) | 0.932 (0.004) | 2.569 (0.024) | 0.996 (0.001) | 3.967 (0.014) | 1.000 (0.000) | 3.980 (0.014) |
| Occupation 0 | 500 | 0.921 (0.003) | 2.911 (0.036) | 0.936 (0.002) | 3.662 (0.011) | 0.913 (0.003) | 2.177 (0.022) | 0.973 (0.002) | 3.803 (0.008) | 1.000 (0.000) | 4.000 (0.000) |
| Occupation 0 | 1000 | 0.914 (0.002) | 2.937 (0.030) | 0.921 (0.002) | 3.502 (0.009) | 0.908 (0.003) | 1.888 (0.017) | 0.958 (0.002) | 3.677 (0.007) | 1.000 (0.000) | 4.000 (0.000) |
| Occupation 0 | 2000 | 0.916 (0.002) | 2.845 (0.028) | 0.918 (0.002) | 3.356 (0.012) | 0.910 (0.002) | 1.624 (0.022) | 0.945 (0.002) | 3.518 (0.008) | 0.996 (0.000) | 3.995 (0.000) |
| Occupation 0 | 5000 | 0.910 (0.002) | 1.679 (0.021) | 0.911 (0.002) | 2.681 (0.019) | 0.907 (0.002) | 1.009 (0.004) | 0.927 (0.002) | 2.768 (0.016) | 0.948 (0.002) | 3.877 (0.003) |
| **Occupation 1** | | | | | | | | | | | |
| Occupation 1 | 200 | 0.739 (0.012) | 2.850 (0.034) | 0.977 (0.005) | 3.907 (0.016) | 0.692 (0.010) | 2.689 (0.026) | 0.993 (0.005) | 3.980 (0.014) | 0.993 (0.005) | 3.980 (0.014) |
| Occupation 1 | 500 | 0.819 (0.011) | 3.031 (0.036) | 0.971 (0.003) | 3.848 (0.011) | 0.630 (0.011) | 2.334 (0.022) | 1.000 (0.000) | 4.000 (0.000) | 1.000 (0.000) | 4.000 (0.000) |
| Occupation 1 | 1000 | 0.844 (0.010) | 3.159 (0.032) | 0.947 (0.005) | 3.699 (0.015) | 0.583 (0.010) | 2.130 (0.019) | 1.000 (0.000) | 3.999 (0.001) | 1.000 (0.000) | 4.000 (0.000) |
| Occupation 1 | 2000 | 0.868 (0.008) | 3.257 (0.019) | 0.917 (0.006) | 3.516 (0.016) | 0.575 (0.010) | 2.034 (0.019) | 0.994 (0.002) | 3.982 (0.004) | 1.000 (0.000) | 4.000 (0.000) |
| Occupation 1 | 5000 | 0.835 (0.008) | 3.114 (0.027) | 0.859 (0.007) | 3.185 (0.025) | 0.679 (0.009) | 2.213 (0.019) | 0.966 (0.004) | 3.885 (0.010) | 1.000 (0.000) | 4.000 (0.000) |
| **Occupation 2** | | | | | | | | | | | |
| Occupation 2 | 200 | 0.938 (0.004) | 2.739 (0.031) | 0.962 (0.003) | 3.841 (0.018) | 0.934 (0.003) | 2.569 (0.024) | 0.996 (0.001) | 3.973 (0.014) | 0.999 (0.001) | 3.980 (0.014) |
| Occupation 2 | 500 | 0.938 (0.002) | 2.947 (0.033) | 0.949 (0.002) | 3.658 (0.012) | 0.939 (0.002) | 2.237 (0.017) | 0.981 (0.001) | 3.797 (0.011) | 1.000 (0.000) | 4.000 (0.000) |
| Occupation 2 | 1000 | 0.939 (0.002) | 3.030 (0.028) | 0.944 (0.002) | 3.486 (0.007) | 0.938 (0.002) | 2.000 (0.015) | 0.970 (0.002) | 3.581 (0.007) | 1.000 (0.000) | 4.000 (0.000) |
| Occupation 2 | 2000 | 0.936 (0.002) | 2.990 (0.026) | 0.938 (0.002) | 3.329 (0.010) | 0.936 (0.002) | 1.763 (0.019) | 0.958 (0.002) | 3.417 (0.008) | 0.997 (0.000) | 3.996 (0.000) |
| Occupation 2 | 5000 | 0.925 (0.002) | 1.971 (0.029) | 0.926 (0.002) | 2.083 (0.030) | 0.922 (0.002) | 1.123 (0.007) | 0.939 (0.002) | 2.269 (0.028) | 0.958 (0.002) | 3.852 (0.003) |

Table A17: Coverage and size of prediction sets constructed with different methods for groups formed by Children. All groups have similar performance, and none of them are subject to unfairness/undercoverage. See corresponding plots in Figure A15.

| Group | Sample size | AFCP Coverage | AFCP Size | AFCP1 Coverage | AFCP1 Size | Marginal Coverage | Marginal Size | Partial Coverage | Partial Size | Exhaustive Coverage | Exhaustive Size |
|---|---|---|---|---|---|---|---|---|---|---|---|
| **1** | | | | | | | | | | | |
| 1 | 200 | 0.919 (0.005) | 2.788 (0.031) | 0.960 (0.003) | 3.844 (0.017) | 0.917 (0.004) | 2.631 (0.026) | 0.996 (0.001) | 3.972 (0.014) | 0.999 (0.001) | 3.980 (0.014) |
| 1 | 500 | 0.916 (0.003) | 2.975 (0.033) | 0.938 (0.003) | 3.699 (0.010) | 0.900 (0.004) | 2.271 (0.018) | 0.975 (0.002) | 3.838 (0.010) | 1.000 (0.000) | 4.000 (0.000) |
| 1 | 1000 | 0.912 (0.003) | 3.066 (0.028) | 0.923 (0.003) | 3.534 (0.008) | 0.896 (0.003) | 2.038 (0.016) | 0.961 (0.002) | 3.677 (0.008) | 1.000 (0.000) | 4.000 (0.000) |
| 1 | 2000 | 0.911 (0.002) | 2.998 (0.026) | 0.916 (0.002) | 3.375 (0.012) | 0.891 (0.003) | 1.779 (0.021) | 0.949 (0.002) | 3.536 (0.009) | 0.998 (0.000) | 3.997 (0.001) |
| 1 | 5000 | 0.901 (0.003) | 1.967 (0.023) | 0.903 (0.003) | 2.498 (0.024) | 0.887 (0.003) | 1.176 (0.007) | 0.928 (0.002) | 2.664 (0.020) | 0.954 (0.002) | 3.880 (0.003) |
| **2** | | | | | | | | | | | |
| 2 | 200 | 0.933 (0.004) | 2.782 (0.030) | 0.967 (0.003) | 3.851 (0.017) | 0.927 (0.004) | 2.613 (0.024) | 0.998 (0.001) | 3.972 (0.014) | 0.999 (0.001) | 3.980 (0.014) |
| 2 | 500 | 0.927 (0.003) | 2.966 (0.034) | 0.946 (0.003) | 3.686 (0.010) | 0.915 (0.003) | 2.251 (0.020) | 0.980 (0.002) | 3.824 (0.008) | 1.000 (0.000) | 4.000 (0.000) |
| 2 | 1000 | 0.923 (0.003) | 3.052 (0.029) | 0.935 (0.002) | 3.536 (0.007) | 0.909 (0.003) | 2.015 (0.014) | 0.969 (0.002) | 3.675 (0.007) | 1.000 (0.000) | 4.000 (0.000) |
| 2 | 2000 | 0.921 (0.003) | 2.972 (0.026) | 0.925 (0.003) | 3.364 (0.011) | 0.902 (0.003) | 1.750 (0.020) | 0.949 (0.002) | 3.506 (0.007) | 0.997 (0.001) | 3.996 (0.001) |
| 2 | 5000 | 0.911 (0.003) | 1.895 (0.024) | 0.912 (0.003) | 2.425 (0.023) | 0.901 (0.003) | 1.150 (0.007) | 0.934 (0.002) | 2.589 (0.022) | 0.957 (0.002) | 3.876 (0.004) |
| **3** | | | | | | | | | | | |
| 3 | 200 | 0.931 (0.004) | 2.721 (0.030) | 0.965 (0.003) | 3.837 (0.017) | 0.923 (0.004) | 2.547 (0.024) | 0.997 (0.001) | 3.971 (0.015) | 1.000 (0.000) | 3.980 (0.014) |
| 3 | 500 | 0.930 (0.003) | 2.928 (0.035) | 0.948 (0.003) | 3.664 (0.010) | 0.920 (0.003) | 2.191 (0.020) | 0.981 (0.002) | 3.803 (0.009) | 1.000 (0.000) | 4.000 (0.000) |
| 3 | 1000 | 0.932 (0.002) | 2.976 (0.029) | 0.944 (0.002) | 3.497 (0.008) | 0.916 (0.003) | 1.931 (0.019) | 0.970 (0.002) | 3.632 (0.007) | 1.000 (0.000) | 4.000 (0.000) |
| 3 | 2000 | 0.933 (0.002) | 2.911 (0.027) | 0.937 (0.002) | 3.347 (0.011) | 0.915 (0.002) | 1.680 (0.021) | 0.961 (0.002) | 3.477 (0.009) | 0.996 (0.001) | 3.995 (0.001) |
| 3 | 5000 | 0.923 (0.002) | 1.865 (0.024) | 0.924 (0.002) | 2.400 (0.023) | 0.911 (0.003) | 1.090 (0.007) | 0.939 (0.002) | 2.559 (0.021) | 0.955 (0.002) | 3.860 (0.004) |
| **more** | | | | | | | | | | | |
| more | 200 | 0.916 (0.004) | 2.688 (0.031) | 0.957 (0.003) | 3.830 (0.017) | 0.910 (0.004) | 2.509 (0.023) | 0.994 (0.001) | 3.967 (0.014) | 0.999 (0.000) | 3.980 (0.014) |
| more | 500 | 0.922 (0.003) | 2.860 (0.038) | 0.943 (0.003) | 3.629 (0.010) | 0.903 (0.003) | 2.137 (0.024) | 0.978 (0.002) | 3.779 (0.009) | 1.000 (0.000) | 4.000 (0.000) |
| more | 1000 | 0.919 (0.003) | 2.880 (0.033) | 0.931 (0.003) | 3.451 (0.009) | 0.898 (0.003) | 1.833 (0.020) | 0.963 (0.002) | 3.611 (0.008) | 1.000 (0.000) | 4.000 (0.000) |
| more | 2000 | 0.926 (0.003) | 2.865 (0.026) | 0.931 (0.003) | 3.324 (0.012) | 0.909 (0.003) | 1.641 (0.019) | 0.957 (0.002) | 3.461 (0.009) | 0.997 (0.001) | 3.996 (0.001) |
| more | 5000 | 0.917 (0.003) | 1.846 (0.024) | 0.919 (0.003) | 2.382 (0.023) | 0.907 (0.003) | 1.093 (0.006) | 0.938 (0.002) | 2.559 (0.021) | 0.956 (0.002) | 3.872 (0.003) |

Table A18: Coverage and size of prediction sets constructed with different methods for groups formed by Finance. All groups have similar performance, and none of them are subject to unfairness/undercoverage. See corresponding plots in Figure A16.

| Group | Sample size | AFCP Coverage | AFCP Size | AFCP1 Coverage | AFCP1 Size | Marginal Coverage | Marginal Size | Partial Coverage | Partial Size | Exhaustive Coverage | Exhaustive Size |
|---|---|---|---|---|---|---|---|---|---|---|---|
| **Convenient** | | | | | | | | | | | |
| Convenient | 200 | 0.925 (0.004) | 2.753 (0.030) | 0.963 (0.002) | 3.843 (0.017) | 0.920 (0.004) | 2.587 (0.024) | 0.996 (0.001) | 3.972 (0.014) | 0.999 (0.001) | 3.980 (0.014) |
| Convenient | 500 | 0.927 (0.002) | 2.957 (0.033) | 0.948 (0.002) | 3.684 (0.009) | 0.913 (0.002) | 2.240 (0.018) | 0.980 (0.001) | 3.822 (0.008) | 1.000 (0.000) | 4.000 (0.000) |
| Convenient | 1000 | 0.923 (0.002) | 3.029 (0.027) | 0.934 (0.002) | 3.517 (0.007) | 0.905 (0.002) | 1.993 (0.015) | 0.966 (0.001) | 3.653 (0.006) | 1.000 (0.000) | 4.000 (0.000) |
| Convenient | 2000 | 0.927 (0.002) | 2.980 (0.026) | 0.931 (0.002) | 3.369 (0.010) | 0.909 (0.002) | 1.757 (0.020) | 0.958 (0.002) | 3.508 (0.008) | 0.997 (0.000) | 3.996 (0.000) |
| Convenient | 5000 | 0.915 (0.002) | 1.908 (0.022) | 0.917 (0.002) | 2.445 (0.022) | 0.908 (0.002) | 1.143 (0.006) | 0.939 (0.002) | 2.614 (0.019) | 0.956 (0.002) | 3.866 (0.003) |
| **Inconvenient** | | | | | | | | | | | |
| Inconvenient | 200 | 0.924 (0.003) | 2.737 (0.029) | 0.961 (0.003) | 3.839 (0.017) | 0.919 (0.003) | 2.564 (0.023) | 0.996 (0.001) | 3.969 (0.014) | 0.999 (0.001) | 3.980 (0.014) |
| Inconvenient | 500 | 0.921 (0.002) | 2.911 (0.035) | 0.940 (0.002) | 3.656 (0.009) | 0.907 (0.003) | 2.188 (0.021) | 0.976 (0.001) | 3.800 (0.008) | 1.000 (0.000) | 4.000 (0.000) |
| Inconvenient | 1000 | 0.921 (0.002) | 2.957 (0.030) | 0.932 (0.002) | 3.492 (0.007) | 0.904 (0.003) | 1.915 (0.016) | 0.966 (0.002) | 3.644 (0.007) | 1.000 (0.000) | 4.000 (0.000) |
| Inconvenient | 2000 | 0.919 (0.002) | 2.892 (0.026) | 0.923 (0.002) | 3.335 (0.011) | 0.899 (0.002) | 1.666 (0.020) | 0.950 (0.001) | 3.483 (0.008) | 0.997 (0.000) | 3.996 (0.000) |
| Inconvenient | 5000 | 0.911 (0.002) | 1.879 (0.023) | 0.913 (0.002) | 2.409 (0.023) | 0.896 (0.002) | 1.112 (0.005) | 0.931 (0.002) | 2.573 (0.020) | 0.955 (0.001) | 3.878 (0.002) |

Table A19: Coverage and size of prediction sets constructed with different methods for groups formed by Social. All groups have similar performance, and none of them are subject to unfairness/undercoverage. See corresponding plots in Figure A17.

| Group | Sample size | AFCP Coverage | AFCP Size | AFCP1 Coverage | AFCP1 Size | Marginal Coverage | Marginal Size | Partial Coverage | Partial Size | Exhaustive Coverage | Exhaustive Size |
|---|---|---|---|---|---|---|---|---|---|---|---|
| **Nonprob** | | | | | | | | | | | |
| Nonprob | 200 | 0.931 (0.004) | 2.768 (0.031) | 0.964 (0.003) | 3.846 (0.017) | 0.924 (0.004) | 2.601 (0.025) | 0.996 (0.001) | 3.971 (0.014) | 0.999 (0.001) | 3.980 (0.014) |
| Nonprob | 500 | 0.926 (0.003) | 2.939 (0.035) | 0.945 (0.003) | 3.674 (0.009) | 0.912 (0.003) | 2.228 (0.020) | 0.979 (0.001) | 3.815 (0.008) | 1.000 (0.000) | 4.000 (0.000) |
| Nonprob | 1000 | 0.924 (0.003) | 3.016 (0.029) | 0.935 (0.002) | 3.515 (0.008) | 0.909 (0.003) | 1.980 (0.016) | 0.968 (0.002) | 3.658 (0.007) | 1.000 (0.000) | 4.000 (0.000) |
| Nonprob | 2000 | 0.921 (0.002) | 2.934 (0.026) | 0.926 (0.002) | 3.355 (0.011) | 0.903 (0.002) | 1.711 (0.021) | 0.954 (0.002) | 3.501 (0.008) | 0.996 (0.001) | 3.996 (0.001) |
| Nonprob | 5000 | 0.915 (0.002) | 1.904 (0.024) | 0.916 (0.002) | 2.437 (0.023) | 0.900 (0.002) | 1.137 (0.007) | 0.934 (0.002) | 2.585 (0.021) | 0.957 (0.002) | 3.875 (0.003) |
| **Problematic** | | | | | | | | | | | |
| Problematic | 200 | 0.924 (0.004) | 2.742 (0.030) | 0.963 (0.003) | 3.843 (0.017) | 0.922 (0.004) | 2.573 (0.023) | 0.997 (0.001) | 3.972 (0.014) | 1.000 (0.000) | 3.980 (0.014) |
| Problematic | 500 | 0.923 (0.003) | 2.941 (0.034) | 0.944 (0.002) | 3.674 (0.009) | 0.911 (0.003) | 2.213 (0.019) | 0.978 (0.002) | 3.810 (0.007) | 1.000 (0.000) | 4.000 (0.000) |
| Problematic | 1000 | 0.923 (0.002) | 2.999 (0.029) | 0.935 (0.002) | 3.508 (0.007) | 0.904 (0.003) | 1.954 (0.016) | 0.963 (0.002) | 3.644 (0.007) | 1.000 (0.000) | 4.000 (0.000) |
| Problematic | 2000 | 0.924 (0.002) | 2.914 (0.026) | 0.929 (0.002) | 3.355 (0.011) | 0.904 (0.003) | 1.683 (0.021) | 0.953 (0.002) | 3.486 (0.008) | 0.997 (0.001) | 3.996 (0.001) |
| Problematic | 5000 | 0.912 (0.002) | 1.863 (0.022) | 0.914 (0.002) | 2.394 (0.022) | 0.903 (0.002) | 1.106 (0.005) | 0.935 (0.002) | 2.588 (0.020) | 0.952 (0.002) | 3.871 (0.003) |
| **Slightly_prob** | | | | | | | | | | | |
| Slightly_prob | 200 | 0.918 (0.004) | 2.722 (0.029) | 0.958 (0.003) | 3.831 (0.017) | 0.913 (0.004) | 2.550 (0.022) | 0.996 (0.001) | 3.968 (0.014) | 0.999 (0.001) | 3.980 (0.014) |
| Slightly_prob | 500 | 0.921 (0.003) | 2.921 (0.034) | 0.943 (0.002) | 3.662 (0.009) | 0.906 (0.003) | 2.201 (0.020) | 0.978 (0.002) | 3.809 (0.009) | 1.000 (0.000) | 4.000 (0.000) |
| Slightly_prob | 1000 | 0.919 (0.002) | 2.966 (0.030) | 0.931 (0.002) | 3.492 (0.007) | 0.902 (0.002) | 1.929 (0.017) | 0.966 (0.001) | 3.646 (0.007) | 1.000 (0.000) | 4.000 (0.000) |
| Slightly_prob | 2000 | 0.923 (0.002) | 2.963 (0.025) | 0.927 (0.002) | 3.347 (0.012) | 0.905 (0.002) | 1.743 (0.019) | 0.954 (0.002) | 3.498 (0.008) | 0.998 (0.000) | 3.997 (0.000) |
| Slightly_prob | 5000 | 0.913 (0.002) | 1.912 (0.023) | 0.914 (0.002) | 2.448 (0.023) | 0.902 (0.002) | 1.139 (0.006) | 0.935 (0.002) | 2.605 (0.019) | 0.958 (0.002) | 3.871 (0.004) |

Table A20: Coverage and size of prediction sets constructed with different methods for groups formed by Health. All groups have similar performance, and none of them are subject to unfairness/undercoverage. See corresponding plots in Figure A18.

| Group | Sample size | AFCP Coverage | AFCP Size | AFCP1 Coverage | AFCP1 Size | Marginal Coverage | Marginal Size | Partial Coverage | Partial Size | Exhaustive Coverage | Exhaustive Size |
|---|---|---|---|---|---|---|---|---|---|---|---|
| **Not-Recommended** | | | | | | | | | | | |
| Not-Recommended | 200 | 0.937 (0.005) | 2.279 (0.038) | 0.947 (0.005) | 3.868 (0.020) | 0.936 (0.005) | 2.193 (0.028) | 0.995 (0.001) | 3.975 (0.014) | 1.000 (0.000) | 3.980 (0.014) |
| Not-Recommended | 500 | 0.929 (0.004) | 2.576 (0.071) | 0.938 (0.003) | 3.915 (0.005) | 0.924 (0.004) | 1.771 (0.033) | 0.981 (0.002) | 3.981 (0.002) | 1.000 (0.000) | 4.000 (0.000) |
| Not-Recommended | 1000 | 0.928 (0.003) | 2.864 (0.069) | 0.936 (0.002) | 3.919 (0.010) | 0.917 (0.003) | 1.504 (0.031) | 0.968 (0.002) | 3.968 (0.002) | 1.000 (0.000) | 4.000 (0.000) |
| Not-Recommended | 2000 | 0.932 (0.002) | 3.129 (0.054) | 0.936 (0.002) | 3.876 (0.018) | 0.921 (0.003) | 1.341 (0.027) | 0.956 (0.002) | 3.956 (0.002) | 0.994 (0.001) | 3.994 (0.001) |
| Not-Recommended | 5000 | 0.914 (0.002) | 2.940 (0.046) | 0.915 (0.002) | 3.541 (0.043) | 0.901 (0.003) | 0.973 (0.005) | 0.929 (0.002) | 3.737 (0.027) | 0.939 (0.002) | 3.939 (0.002) |
| **Priority** | | | | | | | | | | | |
| Priority | 200 | 0.931 (0.004) | 3.055 (0.038) | 0.973 (0.002) | 3.833 (0.018) | 0.925 (0.004) | 2.922 (0.032) | 0.997 (0.001) | 3.966 (0.014) | 1.000 (0.000) | 3.980 (0.014) |
| Priority | 500 | 0.928 (0.003) | 2.989 (0.048) | 0.948 (0.002) | 3.511 (0.015) | 0.914 (0.003) | 2.470 (0.022) | 0.980 (0.002) | 3.699 (0.012) | 1.000 (0.000) | 4.000 (0.000) |
| Priority | 1000 | 0.927 (0.003) | 2.946 (0.035) | 0.939 (0.002) | 3.287 (0.009) | 0.907 (0.003) | 2.142 (0.018) | 0.967 (0.002) | 3.439 (0.009) | 1.000 (0.000) | 4.000 (0.000) |
| Priority | 2000 | 0.916 (0.003) | 2.709 (0.024) | 0.922 (0.003) | 3.069 (0.013) | 0.894 (0.003) | 1.781 (0.023) | 0.952 (0.002) | 3.209 (0.012) | 0.998 (0.000) | 3.996 (0.001) |
| Priority | 5000 | 0.910 (0.002) | 1.263 (0.026) | 0.913 (0.002) | 1.765 (0.027) | 0.898 (0.002) | 1.085 (0.005) | 0.934 (0.002) | 1.879 (0.028) | 0.960 (0.002) | 3.833 (0.004) |
| **Recommended** | | | | | | | | | | | |
| Recommended | 200 | 0.906 (0.005) | 2.899 (0.043) | 0.965 (0.003) | 3.820 (0.020) | 0.897 (0.005) | 2.607 (0.021) | 0.996 (0.002) | 3.970 (0.014) | 0.998 (0.001) | 3.980 (0.014) |
| Recommended | 500 | 0.914 (0.003) | 3.234 (0.042) | 0.945 (0.003) | 3.582 (0.014) | 0.891 (0.003) | 2.403 (0.013) | 0.975 (0.002) | 3.754 (0.011) | 1.000 (0.000) | 4.000 (0.000) |
| Recommended | 1000 | 0.911 (0.003) | 3.174 (0.025) | 0.925 (0.003) | 3.302 (0.011) | 0.890 (0.003) | 2.224 (0.011) | 0.963 (0.002) | 3.534 (0.010) | 1.000 (0.000) | 4.000 (0.000) |
| Recommended | 2000 | 0.920 (0.002) | 2.969 (0.020) | 0.924 (0.002) | 3.106 (0.016) | 0.897 (0.002) | 2.024 (0.015) | 0.954 (0.002) | 3.316 (0.015) | 0.999 (0.000) | 3.998 (0.000) |
| Recommended | 5000 | 0.916 (0.002) | 1.472 (0.025) | 0.916 (0.002) | 1.969 (0.025) | 0.907 (0.002) | 1.325 (0.010) | 0.943 (0.002) | 2.164 (0.025) | 0.968 (0.001) | 3.845 (0.003) |

### A7.1.3   COMPAS data

We extend our experiments to investigate the effectiveness of AFCP using the Correctional Offender Management Profiling for Alternative Sanctions (COMPAS) dataset [50]. The COMPAS dataset is widely studied in the fairness literature for multi-class classification tasks, predicting the risk of recidivism across three categories: 'High', 'Medium', and 'Low'. Following the data preprocessing steps outlined in [59], we exclude rows with missing or low-quality data and compute the length of stay in jail. Additionally, we merge the race groups Asian and Native American, both of which have few occurrences, into the 'Others' category. After preprocessing, the dataset comprises 6,172 instances with five categorical features: charge degree of defendants (2 levels), race (4 levels), age category (3 levels), sex (2 levels), and score category of defendants (3 levels). We regard the first four features as potentially sensitive attributes.

Similar to the Nursery dataset case, we utilize the LabelEncoder function to numerically encode categorical features and outcome labels. To increase prediction difficulty and emphasize algorithmic bias, we introduce independent, uniformly distributed noise to the labels of samples identified as African-American. Additionally, we undersample the African-American group to 200 samples, while the Caucasian group contains 2,103 samples, Hispanic 509, and Others 385.

Figure A19 summarizes the performance of all methods as a function of the total number of training and calibration data points, which range from 200 to 1000. Figure A20 narrows the focus by analyzing the performance relative to the number of restricted calibration data points, as defined in Section 2.2. The results are averaged over 500 randomly selected test points and 100 repeated experiments. In each experiment, 50% of the samples are randomly assigned for training and the remaining 50% for calibration. Once again, we conclude that our AFCP methods outperform the other benchmarks considered, resulting in more informative prediction sets with higher conditional coverage for the African-American subgroup.

Figure A21 shows the selection frequencies of the protected attribute Race as a function of sample size within the same experiment described in Figure A19. This plot demonstrates that both AFCP and AFCP 1 consistently select Race as the most sensitive attribute as the number of samples increases. Also, we report the prediction accuracy in Table A21. The results confirm that the African-American group is disproportionately affected by algorithmic bias, not only in terms of uncertainty estimates but also in prediction accuracy, as they experience significantly lower-than-average test accuracy.

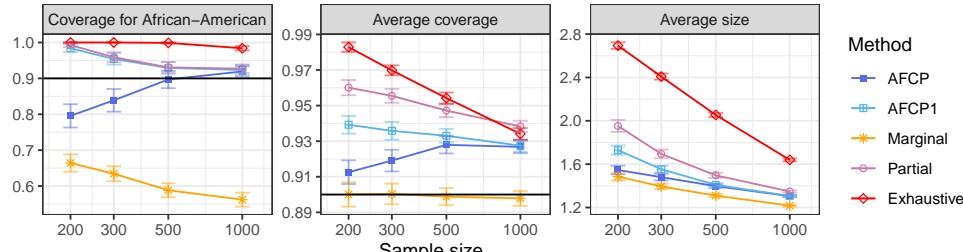

Figure A19: Performance of prediction sets constructed by different methods on the COMPAS data, as a function of the sample size. AFCP leads to more informative predictions with higher coverage conditional on the sensitive attribute, Race (shown for level African-American).

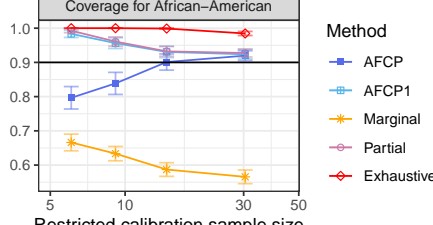

Figure A20: Conditional coverage for the African-American group using different methods on the COMPAS data, as a function of the sample sizes in the restricted calibration data.

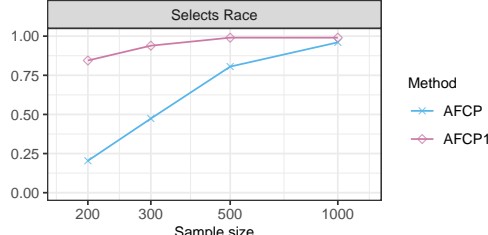

Figure A21: Selection frequency of the relevant attribute Race, using our AFCP method and its variation, AFCP1, in the COMPAS experiments of Figure A19. As the sample size increases, AFCP and AFCP1 become more consistent in selecting the most relevant attribute, Race.

Table A21: Average prediction accuracy and prediction accuracy for African-American group on the COMPAS data, as a function of the sample size.

| Sample Size | Accuracy for African-American | Average Accuracy |
|---|---|---|
| 200 | 0.355 (0.009) | 0.790 (0.003) |
| 300 | 0.370 (0.009) | 0.813 (0.002) |
| 500 | 0.376 (0.008) | 0.839 (0.002) |
| 1000 | 0.375 (0.009) | 0.869 (0.002) |

## A7.2    AFCP for Outlier Detection

### A7.2.1    Setup and Benchmarks

This section demonstrates the empirical performance of our AFCP extension for outlier detection. Firstly, we focus on the implementation described in Algorithm A7, which selects at most one sensitive attribute. Similar to the multi-class classification cases, Our method is compared with three existing approaches, which utilize the same data, ML model, and conformity scores but compute conformal p-values with different guarantees. The first is the *marginal* benchmark, which constructs conformal p-value guaranteeing $\mathbb{P}(\hat{u}^{\text{marginal}}(Z_{n+1}) \leq \alpha) \leq \alpha$ by applying Algorithm A2 without protected attributes. The second is the *exhaustive* equalized benchmark, which evaluates conformal p-values guaranteeing $\mathbb{P}(\hat{u}^{\text{exhaustive}}(Z_{n+1}) \leq \alpha \mid \phi(Z_{n+1}, [K])) \leq \alpha$ by applying Algorithm A2 with all $K$ sensitive attributes simultaneously protected. The third is a *partial* equalized benchmark that separately applies Algorithm A2 with each possible protected attribute $k \in [K]$, and then takes the maximum of all such p values. This is an intuitive approach that can be easily verified to provide a coverage guarantee $\mathbb{P}(\hat{u}^{\text{partial}}(Z_{n+1}) \leq \alpha \mid \phi(Z_{n+1}, \{k\})) \leq \alpha \quad \forall k \in [K]$.

In addition, similar to the multiclass classification experiments, we consider AFCP1 - the AFCP implementation that always selects the most critical protected attribute without conducting the significance test.

For all methods considered, the outlier detection model is based on a three-layer neural network, and the outputs from each layer are batch-normalized. The Adam optimizer and the BCEWithLogitsLoss loss function are used in the training process, with a learning rate set at 0.0001. The loss values demonstrate convergence after 100 epochs of training. For all methods, the miscoverage target level is set at $\alpha = 0.1$. Note that the training and testing data contain both inliers and outliers, while the calibration data only contains inliers.

We assess the performance of the methods based on the False Positive Rate (FPR) and the True Positive Rate (TPR). Ideally, the objective is to achieve a higher conditional FPR for groups experiencing unfairness, thereby maintaining the average FPR below the target threshold of 0.1 while simultaneously achieving a higher TPR. The results presented are averaged over 500 independent test points and 30 experiments.

### A7.2.2    Synthetic Data

We employ the same data settings as in the multi-class classification example, designating Color as the sensitive attribute associated with the Blue group, which suffers from undercoverage. While Age Group and Region are also sensitive attributes, they are not subject to biases in this context. The outcome labels $Y$ have two possible values: $Y = 0$ if the unit is properly treated and $Y = 1$

otherwise. In our experiment, we treat $Y = 1$ as an outlier and $Y = 0$ as an inlier, with the data generation process described as:

$$\mathbb{P}[Y \mid X] = \begin{cases} \left(\frac{1}{2}, \frac{1}{2}\right), & \text{if Color = Blue,} \\ (1, 0) \text{ or } (0, 1) \text{ w. equal prob}, & \text{if Color = Grey.} \end{cases} \tag{A28}$$

Figure A22 and Table A22 illustrate the performance of conformal p-values generated by various methods on synthetic data, showcasing how performance varies with the number of samples in training and calibration datasets. Figures A23–A24 and Tables A23–A25 separately evaluate the False Positive Rate (FPR) and True Positive Rate (TPR) for each protected attribute. Additionally, Figure A26 explores how the severity of bias affecting groups impacts the selection frequency of our method at various levels. The severity of bias is controlled by the varying percentage of Blue samples, with higher levels indicating less samples and more significant biases in the Blue group.

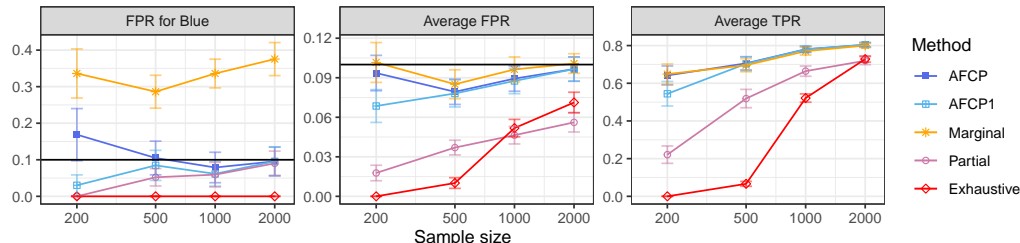

Figure A22: Performance on conformal p-values constructed by different methods on the synthetic data as a function of the total number of training and calibration points. Our method (AFCP) leads to higher TPR and lower FPR on the Blue group. The error bars indicate 2 standard errors.

Table A22: Average FPR and TPR of conformal p-values for all test samples constructed by different methods as a function of the training and calibration size. All methods control FPR under 0.1, while our AFCP and AFCP1 methods, along with the Marginal method, produce the highest TPR. Red numbers indicate high TPR. See corresponding plots in Figure A22.

| Sample Size | AFCP | | AFCP1 | | Marginal | | Partial | | Exhaustive | |
|---|---|---|---|---|---|---|---|---|---|---|
| | FPR | TPR | FPR | TPR | FPR | TPR | FPR | TPR | FPR | TPR |
| 200 | 0.094 (0.007) | 0.642 (0.025) | 0.069 (0.006) | 0.544 (0.033) | 0.102 (0.008) | 0.649 (0.027) | 0.018 (0.003) | 0.221 (0.023) | 0.000 (0.000) | 0.000 (0.000) |
| 500 | 0.079 (0.005) | 0.706 (0.018) | 0.078 (0.005) | 0.703 (0.018) | 0.085 (0.005) | 0.696 (0.017) | 0.037 (0.003) | 0.519 (0.024) | 0.010 (0.002) | 0.066 (0.006) |
| 1000 | 0.089 (0.005) | 0.780 (0.008) | 0.088 (0.005) | 0.780 (0.008) | 0.096 (0.005) | 0.770 (0.010) | 0.046 (0.003) | 0.664 (0.014) | 0.052 (0.003) | 0.521 (0.011) |
| 2000 | 0.097 (0.005) | 0.804 (0.006) | 0.097 (0.005) | 0.804 (0.006) | 0.101 (0.004) | 0.800 (0.006) | 0.056 (0.004) | 0.720 (0.010) | 0.071 (0.004) | 0.728 (0.008) |

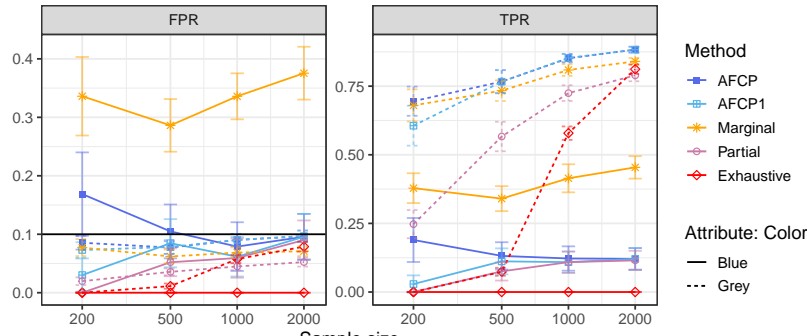

Figure A23: Performance on conformal p-values constructed by different methods for groups formed by Color. See Table A23 for numerical details and standard errors.

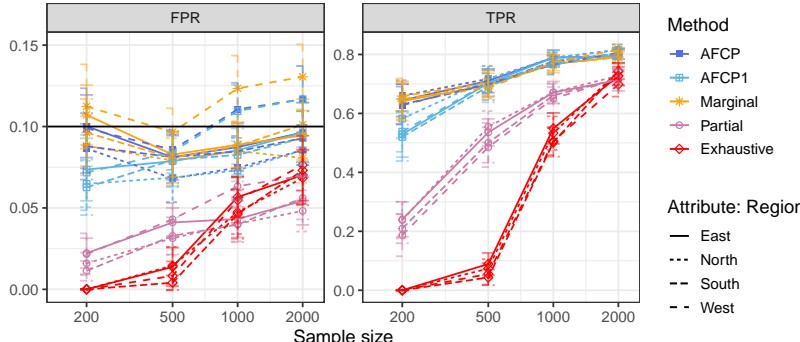

Figure A24: Performance on conformal p-values constructed by different methods for groups formed by Region. See Table A25 for numerical details and standard errors.

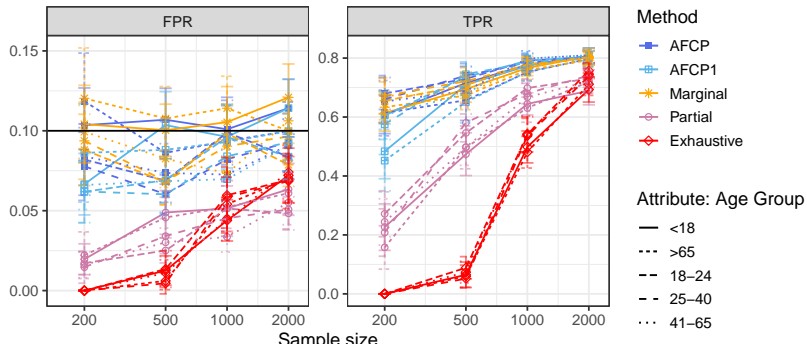

Figure A25: Performance on conformal p-values constructed by different methods for groups formed by Age Group. See Table A24 for numerical details and standard errors.

Table A23: Coverage and size of prediction sets constructed with different methods for groups formed by Color. See corresponding plots in Figure A23.

| Group | Sample Size | AFCP | | AFCP1 | | Marginal | | Partial | | Exhaustive | |
|---|---|---|---|---|---|---|---|---|---|---|---|
| | | FPR | TPR | FPR | TPR | FPR | TPR | FPR | TPR | FPR | TPR |
| **Grey** | | | | | | | | | | | |
| Grey | 200 | 0.085 (0.006) | 0.695 (0.027) | 0.073 (0.007) | 0.606 (0.036) | 0.077 (0.007) | 0.680 (0.029) | 0.020 (0.003) | 0.247 (0.026) | 0.000 (0.000) | 0.000 (0.000) |
| Grey | 500 | 0.077 (0.005) | 0.767 (0.021) | 0.078 (0.006) | 0.766 (0.021) | 0.062 (0.006) | 0.734 (0.019) | 0.036 (0.003) | 0.567 (0.027) | 0.011 (0.002) | 0.074 (0.007) |
| Grey | 1000 | 0.090 (0.005) | 0.852 (0.008) | 0.090 (0.005) | 0.853 (0.008) | 0.069 (0.004) | 0.809 (0.011) | 0.045 (0.003) | 0.725 (0.014) | 0.058 (0.004) | 0.579 (0.012) |
| Grey | 2000 | 0.097 (0.005) | 0.883 (0.005) | 0.097 (0.005) | 0.883 (0.005) | 0.071 (0.004) | 0.840 (0.006) | 0.052 (0.004) | 0.789 (0.011) | 0.079 (0.004) | 0.812 (0.009) |
| **Blue** | | | | | | | | | | | |
| Blue | 200 | 0.169 (0.036) | 0.190 (0.040) | 0.030 (0.014) | 0.029 (0.016) | 0.336 (0.034) | 0.378 (0.027) | 0.000 (0.000) | 0.000 (0.000) | 0.000 (0.000) | 0.000 (0.000) |
| Blue | 500 | 0.105 (0.023) | 0.132 (0.025) | 0.085 (0.021) | 0.112 (0.024) | 0.286 (0.023) | 0.340 (0.023) | 0.052 (0.012) | 0.075 (0.017) | 0.000 (0.000) | 0.000 (0.000) |
| Blue | 1000 | 0.079 (0.021) | 0.122 (0.022) | 0.062 (0.017) | 0.109 (0.019) | 0.336 (0.020) | 0.414 (0.026) | 0.059 (0.017) | 0.109 (0.019) | 0.000 (0.000) | 0.000 (0.000) |
| Blue | 2000 | 0.096 (0.020) | 0.121 (0.020) | 0.096 (0.020) | 0.121 (0.020) | 0.375 (0.023) | 0.454 (0.020) | 0.090 (0.017) | 0.115 (0.018) | 0.000 (0.000) | 0.000 (0.000) |

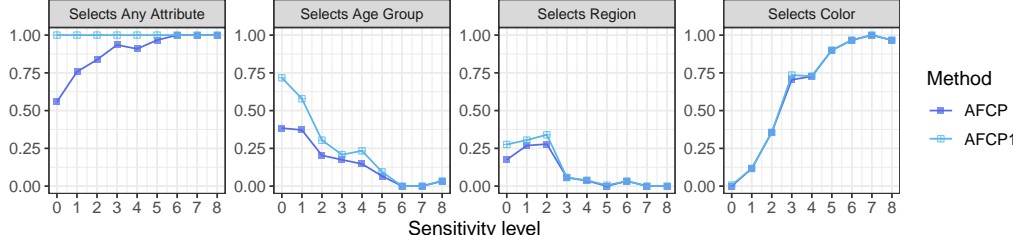

Figure A26: Selection frequencies of each protected attribute as a function of severity level of bias.

Table A24: Coverage and size of prediction sets constructed with different methods for groups formed by Age Group. See corresponding plots in Figure A25.

| Group | Sample Size | AFCP FPR | AFCP TPR | AFCP1 FPR | AFCP1 TPR | Marginal FPR | Marginal TPR | Partial FPR | Partial TPR | Exhaustive FPR | Exhaustive TPR |
|---|---|---|---|---|---|---|---|---|---|---|---|
| **West** | | | | | | | | | | | |
| West | 200 | 0.099 (0.012) | 0.628 (0.032) | 0.072 (0.012) | 0.538 (0.034) | 0.112 (0.013) | 0.649 (0.035) | 0.022 (0.006) | 0.187 (0.036) | 0.000 (0.000) | 0.000 (0.000) |
| West | 500 | 0.086 (0.006) | 0.701 (0.017) | 0.084 (0.007) | 0.705 (0.017) | 0.097 (0.007) | 0.687 (0.015) | 0.043 (0.005) | 0.485 (0.034) | 0.008 (0.004) | 0.055 (0.018) |
| West | 1000 | 0.111 (0.008) | 0.766 (0.017) | 0.109 (0.008) | 0.765 (0.016) | 0.123 (0.010) | 0.773 (0.016) | 0.063 (0.007) | 0.650 (0.025) | 0.055 (0.007) | 0.506 (0.024) |
| West | 2000 | 0.117 (0.010) | 0.805 (0.009) | 0.117 (0.010) | 0.805 (0.009) | 0.131 (0.010) | 0.817 (0.009) | 0.070 (0.009) | 0.718 (0.016) | 0.076 (0.008) | 0.698 (0.021) |
| **East** | | | | | | | | | | | |
| East | 200 | 0.100 (0.010) | 0.644 (0.028) | 0.074 (0.010) | 0.529 (0.039) | 0.107 (0.009) | 0.642 (0.033) | 0.022 (0.005) | 0.241 (0.030) | 0.000 (0.000) | 0.000 (0.000) |
| East | 500 | 0.081 (0.007) | 0.711 (0.020) | 0.079 (0.008) | 0.703 (0.022) | 0.083 (0.007) | 0.696 (0.020) | 0.041 (0.004) | 0.535 (0.023) | 0.014 (0.006) | 0.089 (0.019) |
| East | 1000 | 0.087 (0.007) | 0.790 (0.013) | 0.085 (0.008) | 0.789 (0.012) | 0.089 (0.007) | 0.768 (0.015) | 0.043 (0.006) | 0.671 (0.019) | 0.057 (0.006) | 0.549 (0.026) |
| East | 2000 | 0.096 (0.009) | 0.794 (0.012) | 0.096 (0.009) | 0.794 (0.012) | 0.095 (0.008) | 0.791 (0.011) | 0.054 (0.007) | 0.713 (0.018) | 0.070 (0.008) | 0.727 (0.016) |
| **North** | | | | | | | | | | | |
| North | 200 | 0.087 (0.008) | 0.660 (0.026) | 0.065 (0.007) | 0.582 (0.034) | 0.088 (0.009) | 0.646 (0.028) | 0.016 (0.003) | 0.238 (0.031) | 0.000 (0.000) | 0.000 (0.000) |
| North | 500 | 0.068 (0.007) | 0.717 (0.022) | 0.069 (0.007) | 0.716 (0.024) | 0.079 (0.007) | 0.698 (0.024) | 0.031 (0.005) | 0.555 (0.027) | 0.015 (0.006) | 0.074 (0.016) |
| North | 1000 | 0.075 (0.009) | 0.789 (0.009) | 0.073 (0.009) | 0.789 (0.009) | 0.085 (0.008) | 0.773 (0.013) | 0.040 (0.005) | 0.672 (0.015) | 0.047 (0.007) | 0.533 (0.028) |
| North | 2000 | 0.085 (0.008) | 0.816 (0.009) | 0.085 (0.008) | 0.816 (0.009) | 0.081 (0.008) | 0.802 (0.010) | 0.048 (0.006) | 0.728 (0.013) | 0.068 (0.008) | 0.743 (0.015) |
| **South** | | | | | | | | | | | |
| South | 200 | 0.088 (0.010) | 0.630 (0.034) | 0.063 (0.009) | 0.520 (0.041) | 0.096 (0.010) | 0.658 (0.025) | 0.011 (0.003) | 0.209 (0.025) | 0.000 (0.000) | 0.000 (0.000) |
| South | 500 | 0.081 (0.008) | 0.697 (0.026) | 0.079 (0.008) | 0.691 (0.027) | 0.081 (0.008) | 0.704 (0.022) | 0.033 (0.006) | 0.500 (0.034) | 0.004 (0.002) | 0.043 (0.013) |
| South | 1000 | 0.085 (0.008) | 0.777 (0.010) | 0.083 (0.008) | 0.777 (0.009) | 0.088 (0.007) | 0.768 (0.011) | 0.040 (0.005) | 0.661 (0.014) | 0.046 (0.008) | 0.502 (0.024) |
| South | 2000 | 0.093 (0.008) | 0.803 (0.009) | 0.093 (0.008) | 0.803 (0.009) | 0.101 (0.008) | 0.790 (0.009) | 0.056 (0.007) | 0.718 (0.012) | 0.073 (0.011) | 0.744 (0.014) |

Table A25: Coverage and size of prediction sets constructed with different methods for groups formed by Region. See corresponding plots in Figure A24.

| Group | Sample Size | AFCP FPR | AFCP TPR | AFCP1 FPR | AFCP1 TPR | Marginal FPR | Marginal TPR | Partial FPR | Partial TPR | Exhaustive FPR | Exhaustive TPR |
|---|---|---|---|---|---|---|---|---|---|---|---|
| **<18** | | | | | | | | | | | |
| <18 | 200 | 0.104 (0.012) | 0.608 (0.032) | 0.067 (0.010) | 0.484 (0.047) | 0.104 (0.012) | 0.613 (0.031) | 0.020 (0.005) | 0.227 (0.039) | 0.000 (0.000) | 0.000 (0.000) |
| <18 | 500 | 0.107 (0.010) | 0.715 (0.017) | 0.104 (0.010) | 0.710 (0.018) | 0.100 (0.008) | 0.695 (0.022) | 0.049 (0.007) | 0.474 (0.036) | 0.013 (0.005) | 0.065 (0.022) |
| <18 | 1000 | 0.101 (0.010) | 0.777 (0.010) | 0.096 (0.010) | 0.774 (0.010) | 0.105 (0.011) | 0.767 (0.013) | 0.051 (0.009) | 0.645 (0.019) | 0.044 (0.006) | 0.499 (0.027) |
| <18 | 2000 | 0.114 (0.009) | 0.806 (0.014) | 0.114 (0.009) | 0.806 (0.014) | 0.121 (0.010) | 0.806 (0.014) | 0.063 (0.008) | 0.689 (0.024) | 0.072 (0.009) | 0.693 (0.021) |
| **18-24** | | | | | | | | | | | |
| 18-24 | 200 | 0.082 (0.011) | 0.680 (0.027) | 0.062 (0.010) | 0.605 (0.041) | 0.088 (0.011) | 0.655 (0.032) | 0.017 (0.004) | 0.272 (0.039) | 0.000 (0.000) | 0.000 (0.000) |
| 18-24 | 500 | 0.069 (0.007) | 0.730 (0.021) | 0.069 (0.007) | 0.736 (0.018) | 0.068 (0.007) | 0.708 (0.022) | 0.025 (0.004) | 0.546 (0.032) | 0.013 (0.006) | 0.089 (0.019) |
| 18-24 | 1000 | 0.099 (0.010) | 0.791 (0.012) | 0.096 (0.010) | 0.791 (0.012) | 0.098 (0.009) | 0.779 (0.013) | 0.051 (0.007) | 0.698 (0.015) | 0.060 (0.012) | 0.541 (0.030) |
| 18-24 | 2000 | 0.084 (0.007) | 0.804 (0.011) | 0.084 (0.007) | 0.804 (0.011) | 0.079 (0.007) | 0.799 (0.012) | 0.048 (0.005) | 0.731 (0.015) | 0.069 (0.007) | 0.757 (0.013) |
| **25-40** | | | | | | | | | | | |
| 25-40 | 200 | 0.078 (0.011) | 0.660 (0.040) | 0.062 (0.010) | 0.575 (0.051) | 0.094 (0.012) | 0.672 (0.031) | 0.015 (0.005) | 0.208 (0.040) | 0.000 (0.000) | 0.000 (0.000) |
| 25-40 | 500 | 0.060 (0.006) | 0.741 (0.020) | 0.060 (0.007) | 0.745 (0.021) | 0.068 (0.007) | 0.722 (0.023) | 0.034 (0.005) | 0.581 (0.032) | 0.005 (0.003) | 0.052 (0.016) |
| 25-40 | 1000 | 0.082 (0.007) | 0.784 (0.013) | 0.084 (0.007) | 0.784 (0.013) | 0.090 (0.008) | 0.777 (0.013) | 0.045 (0.006) | 0.680 (0.017) | 0.058 (0.010) | 0.537 (0.030) |
| 25-40 | 2000 | 0.093 (0.009) | 0.804 (0.011) | 0.093 (0.009) | 0.804 (0.011) | 0.096 (0.007) | 0.803 (0.011) | 0.051 (0.006) | 0.741 (0.012) | 0.069 (0.009) | 0.738 (0.013) |
| **41-65** | | | | | | | | | | | |
| 41-65 | 200 | 0.084 (0.009) | 0.653 (0.036) | 0.066 (0.008) | 0.607 (0.041) | 0.099 (0.008) | 0.672 (0.031) | 0.016 (0.004) | 0.244 (0.039) | 0.000 (0.000) | 0.000 (0.000) |
| 41-65 | 500 | 0.074 (0.009) | 0.688 (0.025) | 0.068 (0.010) | 0.679 (0.025) | 0.083 (0.009) | 0.680 (0.023) | 0.030 (0.004) | 0.498 (0.031) | 0.011 (0.004) | 0.074 (0.018) |
| 41-65 | 1000 | 0.073 (0.008) | 0.798 (0.013) | 0.070 (0.007) | 0.798 (0.013) | 0.073 (0.008) | 0.771 (0.016) | 0.034 (0.005) | 0.667 (0.024) | 0.045 (0.007) | 0.542 (0.031) |
| 41-65 | 2000 | 0.091 (0.008) | 0.810 (0.010) | 0.091 (0.008) | 0.810 (0.010) | 0.106 (0.007) | 0.801 (0.013) | 0.055 (0.007) | 0.721 (0.016) | 0.074 (0.008) | 0.745 (0.018) |
| **>65** | | | | | | | | | | | |
| >65 | 200 | 0.118 (0.015) | 0.614 (0.039) | 0.086 (0.015) | 0.452 (0.053) | 0.120 (0.016) | 0.633 (0.026) | 0.022 (0.007) | 0.158 (0.037) | 0.000 (0.000) | 0.000 (0.000) |
| >65 | 500 | 0.086 (0.010) | 0.657 (0.034) | 0.088 (0.010) | 0.651 (0.035) | 0.108 (0.010) | 0.673 (0.024) | 0.046 (0.006) | 0.497 (0.036) | 0.006 (0.003) | 0.060 (0.019) |
| >65 | 1000 | 0.094 (0.011) | 0.752 (0.015) | 0.093 (0.011) | 0.754 (0.015) | 0.114 (0.010) | 0.759 (0.016) | 0.052 (0.007) | 0.631 (0.024) | 0.054 (0.008) | 0.481 (0.027) |
| >65 | 2000 | 0.100 (0.009) | 0.797 (0.013) | 0.100 (0.009) | 0.797 (0.013) | 0.100 (0.009) | 0.793 (0.012) | 0.061 (0.006) | 0.720 (0.020) | 0.070 (0.007) | 0.711 (0.019) |

### A7.2.3 Adult Income Data

We apply our method to the open-domain Adult Income dataset [51], a widely utilized resource in fairness studies. In this dataset, individuals with an income exceeding \$50,000 are treated as outliers, while those with an income of \$50,000 or less are considered inliers. All categorical variables in this dataset are treated as sensitive attributes, with levels that have small sample sizes grouped together during the pre-processing stages. After pre-processing, the sensitive attributes and their associated levels are as follows: Native-country (United-States, Others); Education (Bachelors, Some-college, HS-grad, Others); Work Class (Private, Non-private); Marital-status (Divorced, Married-civ-spouse, Never-married, Others); Occupation (Adm-clerical, Craft-repair, Other-service, Sales, Exec-managerial, Prof-specialty, Others); Relationship (Own-child, Husband, Not-in-family, Unmarried, Others); Race (White, Others); and Sex (female, male).

This section evaluates and compares the performance of AFCP, AFCP1, and AFCP+, which is the AFCP implementation capable of selecting multiple protected attributes, along with other benchmark methods described in Appendix A7.2.1. All results presented in this section average over 500 test points and 30 independent experiments.

For this dataset, the groups suffering from algorithmic biases are unknown. Figure A27 evaluates the performance of different methods on several groups that are observed to exhibit higher FPR using the Marginal method. In all such cases, our AFCP, AFPC1, and AFCP+ methods effectively identify and correct for the protected attributes corresponding to at least one group suffering from significantly higher FPR.

Figure A28 and Table A26 present the average FPR and TPR of conformal p-values computed using different methods across all test samples. On average, all methods can control the FPR below the target level of 0.1, while our AFCP methods generally achieve higher TPR.

In addition, Figures A29–A36 and Table A27–A34 separately assess the FPR and TPR of conformal p-values for each protected attribute.

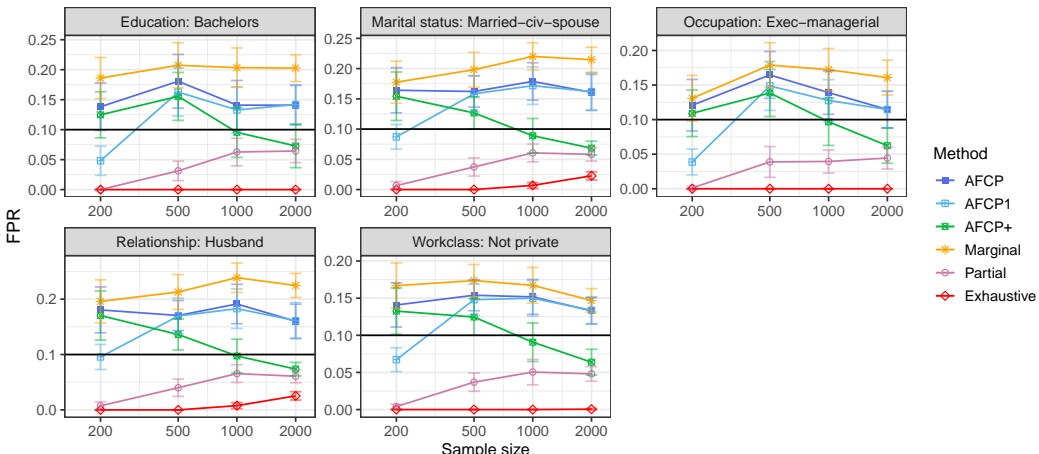

Figure A27: FPR of conformal p-values constructed with different methods for groups that are observed to have significantly higher FPR when using the Marginal method. On average, our methods identify those groups and perform corrections on the FPR.

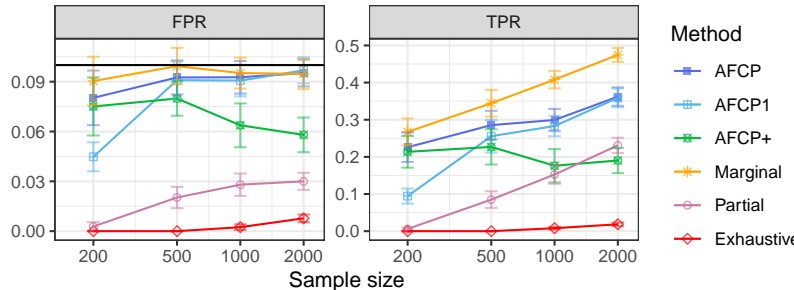

Figure A28: Average FPR and TPR of conformal p-values constructed with different methods on all test samples. All methods control FPR under the target level 0.1. Our AFCP methods can produce generally higher TPR than other methods except the Marginal one. However, as depicted in Figure A27, the Marginal method suffers from unfairness.

Table A26: Average FPR and TPR of conformal p-values constructed with different methods on all test samples. All methods control FPR under the target level 0.1. Red numbers indicate high TPR. See corresponding plots in Figure A28.

| Sample size | AFCP | | AFCP1 | | AFCP+ | | Marginal | | Partial | | Exhaustive | |
|---|---|---|---|---|---|---|---|---|---|---|---|---|
| | FPR | TPR | FPR | TPR | FPR | TPR | FPR | TPR | FPR | TPR | FPR | TPR |
| 1 | | | | | | | | | | | | |
| 200 | 0.080 (0.008) | 0.226 (0.020) | 0.045 (0.004) | 0.094 (0.010) | 0.075 (0.009) | 0.214 (0.021) | 0.090 (0.007) | 0.267 (0.018) | 0.003 (0.001) | 0.006 (0.002) | 0.000 (0.000) | 0.000 (0.000) |
| 1 | | | | | | | | | | | | |
| 500 | 0.093 (0.005) | 0.285 (0.019) | 0.091 (0.006) | 0.256 (0.022) | 0.080 (0.005) | 0.227 (0.024) | 0.099 (0.006) | 0.344 (0.018) | 0.020 (0.003) | 0.085 (0.011) | 0.000 (0.000) | 0.000 (0.000) |
| 1 | | | | | | | | | | | | |
| 1000 | 0.093 (0.005) | 0.299 (0.015) | 0.091 (0.005) | 0.283 (0.014) | 0.064 (0.007) | 0.176 (0.022) | 0.095 (0.005) | 0.408 (0.012) | 0.028 (0.003) | 0.153 (0.013) | 0.002 (0.001) | 0.008 (0.002) |
| 1 | | | | | | | | | | | | |
| 2000 | 0.095 (0.004) | 0.362 (0.012) | 0.097 (0.004) | 0.359 (0.013) | 0.058 (0.005) | 0.190 (0.017) | 0.095 (0.005) | 0.475 (0.009) | 0.030 (0.003) | 0.231 (0.010) | 0.008 (0.001) | 0.019 (0.002) |

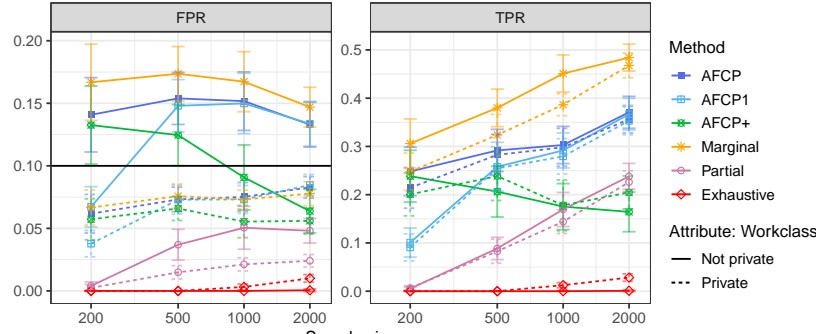

Figure A29: Performance on conformal p-values constructed by different methods for groups formed by Work Class. See Table A27 for numerical details and standard errors.

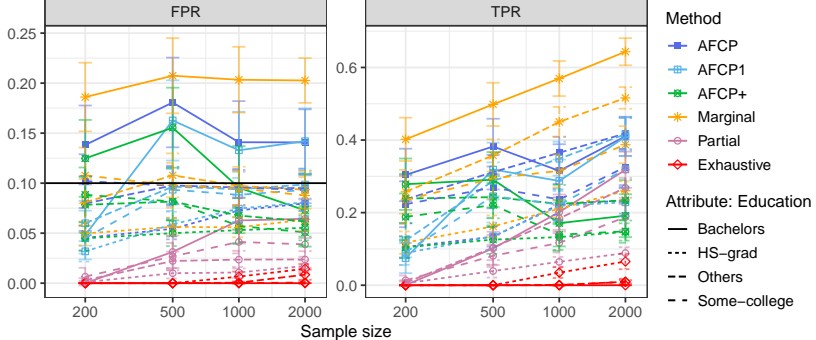

Figure A30: Performance on conformal p-values constructed by different methods for groups formed by Education. See Table A28 for numerical details and standard errors.

Table A27: Coverage and size of prediction sets constructed with different methods for groups formed by Work Class. See corresponding plots in Figure A29.

| Attribute: Workclass | Sample size | AFCP | | AFCP1 | | AFCP+ | | Marginal | | Partial | | Exhaustive | |
|---|---|---|---|---|---|---|---|---|---|---|---|---|---|
| | | FPR | TPR | FPR | TPR | FPR | TPR | FPR | TPR | FPR | TPR | FPR | TPR |
| **Not private** | | | | | | | | | | | | | |
| Not private | 200 | 0.141 (0.015) | 0.248 (0.025) | 0.067 (0.008) | 0.101 (0.015) | 0.133 (0.016) | 0.238 (0.027) | 0.167 (0.015) | 0.306 (0.026) | 0.004 (0.002) | 0.005 (0.002) | 0.000 (0.000) | 0.000 (0.000) |
| Not private | 500 | 0.154 (0.010) | 0.292 (0.022) | 0.148 (0.010) | 0.258 (0.025) | 0.125 (0.012) | 0.206 (0.026) | 0.174 (0.011) | 0.380 (0.019) | 0.037 (0.006) | 0.089 (0.012) | 0.000 (0.000) | 0.000 (0.000) |
| Not private | 1000 | 0.152 (0.012) | 0.303 (0.019) | 0.150 (0.012) | 0.292 (0.018) | 0.091 (0.013) | 0.175 (0.024) | 0.167 (0.012) | 0.451 (0.019) | 0.051 (0.009) | 0.169 (0.018) | 0.000 (0.000) | 0.000 (0.000) |
| Not private | 2000 | 0.133 (0.009) | 0.371 (0.017) | 0.134 (0.009) | 0.367 (0.016) | 0.064 (0.009) | 0.164 (0.021) | 0.147 (0.008) | 0.484 (0.014) | 0.048 (0.005) | 0.238 (0.013) | 0.001 (0.001) | 0.001 (0.001) |
| **Private** | | | | | | | | | | | | | |
| Private | 200 | 0.062 (0.008) | 0.214 (0.021) | 0.038 (0.005) | 0.091 (0.014) | 0.057 (0.008) | 0.201 (0.022) | 0.067 (0.007) | 0.248 (0.019) | 0.002 (0.001) | 0.006 (0.002) | 0.000 (0.000) | 0.000 (0.000) |
| Private | 500 | 0.073 (0.005) | 0.283 (0.021) | 0.073 (0.006) | 0.254 (0.022) | 0.066 (0.005) | 0.239 (0.025) | 0.076 (0.005) | 0.324 (0.021) | 0.015 (0.003) | 0.083 (0.012) | 0.000 (0.000) | 0.000 (0.000) |
| Private | 1000 | 0.075 (0.006) | 0.298 (0.020) | 0.073 (0.006) | 0.279 (0.018) | 0.055 (0.006) | 0.178 (0.026) | 0.073 (0.004) | 0.386 (0.011) | 0.021 (0.003) | 0.144 (0.012) | 0.003 (0.001) | 0.012 (0.003) |
| Private | 2000 | 0.083 (0.004) | 0.356 (0.014) | 0.085 (0.004) | 0.353 (0.015) | 0.056 (0.005) | 0.205 (0.017) | 0.078 (0.005) | 0.468 (0.013) | 0.024 (0.003) | 0.226 (0.012) | 0.010 (0.002) | 0.028 (0.004) |

Table A28: Coverage and size of prediction sets constructed with different methods for groups formed by Education. See corresponding plots in Figure A30.

| Attribute: Education | Sample size | AFCP | | AFCP1 | | AFCP+ | | Marginal | | Partial | | Exhaustive | |
|---|---|---|---|---|---|---|---|---|---|---|---|---|---|
| | | FPR | TPR | FPR | TPR | FPR | TPR | FPR | TPR | FPR | TPR | FPR | TPR |
| **Bachelors** | | | | | | | | | | | | | |
| Bachelors | 200 | 0.138 (0.020) | 0.304 (0.036) | 0.048 (0.012) | 0.075 (0.021) | 0.125 (0.019) | 0.278 (0.035) | 0.186 (0.017) | 0.402 (0.030) | 0.000 (0.000) | 0.000 (0.000) | 0.000 (0.000) | 0.000 (0.000) |
| Bachelors | 500 | 0.181 (0.022) | 0.382 (0.038) | 0.163 (0.020) | 0.319 (0.037) | 0.155 (0.020) | 0.291 (0.038) | 0.207 (0.019) | 0.498 (0.030) | 0.031 (0.008) | 0.102 (0.023) | 0.000 (0.000) | 0.000 (0.000) |
| Bachelors | 1000 | 0.141 (0.021) | 0.315 (0.037) | 0.133 (0.019) | 0.287 (0.029) | 0.095 (0.021) | 0.171 (0.032) | 0.203 (0.016) | 0.570 (0.024) | 0.063 (0.012) | 0.201 (0.024) | 0.000 (0.000) | 0.000 (0.000) |
| Bachelors | 2000 | 0.141 (0.017) | 0.410 (0.027) | 0.142 (0.016) | 0.409 (0.027) | 0.073 (0.018) | 0.192 (0.029) | 0.203 (0.011) | 0.643 (0.019) | 0.064 (0.010) | 0.318 (0.025) | 0.000 (0.000) | 0.000 (0.000) |
| **HS-grad** | | | | | | | | | | | | | |
| HS-grad | 200 | 0.045 (0.006) | 0.105 (0.015) | 0.032 (0.005) | 0.089 (0.016) | 0.045 (0.006) | 0.105 (0.015) | 0.050 (0.006) | 0.117 (0.015) | 0.001 (0.001) | 0.005 (0.003) | 0.000 (0.000) | 0.000 (0.000) |
| HS-grad | 500 | 0.054 (0.005) | 0.134 (0.013) | 0.059 (0.006) | 0.135 (0.019) | 0.050 (0.006) | 0.126 (0.020) | 0.056 (0.005) | 0.162 (0.015) | 0.010 (0.003) | 0.039 (0.009) | 0.000 (0.000) | 0.000 (0.000) |
| HS-grad | 1000 | 0.073 (0.006) | 0.223 (0.021) | 0.075 (0.007) | 0.219 (0.020) | 0.053 (0.007) | 0.133 (0.018) | 0.056 (0.005) | 0.211 (0.012) | 0.011 (0.003) | 0.064 (0.007) | 0.007 (0.003) | 0.035 (0.007) |
| HS-grad | 2000 | 0.080 (0.008) | 0.234 (0.018) | 0.081 (0.008) | 0.228 (0.019) | 0.055 (0.006) | 0.147 (0.012) | 0.064 (0.006) | 0.260 (0.013) | 0.017 (0.003) | 0.089 (0.008) | 0.014 (0.002) | 0.065 (0.011) |
| **Others** | | | | | | | | | | | | | |
| Others | 200 | 0.080 (0.014) | 0.237 (0.033) | 0.060 (0.011) | 0.123 (0.023) | 0.078 (0.014) | 0.239 (0.034) | 0.081 (0.012) | 0.257 (0.030) | 0.003 (0.001) | 0.007 (0.003) | 0.000 (0.000) | 0.000 (0.000) |
| Others | 500 | 0.098 (0.009) | 0.312 (0.024) | 0.097 (0.009) | 0.287 (0.026) | 0.082 (0.009) | 0.243 (0.029) | 0.108 (0.011) | 0.358 (0.022) | 0.022 (0.005) | 0.104 (0.017) | 0.000 (0.000) | 0.000 (0.000) |
| Others | 1000 | 0.096 (0.009) | 0.365 (0.022) | 0.094 (0.009) | 0.349 (0.023) | 0.068 (0.009) | 0.224 (0.026) | 0.097 (0.010) | 0.450 (0.021) | 0.024 (0.004) | 0.184 (0.018) | 0.001 (0.001) | 0.001 (0.001) |
| Others | 2000 | 0.095 (0.008) | 0.418 (0.022) | 0.099 (0.007) | 0.416 (0.022) | 0.062 (0.008) | 0.234 (0.028) | 0.093 (0.009) | 0.516 (0.015) | 0.024 (0.004) | 0.268 (0.014) | 0.008 (0.003) | 0.009 (0.002) |
| **Some-college** | | | | | | | | | | | | | |
| Some-college | 200 | 0.102 (0.013) | 0.223 (0.032) | 0.045 (0.008) | 0.091 (0.018) | 0.088 (0.014) | 0.188 (0.032) | 0.108 (0.014) | 0.235 (0.033) | 0.007 (0.004) | 0.014 (0.007) | 0.000 (0.000) | 0.000 (0.000) |
| Some-college | 500 | 0.097 (0.010) | 0.269 (0.029) | 0.093 (0.011) | 0.245 (0.030) | 0.082 (0.009) | 0.221 (0.030) | 0.098 (0.011) | 0.294 (0.027) | 0.027 (0.007) | 0.081 (0.015) | 0.000 (0.000) | 0.000 (0.000) |
| Some-college | 1000 | 0.095 (0.009) | 0.236 (0.022) | 0.088 (0.008) | 0.222 (0.022) | 0.057 (0.009) | 0.139 (0.022) | 0.095 (0.009) | 0.315 (0.023) | 0.041 (0.006) | 0.117 (0.013) | 0.000 (0.000) | 0.000 (0.000) |
| Some-college | 2000 | 0.093 (0.011) | 0.326 (0.025) | 0.093 (0.011) | 0.319 (0.025) | 0.051 (0.006) | 0.149 (0.016) | 0.088 (0.009) | 0.387 (0.026) | 0.039 (0.006) | 0.188 (0.018) | 0.001 (0.001) | 0.009 (0.004) |

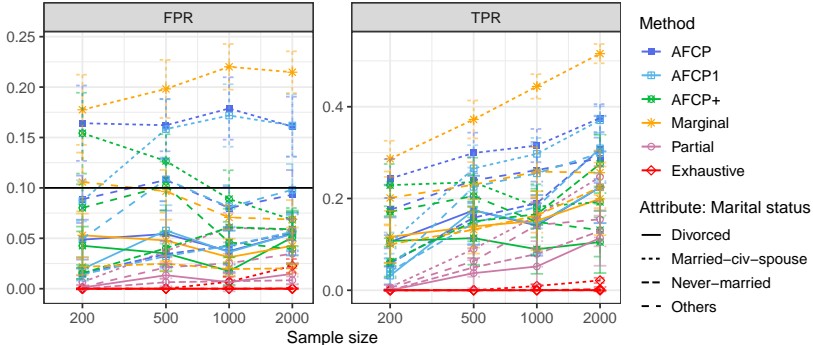

Figure A31: Performance on conformal p-values constructed by different methods for groups formed by Marital Status. See Table A29 for numerical details and standard errors.

Table A29: Coverage and size of prediction sets constructed with different methods for groups formed by Marital Status. See corresponding plots in Figure A31.

| Attribute: Marital status | Sample size | AFCP | | AFCP1 | | AFCP+ | | Marginal | | Partial | | Exhaustive | |
|---|---|---|---|---|---|---|---|---|---|---|---|---|---|
| | | FPR | TPR | FPR | TPR | FPR | TPR | FPR | TPR | FPR | TPR | FPR | TPR |
| **Divorced** | | | | | | | | | | | | | |
| Divorced | 200 | 0.049 (0.010) | 0.105 (0.027) | 0.020 (0.006) | 0.032 (0.014) | 0.043 (0.010) | 0.107 (0.027) | 0.053 (0.011) | 0.116 (0.028) | 0.001 (0.001) | 0.000 (0.000) | 0.000 (0.000) | 0.000 (0.000) |
| Divorced | 500 | 0.054 (0.012) | 0.174 (0.038) | 0.058 (0.013) | 0.174 (0.038) | 0.035 (0.009) | 0.114 (0.029) | 0.048 (0.010) | 0.139 (0.030) | 0.013 (0.006) | 0.037 (0.016) | 0.000 (0.000) | 0.000 (0.000) |
| Divorced | 1000 | 0.037 (0.010) | 0.141 (0.033) | 0.036 (0.011) | 0.144 (0.032) | 0.017 (0.007) | 0.089 (0.027) | 0.031 (0.009) | 0.151 (0.032) | 0.006 (0.005) | 0.052 (0.020) | 0.000 (0.000) | 0.000 (0.000) |
| Divorced | 2000 | 0.054 (0.011) | 0.224 (0.039) | 0.054 (0.010) | 0.224 (0.039) | 0.051 (0.013) | 0.105 (0.034) | 0.043 (0.009) | 0.199 (0.040) | 0.015 (0.005) | 0.114 (0.030) | 0.000 (0.000) | 0.000 (0.000) |
| **Married-civ-spouse** | | | | | | | | | | | | | |
| Married-civ-spouse | 200 | 0.164 (0.019) | 0.243 (0.021) | 0.087 (0.010) | 0.103 (0.012) | 0.154 (0.020) | 0.229 (0.023) | 0.178 (0.017) | 0.286 (0.020) | 0.007 (0.003) | 0.007 (0.002) | 0.000 (0.000) | 0.000 (0.000) |
| Married-civ-spouse | 500 | 0.162 (0.013) | 0.299 (0.022) | 0.158 (0.015) | 0.266 (0.025) | 0.127 (0.013) | 0.236 (0.027) | 0.198 (0.014) | 0.372 (0.021) | 0.037 (0.008) | 0.090 (0.012) | 0.000 (0.000) | 0.000 (0.000) |
| Married-civ-spouse | 1000 | 0.179 (0.015) | 0.315 (0.018) | 0.172 (0.015) | 0.297 (0.017) | 0.089 (0.014) | 0.182 (0.024) | 0.220 (0.011) | 0.444 (0.013) | 0.061 (0.007) | 0.162 (0.014) | 0.007 (0.002) | 0.009 (0.002) |
| Married-civ-spouse | 2000 | 0.161 (0.015) | 0.375 (0.015) | 0.162 (0.016) | 0.371 (0.015) | 0.068 (0.006) | 0.192 (0.017) | 0.215 (0.010) | 0.516 (0.011) | 0.058 (0.006) | 0.247 (0.012) | 0.023 (0.003) | 0.021 (0.003) |
| **Never-married** | | | | | | | | | | | | | |
| Never-married | 200 | 0.015 (0.003) | 0.059 (0.016) | 0.014 (0.003) | 0.041 (0.014) | 0.015 (0.003) | 0.063 (0.016) | 0.022 (0.004) | 0.100 (0.021) | 0.000 (0.000) | 0.000 (0.000) | 0.000 (0.000) | 0.000 (0.000) |
| Never-married | 500 | 0.034 (0.006) | 0.158 (0.023) | 0.032 (0.006) | 0.147 (0.022) | 0.040 (0.008) | 0.150 (0.022) | 0.025 (0.004) | 0.132 (0.020) | 0.006 (0.002) | 0.050 (0.010) | 0.000 (0.000) | 0.000 (0.000) |
| Never-married | 1000 | 0.043 (0.009) | 0.190 (0.024) | 0.043 (0.009) | 0.182 (0.023) | 0.061 (0.009) | 0.166 (0.025) | 0.019 (0.003) | 0.163 (0.019) | 0.007 (0.002) | 0.079 (0.014) | 0.000 (0.000) | 0.000 (0.000) |
| Never-married | 2000 | 0.055 (0.009) | 0.307 (0.036) | 0.056 (0.009) | 0.309 (0.036) | 0.059 (0.008) | 0.275 (0.032) | 0.020 (0.003) | 0.224 (0.023) | 0.008 (0.002) | 0.129 (0.017) | 0.000 (0.000) | 0.002 (0.002) |
| **Others** | | | | | | | | | | | | | |
| Others | 200 | 0.089 (0.012) | 0.175 (0.028) | 0.050 (0.009) | 0.049 (0.011) | 0.080 (0.012) | 0.171 (0.029) | 0.106 (0.015) | 0.201 (0.029) | 0.002 (0.001) | 0.005 (0.003) | 0.000 (0.000) | 0.000 (0.000) |
| Others | 500 | 0.108 (0.011) | 0.240 (0.025) | 0.108 (0.011) | 0.225 (0.021) | 0.102 (0.012) | 0.205 (0.023) | 0.097 (0.011) | 0.230 (0.025) | 0.021 (0.006) | 0.063 (0.014) | 0.000 (0.000) | 0.000 (0.000) |
| Others | 1000 | 0.080 (0.011) | 0.262 (0.031) | 0.081 (0.011) | 0.255 (0.032) | 0.046 (0.010) | 0.149 (0.036) | 0.071 (0.006) | 0.259 (0.027) | 0.025 (0.005) | 0.142 (0.025) | 0.000 (0.000) | 0.000 (0.000) |
| Others | 2000 | 0.093 (0.012) | 0.288 (0.022) | 0.098 (0.013) | 0.299 (0.022) | 0.039 (0.007) | 0.131 (0.029) | 0.069 (0.010) | 0.256 (0.024) | 0.035 (0.007) | 0.155 (0.020) | 0.000 (0.000) | 0.000 (0.000) |

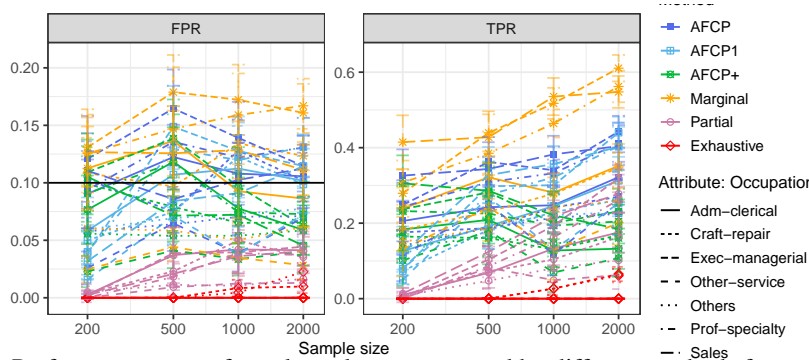

Figure A32: Performance on conformal p-values constructed by different methods for groups formed by Occupation. See Table A30 for numerical details and standard errors.

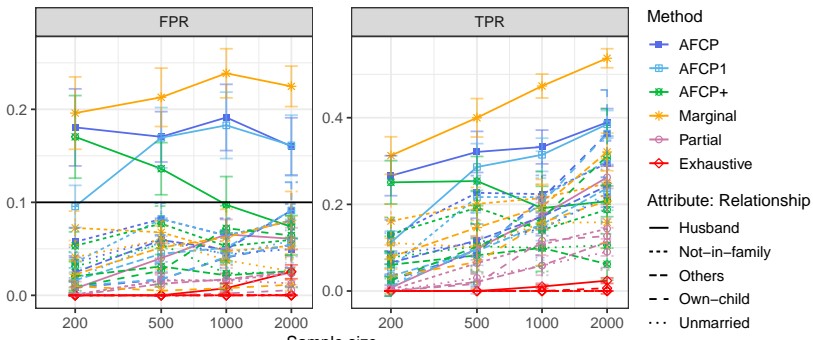

Figure A33: Performance on conformal p-values constructed by different methods for groups formed by Relationship. See Table A31 for numerical details and standard errors.

Table A30: Coverage and size of prediction sets constructed with different methods for groups formed by Occupation. See corresponding plots in Figure A32.

| Attribute: Occupation | Sample size | AFCP | | AFCP1 | | AFCP+ | | Marginal | | Partial | | Exhaustive | |
|---|---|---|---|---|---|---|---|---|---|---|---|---|---|
| | | FPR | TPR | FPR | TPR | FPR | TPR | FPR | TPR | FPR | TPR | FPR | TPR |
| **Adm-clerical** | | | | | | | | | | | | | |
| Adm-clerical | 200 | 0.090 (0.015) | 0.206 (0.027) | 0.058 (0.014) | 0.126 (0.032) | 0.077 (0.015) | 0.183 (0.028) | 0.113 (0.018) | 0.239 (0.028) | 0.003 (0.002) | 0.010 (0.010) | 0.000 (0.000) | 0.000 (0.000) |
| Adm-clerical | 500 | 0.122 (0.014) | 0.242 (0.030) | 0.107 (0.014) | 0.192 (0.030) | 0.118 (0.015) | 0.209 (0.031) | 0.136 (0.015) | 0.321 (0.028) | 0.037 (0.012) | 0.069 (0.020) | 0.000 (0.000) | 0.000 (0.000) |
| Adm-clerical | 1000 | 0.108 (0.014) | 0.248 (0.027) | 0.112 (0.015) | 0.245 (0.025) | 0.078 (0.015) | 0.128 (0.022) | 0.093 (0.012) | 0.281 (0.025) | 0.042 (0.010) | 0.133 (0.022) | 0.000 (0.000) | 0.000 (0.000) |
| Adm-clerical | 2000 | 0.106 (0.014) | 0.322 (0.033) | 0.102 (0.014) | 0.312 (0.032) | 0.059 (0.011) | 0.132 (0.027) | 0.086 (0.012) | 0.352 (0.029) | 0.041 (0.007) | 0.176 (0.027) | 0.000 (0.000) | 0.000 (0.000) |
| **Craft-repair** | | | | | | | | | | | | | |
| Craft-repair | 200 | 0.103 (0.017) | 0.176 (0.030) | 0.050 (0.012) | 0.073 (0.016) | 0.099 (0.016) | 0.165 (0.029) | 0.108 (0.015) | 0.183 (0.029) | 0.004 (0.004) | 0.008 (0.005) | 0.000 (0.000) | 0.000 (0.000) |
| Craft-repair | 500 | 0.075 (0.011) | 0.187 (0.023) | 0.075 (0.013) | 0.182 (0.024) | 0.079 (0.011) | 0.160 (0.023) | 0.097 (0.011) | 0.230 (0.023) | 0.023 (0.006) | 0.071 (0.014) | 0.000 (0.000) | 0.000 (0.000) |
| Craft-repair | 1000 | 0.128 (0.014) | 0.242 (0.027) | 0.123 (0.013) | 0.229 (0.028) | 0.065 (0.013) | 0.131 (0.023) | 0.134 (0.012) | 0.278 (0.021) | 0.050 (0.009) | 0.104 (0.015) | 0.008 (0.004) | 0.027 (0.009) |
| Craft-repair | 2000 | 0.102 (0.012) | 0.270 (0.022) | 0.099 (0.012) | 0.262 (0.022) | 0.065 (0.009) | 0.168 (0.017) | 0.123 (0.012) | 0.346 (0.020) | 0.035 (0.005) | 0.163 (0.013) | 0.010 (0.003) | 0.065 (0.010) |
| **Exec-managerial** | | | | | | | | | | | | | |
| Exec-managerial | 200 | 0.121 (0.019) | 0.248 (0.033) | 0.039 (0.009) | 0.079 (0.021) | 0.109 (0.017) | 0.237 (0.033) | 0.131 (0.017) | 0.279 (0.031) | 0.001 (0.001) | 0.003 (0.003) | 0.000 (0.000) | 0.000 (0.000) |
| Exec-managerial | 500 | 0.165 (0.017) | 0.362 (0.033) | 0.149 (0.018) | 0.298 (0.033) | 0.139 (0.017) | 0.282 (0.034) | 0.179 (0.016) | 0.439 (0.025) | 0.039 (0.011) | 0.103 (0.017) | 0.000 (0.000) | 0.000 (0.000) |
| Exec-managerial | 1000 | 0.139 (0.016) | 0.340 (0.031) | 0.128 (0.015) | 0.300 (0.024) | 0.097 (0.017) | 0.201 (0.038) | 0.172 (0.015) | 0.517 (0.021) | 0.040 (0.008) | 0.193 (0.023) | 0.000 (0.000) | 0.000 (0.000) |
| Exec-managerial | 2000 | 0.115 (0.013) | 0.442 (0.021) | 0.115 (0.013) | 0.440 (0.021) | 0.062 (0.013) | 0.231 (0.030) | 0.161 (0.013) | 0.611 (0.018) | 0.044 (0.008) | 0.319 (0.018) | 0.000 (0.000) | 0.000 (0.000) |
| **Other-service** | | | | | | | | | | | | | |
| Other-service | 200 | 0.026 (0.007) | 0.114 (0.034) | 0.016 (0.006) | 0.050 (0.022) | 0.023 (0.007) | 0.101 (0.034) | 0.025 (0.006) | 0.131 (0.036) | 0.000 (0.000) | 0.000 (0.000) | 0.000 (0.000) | 0.000 (0.000) |
| Other-service | 500 | 0.065 (0.012) | 0.249 (0.055) | 0.071 (0.012) | 0.269 (0.053) | 0.041 (0.008) | 0.180 (0.044) | 0.044 (0.006) | 0.242 (0.045) | 0.009 (0.004) | 0.082 (0.028) | 0.000 (0.000) | 0.000 (0.000) |
| Other-service | 1000 | 0.039 (0.008) | 0.118 (0.028) | 0.038 (0.008) | 0.129 (0.033) | 0.034 (0.007) | 0.069 (0.026) | 0.035 (0.006) | 0.129 (0.040) | 0.012 (0.004) | 0.049 (0.018) | 0.000 (0.000) | 0.000 (0.000) |
| Other-service | 2000 | 0.070 (0.011) | 0.253 (0.042) | 0.073 (0.011) | 0.253 (0.042) | 0.040 (0.009) | 0.107 (0.025) | 0.028 (0.006) | 0.201 (0.029) | 0.013 (0.004) | 0.062 (0.018) | 0.000 (0.000) | 0.000 (0.000) |
| **Others** | | | | | | | | | | | | | |
| Others | 200 | 0.058 (0.012) | 0.141 (0.021) | 0.059 (0.012) | 0.118 (0.024) | 0.056 (0.012) | 0.137 (0.021) | 0.060 (0.012) | 0.151 (0.024) | 0.005 (0.002) | 0.007 (0.004) | 0.000 (0.000) | 0.000 (0.000) |
| Others | 500 | 0.064 (0.009) | 0.187 (0.024) | 0.069 (0.009) | 0.195 (0.024) | 0.054 (0.010) | 0.163 (0.026) | 0.058 (0.008) | 0.187 (0.023) | 0.011 (0.004) | 0.047 (0.011) | 0.000 (0.000) | 0.000 (0.000) |
| Others | 1000 | 0.070 (0.007) | 0.214 (0.019) | 0.073 (0.007) | 0.221 (0.022) | 0.054 (0.008) | 0.146 (0.017) | 0.050 (0.007) | 0.243 (0.013) | 0.010 (0.003) | 0.081 (0.010) | 0.004 (0.002) | 0.027 (0.009) |
| Others | 2000 | 0.071 (0.008) | 0.225 (0.018) | 0.077 (0.010) | 0.229 (0.018) | 0.059 (0.005) | 0.161 (0.017) | 0.062 (0.008) | 0.259 (0.016) | 0.017 (0.003) | 0.102 (0.012) | 0.023 (0.004) | 0.061 (0.009) |
| **Prof-specialty** | | | | | | | | | | | | | |
| Prof-specialty | 200 | 0.094 (0.017) | 0.243 (0.032) | 0.040 (0.012) | 0.079 (0.020) | 0.091 (0.016) | 0.232 (0.033) | 0.125 (0.017) | 0.308 (0.029) | 0.000 (0.000) | 0.000 (0.000) | 0.000 (0.000) | 0.000 (0.000) |
| Prof-specialty | 500 | 0.136 (0.024) | 0.309 (0.033) | 0.129 (0.019) | 0.275 (0.034) | 0.118 (0.021) | 0.225 (0.031) | 0.147 (0.027) | 0.385 (0.029) | 0.018 (0.006) | 0.086 (0.015) | 0.000 (0.000) | 0.000 (0.000) |
| Prof-specialty | 1000 | 0.122 (0.016) | 0.332 (0.018) | 0.120 (0.016) | 0.321 (0.019) | 0.074 (0.015) | 0.193 (0.026) | 0.159 (0.018) | 0.464 (0.019) | 0.039 (0.010) | 0.163 (0.018) | 0.000 (0.000) | 0.000 (0.000) |
| Prof-specialty | 2000 | 0.131 (0.013) | 0.411 (0.024) | 0.131 (0.013) | 0.404 (0.026) | 0.073 (0.014) | 0.205 (0.029) | 0.167 (0.012) | 0.562 (0.021) | 0.037 (0.008) | 0.262 (0.017) | 0.000 (0.000) | 0.000 (0.000) |
| **Sales** | | | | | | | | | | | | | |
| Sales | 200 | 0.110 (0.016) | 0.326 (0.035) | 0.032 (0.008) | 0.120 (0.022) | 0.105 (0.016) | 0.306 (0.037) | 0.127 (0.015) | 0.415 (0.036) | 0.002 (0.002) | 0.013 (0.007) | 0.000 (0.000) | 0.000 (0.000) |
| Sales | 500 | 0.086 (0.015) | 0.344 (0.035) | 0.084 (0.014) | 0.327 (0.037) | 0.071 (0.011) | 0.286 (0.038) | 0.126 (0.019) | 0.428 (0.034) | 0.021 (0.005) | 0.124 (0.024) | 0.000 (0.000) | 0.000 (0.000) |
| Sales | 1000 | 0.103 (0.016) | 0.382 (0.025) | 0.091 (0.015) | 0.357 (0.024) | 0.072 (0.013) | 0.221 (0.032) | 0.128 (0.012) | 0.535 (0.025) | 0.036 (0.008) | 0.218 (0.021) | 0.000 (0.000) | 0.000 (0.000) |
| Sales | 2000 | 0.112 (0.014) | 0.404 (0.031) | 0.114 (0.014) | 0.402 (0.031) | 0.045 (0.009) | 0.185 (0.032) | 0.111 (0.011) | 0.548 (0.021) | 0.041 (0.009) | 0.278 (0.025) | 0.000 (0.000) | 0.000 (0.000) |

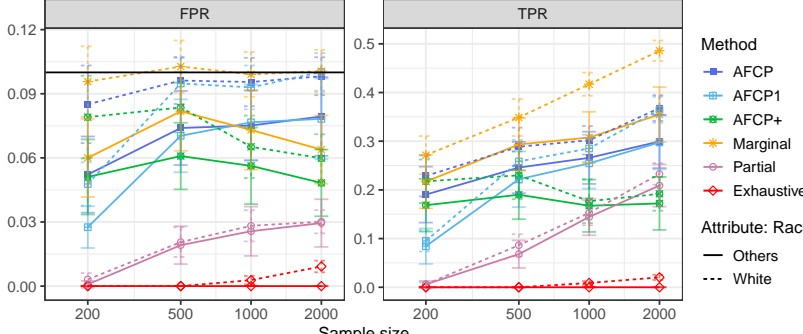

Figure A34: Performance on conformal p-values constructed by different methods for groups formed by Race. See Table A32 for numerical details and standard errors.

Table A31: Coverage and size of prediction sets constructed with different methods for groups formed by Relationship. See corresponding plots in Figure A33.

| Attribute: Relationship | Sample size | AFCP FPR | AFCP TPR | AFCP1 FPR | AFCP1 TPR | AFCP+ FPR | AFCP+ TPR | Marginal FPR | Marginal TPR | Partial FPR | Partial TPR | Exhaustive FPR | Exhaustive TPR |
|---|---|---|---|---|---|---|---|---|---|---|---|---|---|
| **Husband** | | | | | | | | | | | | | |
| Husband | 200 | 0.181 (0.021) | 0.266 (0.023) | 0.096 (0.011) | 0.112 (0.013) | 0.170 (0.022) | 0.251 (0.025) | 0.196 (0.019) | 0.313 (0.022) | 0.007 (0.004) | 0.007 (0.003) | 0.000 (0.000) | 0.000 (0.000) |
| Husband | 500 | 0.170 (0.014) | 0.321 (0.024) | 0.169 (0.016) | 0.286 (0.027) | 0.136 (0.014) | 0.254 (0.028) | 0.213 (0.016) | 0.400 (0.022) | 0.040 (0.008) | 0.098 (0.013) | 0.000 (0.000) | 0.000 (0.000) |
| Husband | 1000 | 0.191 (0.018) | 0.332 (0.019) | 0.183 (0.018) | 0.314 (0.019) | 0.097 (0.015) | 0.191 (0.025) | 0.239 (0.013) | 0.473 (0.014) | 0.066 (0.008) | 0.174 (0.015) | 0.008 (0.003) | 0.010 (0.002) |
| Husband | 2000 | 0.160 (0.015) | 0.389 (0.016) | 0.161 (0.016) | 0.385 (0.016) | 0.074 (0.006) | 0.207 (0.018) | 0.225 (0.011) | 0.537 (0.011) | 0.061 (0.006) | 0.262 (0.013) | 0.025 (0.004) | 0.024 (0.003) |
| **Not-in-family** | | | | | | | | | | | | | |
| Not-in-family | 200 | 0.057 (0.007) | 0.127 (0.018) | 0.035 (0.005) | 0.054 (0.010) | 0.053 (0.007) | 0.131 (0.019) | 0.072 (0.009) | 0.161 (0.020) | 0.001 (0.001) | 0.003 (0.002) | 0.000 (0.000) | 0.000 (0.000) |
| Not-in-family | 500 | 0.082 (0.009) | 0.227 (0.020) | 0.082 (0.008) | 0.217 (0.018) | 0.077 (0.009) | 0.193 (0.019) | 0.068 (0.006) | 0.201 (0.019) | 0.016 (0.003) | 0.064 (0.009) | 0.000 (0.000) | 0.000 (0.000) |
| Not-in-family | 1000 | 0.064 (0.009) | 0.224 (0.022) | 0.064 (0.009) | 0.217 (0.022) | 0.053 (0.009) | 0.141 (0.022) | 0.048 (0.005) | 0.215 (0.016) | 0.016 (0.004) | 0.108 (0.012) | 0.000 (0.000) | 0.000 (0.000) |
| Not-in-family | 2000 | 0.080 (0.009) | 0.295 (0.026) | 0.081 (0.009) | 0.300 (0.025) | 0.060 (0.008) | 0.188 (0.028) | 0.053 (0.004) | 0.248 (0.015) | 0.023 (0.004) | 0.144 (0.012) | 0.000 (0.000) | 0.000 (0.000) |
| **Others** | | | | | | | | | | | | | |
| Others | 200 | 0.025 (0.008) | 0.066 (0.013) | 0.016 (0.007) | 0.035 (0.010) | 0.021 (0.008) | 0.061 (0.014) | 0.022 (0.007) | 0.078 (0.014) | 0.000 (0.000) | 0.000 (0.000) | 0.000 (0.000) | 0.000 (0.000) |
| Others | 500 | 0.059 (0.012) | 0.117 (0.019) | 0.045 (0.009) | 0.093 (0.017) | 0.031 (0.010) | 0.082 (0.019) | 0.051 (0.012) | 0.146 (0.021) | 0.012 (0.006) | 0.021 (0.009) | 0.000 (0.000) | 0.000 (0.000) |
| Others | 1000 | 0.049 (0.008) | 0.172 (0.022) | 0.051 (0.008) | 0.154 (0.019) | 0.021 (0.008) | 0.100 (0.027) | 0.062 (0.015) | 0.195 (0.026) | 0.017 (0.006) | 0.059 (0.011) | 0.000 (0.000) | 0.000 (0.000) |
| Others | 2000 | 0.091 (0.015) | 0.243 (0.025) | 0.091 (0.015) | 0.237 (0.025) | 0.026 (0.009) | 0.062 (0.018) | 0.081 (0.015) | 0.321 (0.022) | 0.026 (0.008) | 0.114 (0.016) | 0.000 (0.000) | 0.000 (0.000) |
| **Own-child** | | | | | | | | | | | | | |
| Own-child | 200 | 0.008 (0.003) | 0.025 (0.018) | 0.010 (0.003) | 0.011 (0.011) | 0.009 (0.003) | 0.025 (0.018) | 0.010 (0.003) | 0.033 (0.020) | 0.001 (0.001) | 0.000 (0.000) | 0.000 (0.000) | 0.000 (0.000) |
| Own-child | 500 | 0.017 (0.009) | 0.089 (0.040) | 0.017 (0.009) | 0.097 (0.042) | 0.026 (0.009) | 0.103 (0.041) | 0.005 (0.002) | 0.067 (0.030) | 0.000 (0.000) | 0.017 (0.012) | 0.000 (0.000) | 0.000 (0.000) |
| Own-child | 1000 | 0.041 (0.014) | 0.214 (0.060) | 0.041 (0.014) | 0.203 (0.060) | 0.072 (0.012) | 0.167 (0.055) | 0.008 (0.002) | 0.158 (0.050) | 0.002 (0.001) | 0.117 (0.049) | 0.000 (0.000) | 0.000 (0.000) |
| Own-child | 2000 | 0.054 (0.012) | 0.363 (0.050) | 0.054 (0.012) | 0.363 (0.050) | 0.064 (0.011) | 0.312 (0.055) | 0.011 (0.003) | 0.208 (0.049) | 0.006 (0.002) | 0.127 (0.036) | 0.000 (0.000) | 0.007 (0.007) |
| **Unmarried** | | | | | | | | | | | | | |
| Unmarried | 200 | 0.035 (0.008) | 0.084 (0.025) | 0.018 (0.006) | 0.015 (0.011) | 0.031 (0.008) | 0.078 (0.024) | 0.041 (0.009) | 0.111 (0.029) | 0.000 (0.000) | 0.000 (0.000) | 0.000 (0.000) | 0.000 (0.000) |
| Unmarried | 500 | 0.060 (0.010) | 0.112 (0.027) | 0.057 (0.010) | 0.102 (0.027) | 0.055 (0.011) | 0.103 (0.027) | 0.060 (0.010) | 0.101 (0.025) | 0.017 (0.007) | 0.030 (0.013) | 0.000 (0.000) | 0.000 (0.000) |
| Unmarried | 1000 | 0.044 (0.007) | 0.144 (0.034) | 0.044 (0.009) | 0.142 (0.035) | 0.024 (0.007) | 0.099 (0.029) | 0.038 (0.007) | 0.157 (0.031) | 0.016 (0.004) | 0.059 (0.023) | 0.000 (0.000) | 0.000 (0.000) |
| Unmarried | 2000 | 0.050 (0.010) | 0.218 (0.031) | 0.058 (0.012) | 0.226 (0.031) | 0.025 (0.008) | 0.106 (0.027) | 0.026 (0.006) | 0.159 (0.030) | 0.013 (0.005) | 0.090 (0.021) | 0.000 (0.000) | 0.000 (0.000) |

Table A32: Coverage and size of prediction sets constructed with different methods for groups formed by Race. See corresponding plots in Figure A34.

| Attribute: Race | Sample size | AFCP FPR | AFCP TPR | AFCP1 FPR | AFCP1 TPR | AFCP+ FPR | AFCP+ TPR | Marginal FPR | Marginal TPR | Partial FPR | Partial TPR | Exhaustive FPR | Exhaustive TPR |
|---|---|---|---|---|---|---|---|---|---|---|---|---|---|
| **Others** | | | | | | | | | | | | | |
| Others | 200 | 0.052 (0.009) | 0.190 (0.029) | 0.028 (0.005) | 0.084 (0.018) | 0.051 (0.009) | 0.169 (0.027) | 0.060 (0.009) | 0.217 (0.028) | 0.001 (0.001) | 0.006 (0.004) | 0.000 (0.000) | 0.000 (0.000) |
| Others | 500 | 0.074 (0.009) | 0.246 (0.026) | 0.070 (0.009) | 0.222 (0.029) | 0.061 (0.008) | 0.190 (0.029) | 0.082 (0.009) | 0.293 (0.027) | 0.019 (0.004) | 0.068 (0.014) | 0.000 (0.000) | 0.000 (0.000) |
| Others | 1000 | 0.075 (0.008) | 0.266 (0.027) | 0.077 (0.009) | 0.254 (0.026) | 0.056 (0.009) | 0.167 (0.027) | 0.073 (0.009) | 0.308 (0.026) | 0.026 (0.006) | 0.144 (0.019) | 0.000 (0.000) | 0.000 (0.000) |
| Others | 2000 | 0.079 (0.009) | 0.299 (0.027) | 0.078 (0.009) | 0.298 (0.028) | 0.048 (0.008) | 0.172 (0.027) | 0.064 (0.008) | 0.355 (0.028) | 0.029 (0.006) | 0.209 (0.022) | 0.000 (0.000) | 0.000 (0.000) |
| **White** | | | | | | | | | | | | | |
| White | 200 | 0.085 (0.009) | 0.229 (0.021) | 0.048 (0.005) | 0.095 (0.011) | 0.079 (0.010) | 0.218 (0.022) | 0.096 (0.008) | 0.271 (0.020) | 0.003 (0.001) | 0.006 (0.002) | 0.000 (0.000) | 0.000 (0.000) |
| White | 500 | 0.096 (0.005) | 0.289 (0.020) | 0.095 (0.006) | 0.258 (0.022) | 0.084 (0.006) | 0.230 (0.024) | 0.103 (0.006) | 0.349 (0.019) | 0.021 (0.003) | 0.086 (0.011) | 0.000 (0.000) | 0.000 (0.000) |
| White | 1000 | 0.095 (0.006) | 0.302 (0.015) | 0.093 (0.005) | 0.285 (0.014) | 0.065 (0.007) | 0.177 (0.023) | 0.099 (0.005) | 0.417 (0.012) | 0.028 (0.004) | 0.153 (0.013) | 0.003 (0.001) | 0.009 (0.002) |
| White | 2000 | 0.098 (0.004) | 0.368 (0.013) | 0.100 (0.004) | 0.365 (0.013) | 0.060 (0.006) | 0.192 (0.017) | 0.101 (0.005) | 0.486 (0.011) | 0.030 (0.003) | 0.233 (0.011) | 0.009 (0.001) | 0.020 (0.003) |

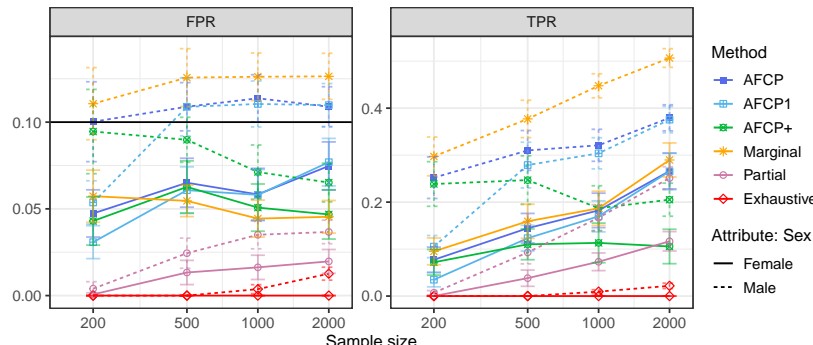

Figure A35: Performance on conformal p-values constructed by different methods for groups formed by Sex. See Table A33 for numerical details and standard errors.

Table A33: Coverage and size of prediction sets constructed with different methods for groups formed by Sex. See corresponding plots in Figure A35.

| Attribute: Sex | Sample size | AFCP | | AFCP1 | | AFCP+ | | Marginal | | Partial | | Exhaustive | |
|---|---|---|---|---|---|---|---|---|---|---|---|---|---|
| | | FPR | TPR | FPR | TPR | FPR | TPR | FPR | TPR | FPR | TPR | FPR | TPR |
| **Female** | | | | | | | | | | | | | |
| Female | 200 | 0.047 (0.007) | 0.077 (0.013) | 0.031 (0.005) | 0.035 (0.008) | 0.043 (0.007) | 0.072 (0.014) | 0.057 (0.008) | 0.095 (0.014) | 0.001 (0.000) | 0.000 (0.000) | 0.000 (0.000) | 0.000 (0.000) |
| Female | 500 | 0.065 (0.007) | 0.144 (0.016) | 0.061 (0.007) | 0.123 (0.014) | 0.063 (0.008) | 0.110 (0.016) | 0.055 (0.005) | 0.159 (0.018) | 0.013 (0.003) | 0.038 (0.009) | 0.000 (0.000) | 0.000 (0.000) |
| Female | 1000 | 0.058 (0.008) | 0.182 (0.018) | 0.058 (0.007) | 0.170 (0.017) | 0.051 (0.007) | 0.113 (0.021) | 0.044 (0.005) | 0.187 (0.018) | 0.016 (0.003) | 0.073 (0.009) | 0.000 (0.000) | 0.000 (0.000) |
| Female | 2000 | 0.075 (0.007) | 0.266 (0.019) | 0.077 (0.007) | 0.265 (0.020) | 0.047 (0.007) | 0.106 (0.018) | 0.045 (0.005) | 0.290 (0.018) | 0.020 (0.003) | 0.117 (0.010) | 0.000 (0.000) | 0.000 (0.000) |
| **Male** | | | | | | | | | | | | | |
| Male | 200 | 0.100 (0.012) | 0.252 (0.022) | 0.054 (0.006) | 0.105 (0.012) | 0.095 (0.012) | 0.239 (0.024) | 0.111 (0.010) | 0.297 (0.021) | 0.004 (0.002) | 0.007 (0.002) | 0.000 (0.000) | 0.000 (0.000) |
| Male | 500 | 0.109 (0.007) | 0.310 (0.021) | 0.109 (0.008) | 0.279 (0.024) | 0.090 (0.006) | 0.247 (0.026) | 0.126 (0.008) | 0.377 (0.020) | 0.024 (0.004) | 0.093 (0.012) | 0.000 (0.000) | 0.000 (0.000) |
| Male | 1000 | 0.114 (0.007) | 0.321 (0.017) | 0.110 (0.007) | 0.304 (0.017) | 0.071 (0.008) | 0.187 (0.024) | 0.126 (0.007) | 0.448 (0.013) | 0.035 (0.004) | 0.167 (0.014) | 0.004 (0.001) | 0.009 (0.002) |
| Male | 2000 | 0.109 (0.006) | 0.379 (0.014) | 0.110 (0.006) | 0.376 (0.014) | 0.065 (0.006) | 0.205 (0.018) | 0.126 (0.007) | 0.507 (0.010) | 0.037 (0.003) | 0.251 (0.012) | 0.013 (0.002) | 0.022 (0.003) |

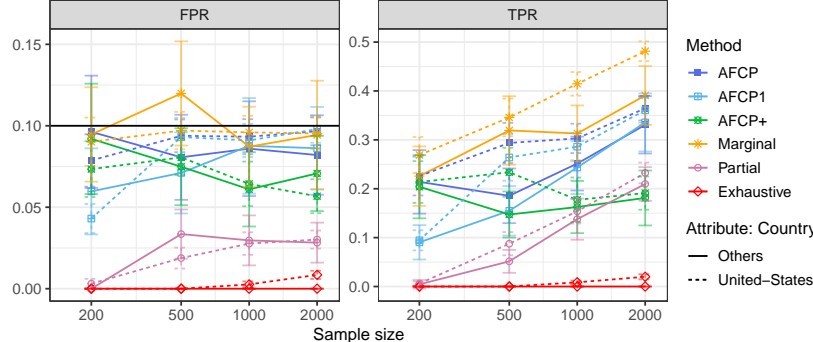

Figure A36: Performance on conformal p-values constructed by different methods for groups formed by Country. See Table A34 for numerical details and standard errors.

Table A34: Coverage and size of prediction sets constructed with different methods for groups formed by Country. See corresponding plots in Figure A36.

| Attribute: Country | Sample size | AFCP | | AFCP1 | | AFCP+ | | Marginal | | Partial | | Exhaustive | |
|---|---|---|---|---|---|---|---|---|---|---|---|---|---|
| | | FPR | TPR | FPR | TPR | FPR | TPR | FPR | TPR | FPR | TPR | FPR | TPR |
| **Others** | | | | | | | | | | | | | |
| Others | 200 | 0.096 (0.017) | 0.214 (0.032) | 0.060 (0.013) | 0.090 (0.018) | 0.092 (0.017) | 0.204 (0.032) | 0.095 (0.014) | 0.226 (0.030) | 0.000 (0.000) | 0.004 (0.004) | 0.000 (0.000) | 0.000 (0.000) |
| Others | 500 | 0.081 (0.013) | 0.186 (0.028) | 0.071 (0.012) | 0.155 (0.025) | 0.075 (0.012) | 0.147 (0.024) | 0.120 (0.016) | 0.319 (0.035) | 0.034 (0.008) | 0.051 (0.012) | 0.000 (0.000) | 0.000 (0.000) |
| Others | 1000 | 0.086 (0.015) | 0.252 (0.028) | 0.088 (0.015) | 0.243 (0.027) | 0.061 (0.011) | 0.163 (0.027) | 0.087 (0.012) | 0.313 (0.029) | 0.030 (0.008) | 0.138 (0.021) | 0.000 (0.000) | 0.000 (0.000) |
| Others | 2000 | 0.082 (0.012) | 0.331 (0.029) | 0.086 (0.013) | 0.336 (0.030) | 0.071 (0.012) | 0.181 (0.028) | 0.094 (0.017) | 0.391 (0.030) | 0.028 (0.006) | 0.210 (0.017) | 0.000 (0.000) | 0.000 (0.000) |
| **United-States** | | | | | | | | | | | | | |
| United-States | 200 | 0.079 (0.008) | 0.226 (0.020) | 0.043 (0.004) | 0.094 (0.010) | 0.073 (0.009) | 0.214 (0.021) | 0.090 (0.007) | 0.269 (0.018) | 0.003 (0.001) | 0.006 (0.002) | 0.000 (0.000) | 0.000 (0.000) |
| United-States | 500 | 0.094 (0.005) | 0.294 (0.021) | 0.093 (0.006) | 0.264 (0.024) | 0.081 (0.005) | 0.234 (0.025) | 0.097 (0.006) | 0.346 (0.019) | 0.019 (0.003) | 0.088 (0.012) | 0.000 (0.000) | 0.000 (0.000) |
| United-States | 1000 | 0.093 (0.005) | 0.303 (0.015) | 0.091 (0.005) | 0.286 (0.014) | 0.064 (0.007) | 0.177 (0.023) | 0.096 (0.005) | 0.415 (0.012) | 0.028 (0.003) | 0.154 (0.012) | 0.003 (0.001) | 0.009 (0.002) |
| United-States | 2000 | 0.097 (0.004) | 0.365 (0.012) | 0.098 (0.004) | 0.360 (0.013) | 0.057 (0.005) | 0.191 (0.017) | 0.095 (0.005) | 0.481 (0.010) | 0.030 (0.003) | 0.232 (0.011) | 0.008 (0.001) | 0.020 (0.003) |

