# OpenReview forum: "Conformal Classification with Equalized Coverage for Adaptively Selected Groups"
_NeurIPS.cc/2024/Conference — NeurIPS 2024 poster_

### Official Review · Reviewer_92Kn · 2024-07-10

**Soundness:** 4
**Presentation:** 4
**Contribution:** 3
**Rating:** 7
**Confidence:** 3

**Summary:**

This paper presents a conformal inference method to assess uncertainty in classification by generating prediction sets with valid coverage based on adaptively chosen features. Falling between marginal and strictly conditional coverage, the features in the proposed method are adaptively selected to address potential model limitations or biases, balancing the need for informative predictions with ensuring algorithmic fairness.

**Strengths:**

- The paper is well written, with good organization, clear problem statement and easy-understanding method description.
-  The proposed fairness notion is novel and critical for real-world application. It provides a feasible solution to migitage the tradeoff between fairness and efficiency. It is also impressing that the method works with small sample size.
- The experiments and theorem are comphrensive and supportive of method and claims.

**Weaknesses:**

- It seems that *adaptive equalized coverage* requires each group to surpass a given coverage rate. Desirably like RAPS, the coverage rate should be the same at desired level to guarantee exact demographic parity.
- Though a little out of scope, this paper would be better to provide performance evaluation with presence of distribution shift, especially label shift and shift in protected attributes.

**Questions:**

- Would your method preserve satisfy more rigorous fairness (Q1) with better design? I think this may be related with union of sub-intervals. Adaptively adjust the coverage rate of subgroup may help.
- Also, the union of sub-intervals may make less sense in some real-world applications. Say, in some data, the target variables are some categorical and also ordered labels, say level 1~5. Predicting a sample to be either level 1 or 5  is confusing. Have this phenomenon be considered and addressed in your paper?

**Limitations:**

Yes

---

> ### Author Rebuttal · Authors · 2024-08-06
>
> Thank you for your careful review and insightful feedback.
>
> ## Seeking Different Fairness Criteria
>
> The potential for our method to be adapted based on different fairness criteria, going beyond our notion of selectively equalized coverage, is indeed both intriguing and promising. Although exploring those questions goes beyond the immediate scope of this paper, we hope that others will be inspired to improve and build upon our work in this direction. We will definitely add this suggestion in the revised manuscript.
>
> ## Classification with Ordered Labels
>
> In our current framework, we have focused on predicting categorical (unordered) labels. If the labels are ordered, an easy modification of our method would require only a small change to Equation (8). Specifically, when the target variables are ordered labels (say level 1 $<$ level 5), then $\hat{C}(X_{n+1})$ becomes the discrete convex hull of all its components on the right-hand side in Equation 8. Because the prediction set resulting from taking unions will be a subset of the prediction set resulting from taking discrete convex hulls, the new prediction set will automatically satisfy the adaptive equalized coverage defined in Equation (3). An intriguing question, which may be investigated in future work, is whether it is possible to devise a better approach for ordinal labels. Thank you for this helpful comment. We will include this discussion in the revised paper.

---

### Official Review · Reviewer_FDCZ · 2024-07-12

**Soundness:** 3
**Presentation:** 3
**Contribution:** 2
**Rating:** 5
**Confidence:** 4

**Summary:**

The paper introduces a conformal inference method to assess uncertainty in classification by generating prediction sets with valid coverage, conditional on adaptively chosen features. These features are selected to address model limitations or biases, balancing efficiency and fairness by ensuring equalized coverage for sensitive groups. The paper demonstrates this method's validity and effectiveness on both simulated and real datasets.

**Strengths:**

1. The method efficiently identifies and addresses algorithmic biases, ensuring fair treatment of sensitive groups without sacrificing informativeness.
2. AFCP provides a practical compromise between efficient, informative predictions and algorithmic fairness by adjusting prediction sets for sensitive groups.
3. Demonstrated effectiveness on both synthetic and real-world datasets, outperforming traditional methods in terms of both fairness and prediction informativeness.

**Weaknesses:**

1.	Limitation on Sensitive Attribute Selection: The current method may not always identify the most relevant sensitive attribute, especially with limited sample sizes or overlapping biases. The ‘Automatic Attribute Selection’ section is somewhat challenging to follow. For instance, in Equation 6, it seems that only the argmax element is included in the set, and the algorithm does not seem to provide a sensitive attribute with formal guarantees. Since much of the paper’s contribution hinges on this algorithm, the lack of clarity in its description makes it hard to be convinced of its effectiveness.
2.	Sample Size Sensitivity: AFCP’s performance can be constrained by the sample size. Smaller sample sizes may result in less reliable attribute selection and less informative prediction sets. In practical datasets, sample sizes tend to be small when many features are selected. The Sensitive Attribute Selection algorithm may lead to a large label set, resulting in too few samples to accurately determine the thresholds.
3.	Computational Complexity: With a large label set, such as in ImageNet which has 1000 labels, this method involves complex procedures for attribute selection and prediction set construction. This complexity may make the method computationally intensive.

**Questions:**

1.	Coverage for Occupation 1 in Figure 5: The coverage for Occupation 1 in Figure 5 shows a significant increase in the Marginal method when the sample size increases from 2000 to 5000. This rapid increase is puzzling, particularly since the average size of the prediction set also decreases in this interval. Could you clarify why this behavior occurs? It seems counterintuitive, as one would generally expect a consistent relationship between coverage and prediction set size.
2.	Impact of Calibration Set Size on Sensitive Attribute Selection and Final Prediction Set: How do you think the calibration set size influences the algorithm for Sensitive Attribute Selection and the final prediction set? Specifically:

**Limitations:**

Please refer to the weakness.

---

> ### Author Rebuttal · Authors · 2024-08-06
>
> Thank you for your detailed review, which allows us to clarify key points and address potential misunderstandings. We believe these clarifications will enhance the paper's accessibility.
>
> ## Clarification on Method Aims
>
> While other reviewers have praised the paper for its clarity, we appreciate the opportunity to refine our explanations and address your concerns. Specifically, we would like to clarify the role of the attribute selection component in our method, and the reason why we provide formal guarantees for the coverage of the prediction sets rather than for the "correctness" of the selection procedure.
>
> This paper addresses the problem of constructing prediction sets with guaranteed coverage conditional on adaptively selected sensitive attributes. As you noted, our goal is to bridge the gap between existing approaches for marginal and group-conditional conformal prediction, finding a practical balance between efficiency and fairness.
>
> To obtain reliable inferences and approximately maximize conditional coverage, our method seeks to identify the attribute (or attributes) most likely to result in significant under-coverage under a standard (marginal) conformal inference approach. The specific selection algorithm proposed in this paper has demonstrated strong practical performance, as evidenced by our numerical experiments (see Figure 4 and the new Figure 2 in the PDF supplement accompanying this response). However, the core ideas of our method are flexible enough to accommodate different selection algorithms. In the revised manuscript, we will better highlight this flexibility.
>
> A consequence of this flexibility and "assumption-lean" setup is that establishing formal guarantees about the selection of the "correct" attribute would be challenging and somewhat limiting. It would require stronger assumptions, and it would necessitate a theoretical analysis heavily dependent on the specific implementation details of the selection algorithm. While this may be an interesting direction for further research, potentially strengthening the connection to Cherian and Candès (2023), it falls outside the scope of this paper.
>
> ## Impact of Calibration Sample Size
>
> Constructing informative prediction sets with high conditional coverage is more challenging with smaller sample sizes. This is an inherent challenge, not a specific limitation of our method, and it explains why our experiments focus on comparing the performance of our method against several benchmarks across various sample sizes. These experiments demonstrate that our method consistently performs well across all sample size scenarios. For detailed results, see Figures 3–5 in the paper and the new Figures 1–3 in the PDF supplement accompanying this response.
>
> For example, in the experiments pf Figures 3 and 4, when the sample size is as small as 200, it is inherently challenging to: (1) fit an accurate predictive model, (2) assess conditional coverage, (3) and reliably identify the sensitive attribute associated with the lowest coverage. This is reflected by the relatively large size of the prediction sets output by all methods and by the relatively large discrepancies between the nominal and empirical conditional coverage levels. Nevertheless, our method manages to frequently select the "correct" sensitive attribute, achieving significantly higher conditional coverage compared to the standard "marginal" benchmark, with only a slight increase in prediction set sizes. Further, as the sample size increases, our method becomes highly effective at identifying the attribute with the lowest conditional coverage, as illustrated in Figure 4. This allows our method to achieve high conditional coverage with relatively small prediction sets. Overall, these experiments demonstrate that our method offers distinct advantages over existing approaches in both large-sample and small-sample settings.
>
> ## Applicability and Computational Cost
>
> Our proposed AFCP method is quite broadly applicable and can be efficiently implemented in many classification settings where the number of labels is not exceedingly large. Natural applications include predicting recidivism risks, determining graduation classes, and addressing binary classification problems such as spam detection and credit default risk assessment.
>
> However, in scenarios where the number of possible classes is extremely high, it is true that the computational cost of our method may become a limiting factor. While addressing this issue is beyond the scope of this paper, we believe that extending our method to accommodate extreme classification tasks presents valuable opportunities for future research. In the revision, we will highlight these opportunities and cite relevant works in the conformal inference literature, such as [1]. These references will provide additional context and potential inspiration for future developments.
>
> Reference: [1]"Class-Conditional Conformal Prediction with Many Classes." Ding et. al., NeurIPS 2023.
>
> ## Clarifications on Figure 5
>
> The behavior observed in Figure 5 can be easily explained. The horizontal axis represents the cumulative sample size, which includes both training and calibration samples. As the cumulative sample size increases, more data becomes available to train a more accurate predictive model. This increased accuracy explains why the conditional coverage for Occupation 1 improves while the average prediction set size decreases: the model becomes more precise for all individuals, including those most affected by algorithmic bias.
>
> The main takeaway from Figure 5 is that our method consistently provides relatively informative and reliable predictive inferences compared to other conformal inference approaches, regardless of the amount of available data or the varying accuracy of the underlying machine learning model. Note that the model varies for different sample sizes but remains the same for all conformal inference methods, ensuring a fair comparison.

---

> > ### Comment · Reviewer_FDCZ · 2024-08-11
> >
> > Thank you for the thorough response, I raise my rating to 5.

---

### Official Review · Reviewer_asc2 · 2024-07-13

**Soundness:** 3
**Presentation:** 3
**Contribution:** 3
**Rating:** 6
**Confidence:** 5

**Summary:**

The paper presents a novel conformal inference method aimed at generating prediction sets with valid coverage, conditional on adaptively chosen features. This method is intended to address the dual concerns of efficiency and algorithmic fairness by ensuring equalized coverage for the most sensitive groups, thus providing informative predictions while maintaining fairness. The proposed approach, termed Adaptively Fair Conformal Prediction (AFCP), is validated on both simulated and real datasets.

**Strengths:**

1. The paper addresses a significant and timely problem in machine learning—ensuring fairness and reliability in prediction sets. The introduction of AFCP, which dynamically adjusts for biases in a data-driven manner, is an innovative and practical contribution.
2. The paper provides a strong theoretical basis for AFCP, clearly defining adaptive equalized coverage and offering proofs to support the validity of the method.
3. The empirical results are robust, covering synthetic and real-world datasets. The comparisons with other benchmarks are comprehensive, demonstrating the practical benefits of AFCP in various scenarios.
4. The method's steps, including automatic attribute selection and prediction set construction, are well-explained and logically structured.

**Weaknesses:**

1. While the paper acknowledges the scalability issues associated with current methods for conformal inference with equalized coverage, it does not provide a detailed analysis of the computational complexity of the proposed method. A deeper exploration of scalability, particularly for large-scale datasets, would strengthen the paper.
2. The method's reliance on leave-one-out procedures for attribute selection could be computationally intensive and potentially unstable with small sample sizes. More discussion on the stability and robustness of the attribute selection process, along with empirical evidence, would be beneficial.
3. The literature review, while covering relevant works, could be more exhaustive. Incorporating additional recent studies on robust conformal inference and handling distributional shifts would provide a broader context and highlight the novelty of the proposed approach.

**Questions:**

What is the computational complexity of the proposed method, and how does it scale with large datasets?
Are there any optimization strategies to improve efficiency?

How stable is the attribute selection process across different sample sizes and datasets?
Can the method handle cases where multiple sensitive attributes need to be considered simultaneously?


Can the method be tested on more diverse and larger real-world datasets to validate its scalability and generalizability?
How does the method perform in scenarios with highly imbalanced datasets?

Have other attribute selection procedures been considered, and how do they compare with the proposed leave-one-out approach?

**Limitations:**

yes

---

> ### Author Rebuttal · Authors · 2024-08-06
>
> Thank you for your time and effort in reviewing our paper and providing constructive feedback. Please find our responses below.
>
> ## Computational Complexity Analysis
>
> As (perhaps too) briefly mentioned in Section 1.4, Appendix A6 provides a detailed computational complexity analysis for implementing our algorithms for either a single test sample or multiple test samples. Is this the information you were looking for? We will make the reference to this analysis clearer in the camera-ready version.
>
> Notably, our AFCP methods can be efficiently implemented for multiple test samples using a shortcut. Specifically, the identification of each subgroup in the calibration set does not rely on the test sample and can be reused to calculate group-wise FPRs and miscoverage rates for different test samples. For instance, this shortcut reduces the computational complexity of the AFCP method from $\mathcal{O}(mn\log n + nKMm)$ to $\mathcal{O}(n\log n + nm + MK(n + m))$ in outlier detection settings, where $n$ is the calibration set size, $m$ is the number of test samples, $K$ is the total number of sensitive attributes, and $M$ is the maximum count of groups across all attributes.
>
> ## Impact of Small Sample Sizes
>
> Although any attribute selection procedure may be less stable and effective with small sample sizes, we see this as an inherent challenge of small data sets rather than a flaw of our method. As mentioned in response to another comment, future work could explore the performance of our method in combination with different attribute selection procedures, as it is possible that alternative approaches may perform better in different settings, including with small sample sizes. However, the empirical performance of our AFCP method, as currently implemented, already surpasses that of other benchmark methods applied to the same data, in both large-sample and small-sample settings.
>
> For instance, in the experiments of Figures 3 and 4, it is challenging to accurately identify the attribute associated with significant algorithmic bias when the sample size is as small as 200. However, AFCP manages to select the correct attribute frequently enough to achieve significantly higher conditional coverage compared to the Marginal method, without substantially increasing the average prediction set size. Figure 4 further illustrates this behavior, showing the selection frequency of each attribute. As the sample size increases, our method begins to consistently select the correct attribute in all instances.
>
> These findings align with new experiments based on the COMPAS dataset, summarized in the attached one-page PDF.
>
> We will incorporate these explanations into the revised paper to emphasize the advantages of our methods with small sample sizes. Thank you for your comment!
>
> ## Simultaneous Selection of Multiple Attributes
>
> The appendix of our paper describes variations of our AFCP method that are designed to select and protect multiple attributes simultaneously. Specifically, in Appendix A2, we describe AFCP with multiple selected attributes in the multi-class classification setting, and in Appendix A4.3, we detail AFCP with multiple selected attributes in the outlier detection setting. Appendix 7.2.3 presents experimental results when multiple attributes can be selected using the real-world Adult Income Data. We will strive to better highlight these extensions of our method in the revised paper.
>
>
>
> ## Additional Experiments with Real Datasets
>
> Our proposed method is broadly applicable and can be efficiently applied for a variety of classification and outlier detection tasks. In this paper and appendices, we demonstrated its performance using two synthetic datasets and two real-world datasets. To supplement those results, in response to your feedback and the feedback of other reviewers, we have also conducted new experiments using the widely studied COMPAS dataset. The results obtained with the COMPAS dataset are summarized in the **new Figures 1-3** in the attached one-page PDF document. These additional experiments lead to conclusions that are qualitatively similar to those of the previous experiments. However, since this is a well-studied and interesting data sets, we think it may be valuable to include these new results in the revised paper.
>
> Regarding the applicability of our method to imbalanced data, please note that the new experiment on COMPAS data for recidivism prediction, for example, involve imbalanced data, with roughly 10\% of samples having the response label "High", 20\% having the label "Medium", and 70\% having the label "Low".
>
>
> ## Alternative Attribute Selection Procedures
>
> Thank you for this thoughtful question. While the underlying ideas of our method are indeed flexible enough to incorporate attribute selection procedures that go beyond those considered in this paper—a strength we will emphasize more in the revision—we believe that delving into the subtle empirical trade-offs associated with different selection algorithms is beyond the scope of this paper. We hope that our work will inspire future research to explore this question further.
>
>
> ## Additional References
>
> Following your advice, we will include more references to recent works on robust conformal inference under distribution shift. Although these works address different problems and take different perspectives, they are sufficiently related to merit mention in Section 1.5 (Related Works). The additional page allowed in the camera-ready manuscript will enable us to briefly discuss these references without sacrificing other content. We believe this addition will provide broader context on recent trends and highlight opportunities for integrating ideas in future research. Thank you for this helpful suggestion!

---

> > ### Comment · Reviewer_asc2 · 2024-08-12
> >
> > I thank the authors for the detailed response. I maintain my positive score of weak accept.

---

### Official Review · Reviewer_mW3F · 2024-07-13

**Soundness:** 2
**Presentation:** 2
**Contribution:** 3
**Rating:** 5
**Confidence:** 3

**Summary:**

The paper focuses on the problem of conformal inference with equalized coverage introduced in [1]. The authors propose a new method, Adaptively Fair Conformal Prediction (AFCP) that (i) adaptively selects a sensitive attribute corresponding to the group most negatively affected by algorithmic bias as evaluated by the miscoverage rate, and (ii) constructs a prediction set that satisfies equalized coverage over groups defined by the selected attribute. The authors perform experiments on synthetic and real data that demonstrate the performance of AFCP relative to methods that guarantee marginal coverage and exhaustive equalized coverage i.e., ​​valid coverage conditional on all sensitive attributes.

[1] Yaniv Romano, Rina Foygel Barber, Chiara Sabatti, and Emmanuel Candès. With malice toward none: Assessing uncertainty via equalized coverage. Harvard Data Science Review, 2020.

**Strengths:**

1. The problem of conformal inference with equalized coverage is an important problem and of interest to the community.
2. The trade-off between efficiency and equalized coverage addressed in the paper is challenging and of practical significance.
3. Empirical evaluation demonstrates the performance improvement of AFCP over baselines.

**Weaknesses:**

1. The clarity of the paper can be greatly improved. Detailed comments:
- p2 l37: "A limitation of the current method for conformal inference with equalized coverage...." -- what is the current method?
- p2 l41: what do you mean by efficiency and informativeness here? Is it set size? Please make it clear and precise.
- Section 1.1: Overall, it is hard to understand the motivation from this section. It would be helpful to rewrite this a bit e.g. l32-36: it is not clear how this conveys the rationale for conformal inference with equalized coverage different from rationale for conformal inference more generally.
- The notations are incorrect in some places and unnecessarily complex
   - p2 l71-72: $\phi$ is defined as mapping to $\mathbb{N}$ whereas the next line says it results in vector of length $|A|$. I believe it should be defined more generally
   - p2 l77-84: The notion of equalized coverage in [17] is defined as: $\mathbb{P}(Y_{n+1} \in C(X_{n+1}, A_{n+1}) | A_{n+1} = a) \geq 1 - \alpha$. It is not clear how this is extended to multiple (possibly overlapping) groups defined by $K$ sensitive attributes using $\phi$
- It is not clear at some places whether $A$ refers to single attribute or set
   - Alg 2 l4-5: in line 4, $A$ refers to single attribute, in line 5 it refers to multiple (set of) attributes
   - p6 l197: (7) returns a final selected attribute while the sentence refers to subset of attributes
- p6 l204: minor comment, this does not hyperlink to A1
- typos in table captions (Table A25 onwards)

2. Insufficient empirical evaluation: While the paper includes detailed analysis on the two selected datasets, both these setups are fairly synthetic. Also, the Nursery data seems to be from 1997; the paper lacks evaluation on any recent and common benchmark datasets in literature. The paper also seems to lack experiments that demonstrate coverage and efficiency performance in the presence of multiple sensitive attributes.

3. The AFCP1 variation of AFCP seems to outperform AFCP in all cases -- when the sample size is small, we still see undercoverage for the Blue group using AFCP (Fig 3) and AFCP1 is more robust by selecting at least one attribute. While the paper mentions this, what would be the advantage of using AFCP? Is there any procedure to select variations for different data regimes?

**Questions:**

1. The main text discusses the AFCP algorithm where at most one sensitive attribute may be selected. If we want equalized coverage over multiple attributes, won’t using AFCP result in similar challenges as exhaustive equalized coverage?

2. How does the size of the restricted calibration sample affect the performance of the algorithm?

**Limitations:**

The authors discuss the limitations in the Discussion section.

---

> ### Author Rebuttal · Authors · 2024-08-06
>
> Thank you for your detailed review and constructive feedback. We have addressed your questions and concerns point-by-point below.
>
> ## Additional experiments
>
> Might you have missed some experiments described in the Appendix and mentioned in Section 3.4? We will highlight these experiments more clearly in the main text. In any case, following your suggestion, we have also conducted new experiments using the Correctional Offender Management Profiling for Alternative Sanctions (COMPAS) dataset.
>
> **Experiments in the Appendix.** In addition to using the two datasets described in the main paper (one synthetic dataset and the nursery data), we had conducted experiments using an additional synthetic dataset (Appendix A7.2.2) and a common benchmark dataset, the Adult Income Dataset (Appendix A7.2.3). The experiments with the income data also include an AFCP extension that allows for the selection of multiple sensitive attributes. This extension is referred to as the "APCF+" method (depicted by the green lines) and is presented in Figures A24–A33. We will better highlight these experiments in the revised paper.
>
> **New experiments with COMPAS data.** The COMPAS dataset is used for multi-class classification tasks, predicting the risk of recidivism across three categories: 'High,' 'Medium,' and 'Low.' As illustrated in the **new Figures 1-2** in the attached PDF document, these additional experiments consistently demonstrate that our AFCP method outperforms other benchmarks. Specifically, it achieves higher conditional coverage than the Marginal method and, on average, produces smaller prediction sets than the Exhaustive and Partial methods. These results are consistent with those presented in the paper and will be includes in the revised paper. Thank you!
>
> ## Comparisons between AFCP and AFCP1
>
> Both AFCP and AFCP1 outperform benchmark approaches with small sample sizes, excelling in different areas. AFCP is better suited for scenarios where there is uncertainty about the presence of significant algorithmic bias. On the other hand, AFCP1 is more effective when there is prior knowledge that at least one attribute may be biased. This distinction will be clarified in the revised paper.
>
> In Figure 3, where one group (Color - Blue) is consistently biased, AFCP1 achieves slightly higher conditional coverage than AFCP. While AFCP shows slight undercoverage for the blue group with small sample sizes, it still performs better than the Marginal approach.
>
> AFCP's occasional inability to select a sensitive attribute with small sample sizes reflects the inherent challenges of small datasets rather than a flaw in the method. When the method does not select an attribute, it often indicates a lack of sufficient evidence of algorithmic bias, making it reasonable to calibrate the prediction sets only for marginal coverage.
>
> ## Comparison Between AFCP and the Exhaustive Approach
>
> Our AFCP method offers advantages over the Exhaustive approach, especially when the sample size is small and the specific attributes indicating biased groups are not known beforehand.
>
> For instance, in a dataset with five binary attributes where only two are significantly associated with algorithmic bias, an Exhaustive approach would have to consider all five attributes or all combinations of two attributes to ensure valid coverage. This process is inefficient. In contrast, our method can identify the two most relevant attributes in a data-driven way. It then focuses on calibrating the prediction sets within the subgroups formed by these attributes.
>
> ## Impact of Calibration Sample Size
>
> We are not completely sure about the meaning of "restricted calibration sample" in your question, but we will interpret it as the number of calibration samples belonging to the biased group identified by the selected attribute. Please correct us if this is not what you meant.
>
> In general, larger sample sizes make it intrinsically easier to identify (and thereby enable mitigating) significant algorithmic biases. Conversely, if the sample size within any group is small, it is more difficult to estimate the coverage within that group. This can result in our method selecting the “wrong” sensitive attribute. However, despite this unavoidable difficulty, the experiments demonstrate that our method performs well relative to other approaches both in small-sample and in large-sample settings.
>
> In particular, Figures 3 and 4 study the performance of our method as a function of the total number of samples in the training and calibration data. From these two plots we can see that when the sample size is as small as 200, it is challenging to select the true attribute because the data is subject to greater variability. Nevertheless, our method is able to select the “correct” attribute often enough to obtain informative prediction sets with significantly higher conditional coverage compared to the marginal approach.
>
> Figure 4 shows that, as the sample size increases, it becomes more likely for our method to select the correct sensitive attribute, as anticipated. The new COMPAS experiments we conducted following your suggestion demonstrate consistent behavior and confirm the practical advantages of our method.
>
> To better investigate the effect of “restricted” sample size for the biased group associated with the attribute selected by our method, we counted the number of data points in the protected group for each experiment and averaged the results over 100 independent experiments. The results are plotted in the **new Figure 3** in the attached PDF document. Please let us know if this answers your question.
>
> ## Clarity and Typos
> We will fix the typos and make the clarifications you suggested. Thank you!

---

> > ### Comment · Reviewer_mW3F · 2024-08-12
> >
> > Thank you for the explanation and clarification. I acknowledge the comparison between methods as I did in my previous comment. I am also not raising questions on the associated guarantees of conformal prediction, but it would be helpful to add accuracy metrics in the future versions to highlight whether this classification setting is challenging to begin with, and how the prediction sets provide more helpful information.
> >
> > In the light of the current discussion, I am willing to raise my score to 5.

---

> > > ### Author Response · Authors · 2024-08-12
> > >
> > > Thank you! We will include additional details on the COMPAS data analysis in the revised version of the paper, in support of the three new figures, including accuracy metrics. Please let us know if there is anything else that we should clarify at this point. (In the previous message we didn't realize there's still time until August 13th to discuss, if needed).

---

> ### Comment · Reviewer_mW3F · 2024-08-12
>
> I thank the authors for the detailed response. I will list some comments and questions I have below:
> 1. Thank you for the additional experiments. Feel free to correct me if I'm missing something, but it seems the COMPAS dataset has three classes. I am not sure if this is a good setup to study coverage -- e.g. average size is <=2 in Fig 1. How informative would coverage be in this case compared to accuracy? That said, I acknowledge the results.
>
> 2. "restricted calibration sample" is referred from the paper (e.g., l203)

---

> > ### Author Response · Authors · 2024-08-12
> >
> > Thank you for acknowledging our response. We’re pleased to see that our initial reply addressed your previous questions and concerns.
> >
> > We also appreciate your follow-up question about the COMPAS data analysis, though it’s unfortunate that it arrived so close to the deadline, particularly since the meaning of this question is not entirely clear to us.
> >
> > A 3-class classification problem is a reasonable and informative setting for comparing the performance of different conformal prediction methods. In this case, it shows that our method achieves the desired 90% coverage with prediction sets that are relatively small, averaging below 1.5 in size. It's important to emphasize this is an interesting result, which is not trivial at all to achieve in a 3-class context. In any case, what matters the most here is the comparison between the performances of different conformal prediction methods, which clearly shows the advantages of our approach.
> >
> > In general, an average prediction set size of 1.5 suggests high confidence (prediction sets of size of 1) in the majority of cases, and less informative predictions sets (sizes 2 and 3) only in a minority of cases where the model may be less accurate. Achieving this type of separation between confident and unconfident predictions is precisely what conformal prediction (and, more broadly, uncertainty quantification in machine learning) generally aims to do.

---

### Author Rebuttal · Authors · 2024-08-06

We are grateful to the four referees for their detailed reviews and constructive suggestions. We have addressed their questions and concerns point-by-point below. In addition to providing several clarifications, which can be easily reflected in the camera-ready manuscript to enhance its accessibility and completeness, we conducted additional experiments using the COMPAS [1] dataset. The results of these experiments, which align with those already described in the paper, are summarized in the three figures contained within the attached one-page PDF file.

How we have addressed the main points raised by the reviewers:

- **Extended Empirical Evaluation:** We clarified potentially unclear aspects of our empirical evaluations and highlighted additional experiments described in the Appendices that may have been previously overlooked. We also conducted further experiments using the COMPAS dataset, a well-known benchmark for algorithmic fairness research.

- **Clarifications on the Aims and Performance of the Attribute Selection Procedure:** We provided a more detailed discussion of the attribute selection component of our method, including its performance as a function of sample size. This discussion provides additional details, which will be incorporated in the revised manuscript, and explicitly highlights aspects of the existing figures and discussions that might have been previously overlooked.

- **Deeper Comparison Between Variations of Our Method:** We elaborated further on the relative strengths of the two variations of our method, AFCP and AFCP1. Additionally, we highlighted the presence of additional extensions described in the Appendices, designed to select more than one sensitive attribute simultaneously. These extensions may have been previously overlooked by some reviewers.

-  **Clarifications on Scalability and Complexity:** We pointed out that the Appendices include a detailed discussion of the computational complexity of our method, which may have been missed by one reviewer.

- **Opportunities for Further Research:** We acknowledged and commented on several intriguing suggestions for further extensions of our method and other opportunities for follow-up research proposed by the referees. We are grateful for these ideas and look forward to including their discussion in the camera-ready version of the paper.

- **Clarifications on Notation:** We gratefully acknowledged the presence of some typos and potentially confusing notations pointed out by one reviewer. These issues will be corrected in the camera-ready version of the paper.

Reference: [1] ProPublica. (2016). Compas recidivism risk score data and analysis. Retrieved from https://github.com/propublica/compas-analysis.

---

### Comment · Area_Chair_zmHW · 2024-08-12
**Author-Reviewer Discussion Phase**

Dear Authors and Reviewers!

Thank you for your reviews, rebuttal, additional comments, questions and responses!

We have the last two days of the discussion phase! Please use this time as efficiently as possible :)

Thank you,

NeurIPS 2024 Area Chair

---

### Decision · Program_Chairs · 2024-09-25

**Decision:**

Accept (poster)

**Comment:**

The paper concerns the problem of conformal predictions with fairness. The Authors introduces a new method that adaptively selects a sensitive attribute corresponding to the group most negatively affected by algorithmic bias. It then constructs a prediction set that ensures equalized coverage across groups defined by the selected attribute. Experiments on both simulated and real datasets are conducted to validate the proposed solution.

The reviewers are generally positive about the paper. They appreciate the proposed solution, along with its theoretical and empirical justification, and commend the clarity of the presentation. They also raised several critical comments including notational mistakes, unclear statements, areas of improvement in the empirical studies, or a limited discussion on computational complexity. During the discussion phase, the Authors successfully provided additional results and resolved the main concerns raised by the reviewers.